# Behind the scenes of streamflow model performance

Laurène J. E. Bouaziz[1,2], Fabrizio Fenicia[3], Guillaume Thirel[4], Tanja de Boer-Euser[1], Joost Buitink[5], Claudia C. Brauer[5], Jan De Niel[6], Benjamin J. Dewals[7], Gilles Drogue[8], Benjamin Grelier[8], Lieke A. Melsen[5], Sotirios Moustakas[6], Jiri Nossent[9,10], Fernando Pereira[9], Eric Sprokkereef[11], Jasper Stam[11], Albrecht H. Weerts[2,5], Patrick Willems[6,10], Hubert H. G. Savenije[1], and Markus Hrachowitz[1]

[1]Department of Water Management, Faculty of Civil Engineering and Geosciences, Delft University of Technology, P.O. Box 5048, NL-2600 GA Delft, The Netherlands
[2]Department Catchment and Urban Hydrology, Deltares, Boussinesqweg 1, 2629 HV Delft, The Netherlands
[3]Eawag, Überlandstrasse 133, CH-8600 Dübendorf, Switzerland
[4]Université Paris-Saclay, INRAE, UR HYCAR, 92160, Antony, France
[5]Hydrology and Quantitative Water Management Group, Wageningen University and Research, P.O. Box 47, 6700 AA Wageningen, The Netherlands
[6]Hydraulics division, Department of Civil Engineering, KU Leuven, Kasteelpark Arenberg 40, BE-3001 Leuven, Belgium
[7]Hydraulics in Environmental and Civil Engineering (HECE), University of Liege, Allée de la Découverte 9, 4000 Liege, Belgium
[8]Université de Lorraine, LOTERR, F-57000 Metz, France
[9]Flanders Hydraulics Research, Berchemlei 115, B-2140 Antwerp, Belgium
[10]Vrije Universiteit Brussel (VUB), Department of Hydrology and Hydraulic Engineering, Pleinlaan 2, 1050 Brussels, Belgium
[11]Ministry of Infrastructure and Water Management, Zuiderwagenplein 2, 8224 AD Lelystad, The Netherlands

**Correspondence:** Laurène Bouaziz (L.J.E.Bouaziz@tudelft.nl)

**Abstract.** Streamflow is often the only variable used to evaluate hydrological models. In a previous international comparison study, eight research groups followed an identical protocol to calibrate twelve hydrological models using observed streamflow of catchments within the Meuse basin. In the current study, we hypothesize that these twelve process-based models with similar streamflow performance have similar representations of internal states and fluxes. Next, we assess model behavior plausibility by ranking the models for a set of criteria using remote sensing products of evaporation, snow cover, soil moisture and total storage anomalies. We found substantial dissimilarities between models for annual interception and seasonal evaporation rates, the annual number of days with water stored as snow, the mean annual maximum snow storage and the size of the root-zone storage capacity. These differences in internal process representation imply that these models cannot all simultaneously be close to reality. Modeled annual evaporation rates are consistent with GLEAM estimates. However, there is a large uncertainty in modeled and remote sensing annual interception. Substantial differences are also found between MODIS and modeled number of days with snow storage. Models with relatively small root-zone storage capacities and without root water uptake reduction under dry conditions tend to have an empty root-zone storage for several days each summer, while this is not suggested by remote sensing data of evaporation, soil moisture and vegetation indices. On the other hand, models with relatively large root-zone storage capacities tend to overestimate very dry total storage anomalies of GRACE. None of the models is systematically consistent with the information available from all different (remote sensing) data sources. Yet, we did not reject models given the uncertainties in these data sources and their changing relevance for the system under investigation.

# 1 Introduction

Hydrological models are valuable tools for short-term forecasting of river flows, long-term predictions for strategic water management planning but also to develop a better understanding of the complex interactions of water storage and release processes at the catchment-scale. In spite of the wide variety of existing hydrological models, they mostly include similar functionalities of storage, transmission and release of water to represent the dominant hydrological processes of a particular river basin (Fenicia et al., 2011), differing mostly only in the detail of their parametrizations (Gupta et al., 2012; Gupta and Nearing, 2014; Hrachowitz and Clark, 2017).

In all of these models, each individual model component constitutes a separate hypothesis of how water moves through that specific part of the system. Frequently, the individual hypotheses remain untested. Instead only the model output, i.e. the aggregated response of these multiple hypotheses, is confronted with data of the aggregated response of a catchment to atmospheric forcing. Countless applications of different hydrological models in many different regions across the world over the last decades have shown that these models often provide relatively robust estimates of streamflow dynamics, for both calibration and evaluation periods. However, various combinations of different untested individual hypotheses, can and do lead to similar aggregated outputs, i.e. model equifinality (Beven, 2006; Clark et al., 2016).

To be useful for any of the above applications, it is thus of critical importance that not only the aggregated but also the individual behaviors of the respective hypotheses are consistent with their real-world equivalents. Given the complexity and heterogeneity of natural systems together with the general lack of suitable observations, this remains a major challenge in hydrology (e.g., Jakeman and Hornberger, 1993; Beven, 2000; Gupta et al., 2008; Andréassian et al., 2012).

Studies have addressed the issue by constraining the parameters of specific models through the use of additional data sources besides streamflow. Beven and Kirkby (1979); Güntner et al. (1999) and Blazkova et al. (2002) mapped saturated contributing areas during field surveys to constrain model parameters, while patterns of water tables in piezometers were used by Seibert et al. (1997); Lamb et al. (1998) and Blazkova et al. (2002). Other sources include satellite-based total water storage anomalies (Winsemius et al., 2006; Werth and Güntner, 2010; Yassin et al., 2017), evaporation (Livneh and Lettenmaier, 2012; Rakovec et al., 2016a; Bouaziz et al., 2018; Demirel et al., 2018; Hulsman et al., 2019), near-surface soil moisture (Franks et al., 1998; Brocca et al., 2010; Sutanudjaja et al., 2014; Adnan et al., 2016; Kunnath-Poovakka et al., 2016; López López et al., 2017; Bouaziz et al., 2020), snow cover information (Gao et al., 2017; Bennett et al., 2019; Riboust et al., 2019), or a combination of these variables (Nijzink et al., 2018; Dembélé et al., 2020). Reflecting the results of many studies, Rakovec et al. (2016b) showed that streamflow is necessary but not sufficient to constrain model components to warrant partitioning of incoming precipitation to storage, evaporation and drainage.

Hydrological simulations are, however, not only affected by model parameter uncertainty, but also by the selection of a model structure and its parameterization (i.e. the choice of equations). Modeling efforts over the last four decades have led to a wide variety of hydrological models providing flexibility to test competing modeling philosophies, from spatially lumped model representations of the system to high-resolution small-scale processes numerically integrated to the catchment scale (Hrachowitz and Clark, 2017). Haddeland et al. (2011) and Schewe et al. (2014) compared global hydrological models and

found that differences between models are a major source of uncertainty. Nonetheless, model selection is often driven by personal preference and experience of individual modelers rather than detailed model test procedures (Holländer et al., 2009; Clark et al., 2015; Addor and Melsen, 2019).

A suite of comparison experiments tested and explored differences between alternative modeling structures and parameter-
izations (Perrin et al., 2001; Reed et al., 2004; Duan et al., 2006; Holländer et al., 2009; Knoben et al., 2020). However, these studies mostly restricted themselves to analyses of the models' skills to reproduce streamflow ("aggregated hypothesis"), with little consideration for the model internal processes ("individual hypotheses"). The Framework for Understanding Structural Errors (FUSE) was one of the first initiatives towards a more comprehensive assessment of model structural errors, with special consideration given to individual hypotheses (Clark et al., 2008).

Subsequent efforts towards more rigorous testing of competing model hypotheses, partially based on internal processes include Smith et al. (2012a, b) who tested multiple models for their ability to reproduce in-situ soil moisture observations as part of the Distributed Model Intercomparison Project 2 (DMIP2). They found that only two out of sixteen models provided reasonable estimates of soil moisture. In a similar effort, Koch et al. (2016) and Orth et al. (2015) also compared modeled soil moisture to in-situ observations of soil moisture for a range of hydrological models in different environments. In contrast,
Fenicia et al. (2008) and Hrachowitz et al. (2014) used groundwater observations to test individual components of their models.

Here, in this model comparison study, we instead use globally available remote sensing data to evaluate five different model state and flux variables of twelve process-based models with similar overall streamflow performance, which are calibrated by several research groups following an identical protocol. The calibration on streamflow was conducted in our previous study (de Boer-Euser et al., 2017), in which eight research groups working on the Meuse basin applied their rainfall-runoff model(s)
according to a defined protocol using the same forcing data to reduce the degrees of freedom and enable a fair comparison (Ceola et al., 2015). All models had a high overall streamflow performance based on commonly used metrics. We were able to attribute differences in performance to model structure components by focusing on specific hydrological events (de Boer-Euser et al., 2017). Our analyses were then limited to comparisons with hourly streamflow observations and the modeled response of internal processes remained unused.

In a direct follow-up of the above study, we here hypothesize that process-based models with similar overall streamflow performance rely on similar representations of their internal states and fluxes. We test our hypothesis by quantifying the differences in the magnitudes and dynamics of five internal state and flux variables of twelve models. Our primary aim is to test if models calibrated to streamflow with similar high-performance levels in reproducing streamflow, follow similar pathways to do so, i.e. represent the system in a similar way. A secondary objective is to evaluate the plausibility of model behavior by
introducing a set of "soft" measures based on expert knowledge in combination with remote sensing data of evaporation, snow cover, soil moisture and total water storage anomalies.

## 2 Study area

We test our hypothesis using data from three catchments in the Belgian Ardennes; all of them are part of the Meuse River basin in North-West Europe: the Ourthe upstream of Tabreux (ID1), the nested Ourthe Orientale upstream of Mabompré (ID2) and the Semois upstream of Membre-Pont (ID3), as shown in Figure 1a,b. The Ardennes Massif and Plateau are underlain by relatively impermeable metamorphic Cambrian rock and Early Devonian sandstone. The pronounced streamflow seasonality of these catchments is driven by high summer and low winter evaporation (defined here as the sum of all evaporation components including transpiration, soil evaporation, interception, sublimation and open water evaporation when applicable), as precipitation is relatively constant throughout the year. Snow is not a major component of the water balance, but occurs almost every year with mean annual number of days with precipitation as snow estimated between 35 and 40 days $yr^{-1}$ (Royal Meteorological Institute Belgium, 2015). Even if mean annual snow storage is relatively small, snow can be important for specific events. For example in 2011, when rain on snow caused widespread flooding in these catchments.

The rain-fed Ourthe River at Tabreux (ID1) is fast-responding due to shallow soils and steep slopes in the catchment. Agriculture is the main land cover (27 % crops and 21 % pasture), followed by 46 % forestry and 6 % urban cover in an area of 1607 $km^2$ and an elevation ranging between 107 m and 663 m (de Boer-Euser et al., 2017). Mean annual precipitation, potential evaporation and streamflow are 979 mm $yr^{-1}$, 730 mm $yr^{-1}$ and 433 mm $yr^{-1}$ respectively for the period 2001–2017.

Nested within the Ourthe catchment (ID1), the Ourthe Orientale upstream of Mabompré (ID2) is characterized by a narrow elevation range from 294 m to 662 m, with 65 % of the catchment falling within a 100 m elevation band, making this catchment suitable to analyze snow processes modeled by lumped models. The Ourthe Orientale upstream of Mabompré has an area of 317 $km^2$ which corresponds to 20 % of the Ourthe area upstream of Tabreux and has similar land cover fractions. Mean annual precipitation, potential evaporation and streamflow for the period 2001–2017 are also relatively similar with 1052 mm $yr^{-1}$, 720 mm $yr^{-1}$ and 462 mm $yr^{-1}$, respectively.

Forest is the main land cover in the Semois upstream of Membre-Pont (ID3) with 56 %, followed by agriculture (18 % pasture and 21 % crop) and 5 % urban cover. The Semois upstream of Membre-Pont is 24 % smaller than the Ourthe upstream of Tabreux with 1226 $km^2$ and elevation ranges between 176 m and 569 m. Mean annual precipitation, potential evaporation and streamflow are respectively 38 %, 4 % and 46 % higher in the Semois at Membre-Pont with 1352 mm $yr^{-1}$, 759 mm $yr^{-1}$ and 634 mm $yr^{-1}$.

## 3 Data

### 3.1 Hydrological and meteorological data

Hourly precipitation gauge data are provided by the Service Public de Wallonie (Service Public de Wallonie, 2018) and are spatially interpolated using Thiessen polygons for the period 2000-2017. Daily minimum and maximum temperatures are retrieved from the 0.25° resolution gridded E-OBS dataset (Haylock et al., 2008) and disaggregated to hourly values by linear interpolation using the timing of daily minimum and maximum radiation at Maastricht (Royal Netherlands Meteorological

Institute, 2018). Daily potential evaporation is calculated from daily minimum and maximum temperatures using the Hargreaves formula (Hargreaves and Samani, 1985) and is disaggregated to hourly values using a sine function during the day and no evaporation at night. We use the same forcing for 2000–2010 as in the previous comparison study (de Boer-Euser et al., 2017) and follow the same approach to extend the dataset for the period 2011–2017. Uncertainty in meteorological data is not explicitly considered, but our primary aim is to compare the models forced with identical data. Observed hourly streamflow data for the Ourthe at Tabreux, Ourthe Orientale at Mabompré and Semois at Membre-Pont are provided by the Service Public de Wallonie for the period 2000–2017.

## 3.2 Remote sensing data

### 3.2.1 GLEAM evaporation

The Global Land Evaporation Amsterdam Model (GLEAM, Miralles et al., 2011; Martens et al., 2017) provides daily estimates of land evaporation by maximizing the information recovery on evaporation contained in climate and environmental satellite observations. The Priestley and Taylor (1972) equation is used to calculate potential evaporation for bare soil, short canopy and tall canopy land fractions. Actual evaporation is the sum of interception and potential evaporation reduced by a stress factor. This evaporative stress factor is based on microwave observations of vegetation optical depth and estimates of root-zone soil moisture in a multi-layer water-balance model. Interception evaporation is estimated separately using a Gash analytical model and only depends on precipitation and vegetation characteristics. GLEAM v3.3a relies on reanalysis net radiation and air temperature from the European Centre for Medium-Range Weather Forecasts (ECMWF) ERA5 data, satellite and gauge-based precipitation, satellite-based vegetation optical depth, soil moisture and snow water equivalent. The data are available at $0.25°$ resolution (Figure 1b) and account for subgrid heterogeneity by considering three land cover types. We spatially average GLEAM interception and total actual evaporation estimates over the Ourthe catchment upstream of Tabreux for the period 2001–2017.

### 3.2.2 MODIS Snow Cover

The Moderate Resolution Imaging Spectroradiometer (MODIS) AQUA (MYD10A1, version 6) and TERRA (MOD10A1, version 6) satellites provide daily maps of the areal fraction of snow cover per 500 m $\times$ 500 m cell (Figure 1b) based on the Normalized Difference Snow Index (Hall and Riggs, 2016a, b). For each day, AQUA and TERRA observations are merged into a single observation by taking the mean fraction of snow cover per day. The percentage of cells with a fractional snow cover larger than zero and fraction of cells without missing data (i.e. due to cloud cover) for the catchment of the Ourthe Orientale upstream of Mabompré is calculated for each day. For this study, we disregard observations during the summer months (JJA, when temperatures did not drop below $4°C$) and only use daily observations in which at least 40 % of the catchment area has snow cover retrievals not affected by clouds, implying that we have 1463 valid daily observations of mean fractional snow cover. This corresponds to 32 % of all observations of the Ourthe Orientale catchment upstream of Mabompré between 2001 and 2017.

### 3.2.3 SCATSAR-SWI1km Soil Water Index

SCATSAR-SWI1km is a daily product of soil water content relative to saturation at a 1 km $\times$ 1 km resolution (Figure 1b) obtained by fusing spatio-temporally complementary radar sensors (Bauer-Marschallinger et al., 2018). Estimates of the moisture content relative to saturation at various depths in the soil, referred to as Soil Water Index (SWI), are obtained through temporal filtering of the 25 km METOP ASCAT near-surface soil moisture (Wagner et al., 2013) and 1 km Sentinel-1 near-surface soil moisture (Bauer-Marschallinger et al., 2018). The Soil Water Index features as single parameter the characteristic time length $T$ (Wagner et al., 1999; Albergel et al., 2008). The $T$-value is required to convert near-surface soil moisture observations to estimates of root-zone soil moisture. The $T$-value increases with increasing root-zone storage capacities (Bouaziz et al., 2020), resulting in more smoothing and delaying of the near-surface soil moisture signal. The Copernicus Global Land Service (2019) provides the Soil Water Index for $T$-values of 2, 5, 10, 15, 20, 40, 60 and 100 days. Since Sentinel-1 was launched in 2014, the Soil Water Index is available for the period 2015–2017. We calculate the mean soil moisture over all SCATSAR-SWI1km pixels within the Ourthe upstream of Tabreux for the available period.

### 3.2.4 GRACE Total Water Storage anomalies

The Gravity Recovery and Climate Experiment (GRACE, Swenson and Wahr, 2006; Swenson, 2012) twin satellites, launched in March 2002, measure the Earth's gravity field changes by calculating the changes in the distance between the two satellites as they move one behind the other in the same orbital plane. Monthly total water storage anomalies (in mm) relative to the 2004–2009 time-mean baseline are provided at a spatial sampling of 1° (approximately 78 km x 110 km at the latitude of the study region, Figure 1b) by three centers: U. Texas / Center for Space Research (CSR), GeoForschungsZentrum Potsdam (GFZ) and Jet Propulsion Laboratory (JPL). These centers apply different processing strategies which lead to variations in the gravity fields. These gravity fields require smoothing of the noise induced by attenuated short wavelength. The spatial smoothing decreases the already coarse GRACE resolution even further through signal "leakage" of one location to surrounding areas (Bonin and Chambers, 2013), which increases the uncertainty especially at the relatively small scale of our study catchments. We apply the scaling coefficients provided by NASA to restore some of the signal loss due to processing of GRACE observations (Landerer and Swenson, 2012). The data of the three processing centers are each spatially averaged over the catchments of the Ourthe upstream of Tabreux and the Semois upstream of Membre-Pont for the period April 2002 to February 2017.

### 3.3 Data uncertainty

The hydrological evaluation data are all subject to uncertainties (Beven, 2019). Streamflow is not measured directly but depends on water level measurements and a rating curve. Westerberg et al. (2016) quantify a median streamflow uncertainty of $\pm 12\,\%$, $\pm 24\,\%$ and $\pm 34\,\%$ for average, high and low streamflow conditions, respectively, using a Monte Carlo sampling approach of multiple feasible rating curve for 43 UK catchments. We sample from these uncertainty ranges to transform the streamflow observations (100 realizations). We then quantify signature uncertainty originating from streamflow data uncertainty using the 100 sampled time series for a selection of streamflow signatures (Section 4.2). The 5-95[th] uncertainty bounds of median annual

streamflow, baseflow and flashiness indices result in $\pm 11\%$, $\pm 9\%$ and $\pm 12\%$, respectively. These magnitudes are similar to those reported by Westerberg et al. (2016).

GLEAM evaporation estimates are inferred from models and forcing data which are all affected by uncertainty. Yet, uncertainty estimates of GLEAM evaporation are not available. However, GLEAM evaporation was evaluated against FLUXNET data by Miralles et al. (2011). For the nearby station of Lonzee in Belgium, they report similar annual rates and a high correlation coefficient of 0.91 between the daily time series. GLEAM mean annual evaporation was compared to the ensemble mean of five evaporation datasets in Miralles et al. (2016) and shows higher than average values in Europe (of approximately 60 mm yr$^{-1}$ or 10 % of mean annual rates for our study area). The partitioning of evaporation in different components (transpiration, interception and soil evaporation) differs substantially between different evaporation datasets, as shown by Miralles et al. (2016). GLEAM interception currently only considers tall vegetation and underestimates in-situ data (Zhong et al., 2020) and is $\sim$50 % lower than estimates from other datasets (Miralles et al., 2016). These uncertainties underline that GLEAM (and other remote sensing data) cannot be considered as a reliable representation of real-world quantities. However, the comparison of daily dynamics and absolute values of this independent data source with modeled results is still valuable to detect potential outliers and understand their behavior. Besides, the different methods used to estimate potential evaporation of GLEAM and our model forcing should not impede us from testing the consistency between the resulting actual evaporation (Oudin et al., 2005).

Most frequent errors within the MODIS snow cover products are due to cloud/snow discrimination problems. Daily MODIS snow maps have an accuracy of approximately 93 % at the *pixel scale*, with lower accuracy in forested areas, complex terrain and when snow is thin and ephemeral and higher accuracy in agricultural areas (Hall and Riggs, 2007). However, here, MODIS data is used to estimate the number of days with snow at the *catchment scale*. We expect lower classification errors at the catchment scale as it would require many pixels to be misclassified at the same time. For each day and each pixel of valid MODIS observations, we sample from a binomial distribution with a probability of 93 % that MODIS is correct when the pixel is classified as snow and assume a higher probability of 99 % that MODIS is correct when the pixel is classified as no-snow to prevent overestimating snow for days without snow (Ault et al., 2006; Parajka and Blöschl, 2006). We repeat the experiment for 1000 times in a Monte Carlo procedure. This results in less than $\pm 2\%$ uncertainty in the number of days when MODIS observes snow at the catchment scale.

The soil water content relative to saturation of SCATSAR-SWI1km is estimated from observed radar backscatter through a change detection approach, which interprets changes in backscatter as changes in soil moisture, while other surface properties are assumed static (Wagner et al., 1999). The degree of saturation of the near-surface is given in relative units from 0 % (dry reference) to 100 % (wet reference) and is converted to deeper layers through an exponential filter called the Soil Water Index. The smoothing and delaying effect of the Soil Water Index narrows the range of the near-surface degree of saturation. Therefore, data matching techniques are often used to rescale satellite data to match the variability of modeled or observed data (Brocca et al., 2010), which suggests the difficulty to meaningfully compare the *range* of modeled and remote sensing estimates of root-zone soil moisture content relative to saturation. However, the *dynamics* of SCATSAR-SWI1km data have been evaluated against in-situ stations of the International Soil Moisture Network, despite commensurability issues of comparing in-situ point

measurements and areal satellite data. Spearman rank correlation coefficients of 0.56 are reported for $T$-values up to 15 days and 0.43 for $T$-value of 100 days (Bauer-Marschallinger, 2020).

GRACE estimates of total water storage anomalies suffer from signal degradation due to measurement errors and noise. Filtering approaches are applied to reduce these errors, but induce leakage of signal from surrounding areas. The uncertainty decreases as the size of the region under consideration increases. However, time series of a single pixel may still be used to compare dynamics and amplitudes of total water storage anomalies despite possible large uncertainty (Landerer and Swenson, 2012). We estimate an uncertainty in total water storage anomalies of ∼18 mm in the pixels of our catchments by combining

measurement and leakage errors in quadrature, which are both provided for each grid location (Landerer and Swenson, 2012).

## 4    Methods

### 4.1    Models and Protocol

Eight research groups (Wageningen University, Université de Lorraine, Leuven University, Delft University of Technology, Deltares, Irstea (now INRAE), Eawag and Flanders Hydraulics Research) participated in the comparison experiment and ap-

plied one or several hydrological models (Figure 2). The models include WALRUS (Wageningen Lowland Runoff Simulator, Brauer et al., 2014a, b), PRESAGES (PREvision et Simulation pour l'Annonce et la Gestion des Etiages Sévères, Lang et al., 2006), VHM (Veralgemeend conceptueel Hydrologisch Model, Willems, 2014), FLEX-Topo which was still under development when it was calibrated for our previous study (Savenije, 2010; de Boer-Euser et al., 2017; de Boer-Euser, 2017), a distributed version of the HBV model (Hydrologiska Byråns Vattenbalansavdelning, Lindström et al., 1997), SUPERFLEX

M2 to M5 models (Fenicia et al., 2011, 2014), dS2 (distributed simple dynamical systems, Buitink et al., 2019), GR4H (Génie Rural à 4 paramètres Horaire, Mathevet, 2005; Coron et al., 2017, 2019) combined with the CemaNeige snow module (Valéry et al., 2014) and NAM (NedborAfstrommings Model, Nielsen and Hansen, 1973). Main differences and similarities between models in terms of snow processes, root-zone storage, total storage and evaporation processes are summarized in Tables 1-3.

    In our previous study (de Boer-Euser et al., 2017), we defined a modeling protocol to limit the degrees of freedom in the

modeling decisions of the individual participants (Ceola et al., 2015), allowing us to meaningfully compare the model results. The protocol involved forcing the models with the same input data and calibrating them for the same time period, using the same objective functions. However, participants were free to choose a parameter search method, as we considered it to be part of the modelers experience with the model, even if this would make comparison less straightforward. The models were previously calibrated using streamflow of the Ourthe at Tabreux for the period 2004 to 2007, using 2003 as a warm-up year (de Boer-Euser

et al., 2017). The Nash-Sutcliffe efficiencies of the streamflow and the logarithms of the streamflow were simultaneously used as objective functions to select an ensemble of feasible parameter sets to account for parameter uncertainty and ensure a balance between the models' ability to reproduce both high and low flows. The models were subsequently tested and evaluated for the periods 2001 to 2003 and 2008 to 2010. In addition, by carrying out a proxy-basin differential split-sample test (Klemeš, 1986), not only the models' temporal but also their spatial transferability was tested by applying the calibrated model parameter sets

to nested and neighboring catchments for the period 2001 to 2017, using 2000 as a warm-up year. Results thereof are presented in de Boer-Euser et al. (2017)

In the current study, we run the calibrated models for an additional period from 2011 to 2017 for the Ourthe at Tabreux (ID1), the Ourthe Orientale at Mabompré (ID2) and the Semois at Membre-Pont (ID3). The modeling groups have provided simulation results for each catchment in terms of streamflow, groundwater losses/gains, interception evaporation, root-zone 250 evaporation (transpiration and soil evaporation), total actual evaporation, snow storage, root-zone storage and total storage as a sum of all model storage volumes (Table 2) at an hourly time step for the total period 2001–2017. We compare these modeled states and fluxes and evaluate them against their remote sensing equivalents as further explained in Sections 4.2 and 4.3.

## 4.2   Model evaluation: water balance

All models are evaluated in terms of the long-term water balance, which indicates the partitioning between drainage and 255 evaporative fluxes and allows us to assess long-term conservation of water and energy. We compare mean annual streamflow with observations and mean annual actual evaporation and interception evaporation with GLEAM estimates for the Ourthe at Tabreux during the evaluation period 2008–2017. A detailed description of streamflow performance for specific events (low and high flows, snowmelt event, transition from dry to wet period) has been detailed in the previous paper (de Boer-Euser et al., 2017). In the current study, differences in streamflow dynamics are briefly summarized by assessing observed and modeled 260 baseflow indices ($I_{\text{baseflow}}$, van Dijk, 2010) and flashiness indices ($I_{\text{flashiness}}$, Fenicia et al., 2016), as these are representative of the partitioning of drainage into fast and slow responses. Seasonal dynamics of actual evaporation over potential evaporation and runoff coefficients during winter (Oct-Mar) and summer (Apr-Sep) are compared between models.

## 4.3   Model evaluation: internal states

We compare modeled snow storage, root-zone soil moisture and total storage between models and with remote sensing es-265 timates of MODIS snow cover, SCATSAR-SWI1km Soil Water Index and GRACE total storage anomalies, respectively, as shown in Tables 2-3 and Figure 1c.

### 4.3.1   Snow days

As most models used in this study are lumped, it is not possible to spatially evaluate modeled snow cover versus MODIS snow cover. However, we can classify each day in a binary way according to the occurrence of snow, based on a threshold for the 270 percentage of cells in the catchment where snow cover is detected. MODIS snow cover observations are classified as days with and without snow using thresholds of both 10 and 15 % of snow-covered cells in the catchment to be counted as a day with snow, in a sensitivity analysis. For each model, snow days are distinguished from non-snow days whenever the water stored as snow is above 0.05 mm to account for numerical rounding. For each model (and each retained parameter set), we then compare if modeled snow coincides with 'truly' observed snow by MODIS, for each day with a valid MODIS observation. 275 We create a confusion matrix with counts of true positives when observations and model results agree on the presence of snow

(hits), false positives when the model indicates the presence of snow but this is not observed by MODIS (false alarms), false negatives when the model misses the presence of snow observed by MODIS (miss) and true negatives when observations and model results agree on the absence of snow (correct rejections). From this matrix, we calculate the recall as the ratio of hits over actual positives (number of days when snow is observed by MODIS) and the precision as the ratio of hits over predicted
positives (number of days when snow is modeled). This allows us to identify, on the one hand, the ratio of days when snow observed by MODIS is correctly identified by the model and, on the other hand, the ratio of days when snow is modeled that is actually observed by MODIS. We therefore not only account for hits, but also for false alarms between model and remote sensing observations. We also compare annual maximum snow storage and number of days with snow between the seven models with a snow module (GR4H, M5, NAM, wflow_hbv, M4, FLEX-Topo, WALRUS). The snow analysis is performed
in the catchment of the Ourthe Orientale upstream of Mabompré as it features the narrowest elevation range among the study catchments (i.e. 294-662 m a.s.l. versus 108-662 m for the Ourthe upstream of Tabreux) and thus plausibly permits a lumped representation of the snow component.

### 4.3.2  Root-zone soil moisture

We compare the range of relative root-zone soil moisture ($\overline{S}_\mathrm{R} = S_\mathrm{R}/S_\mathrm{R,max}$) between models for the period in which SCATSAR-
290 SWI1km is available (2015–2017). Time series of catchment-scale root-zone soil moisture are available for all models except WALRUS and dS2 as these models have a combined soil reservoir (Figure 2). The dS2 model only relies on the sensitivity of streamflow to changes in total storage. In WALRUS, the state of the soil reservoir (which includes the root zone) is expressed as a storage deficit and is therefore not bound by an upper limit (Table 2). Root-zone storage capacities ($S_\mathrm{R,max}$, mm) are available as calibration parameter for all other models. We relate the range in relative root-zone soil moisture to the maximum
root-zone storage capacity $S_\mathrm{R,max}$, because we expect models with small root-zone storage capacities $S_\mathrm{R,max}$ to entirely utilize the available storage, through complete drying and saturation.

We then compare the similarity of the dynamics of modeled time series of the relative root-zone soil moisture with remotely sensed SCATSAR-SWI1km Soil Water Index for several values of the characteristic time length parameter ($T$ in days). The $T$-value has previously been positively correlated with root-zone storage capacity, assuming a high temporal variability of root-
300 zone soil moisture and therefore a low $T$-value for small root-zone storage capacities $S_\mathrm{R,max}$ (Bouaziz et al., 2020). For each model and feasible realization, we identify the $T$-value that yields the highest Spearman rank correlation between modeled root-zone soil moisture and Soil Water Index. We then relate the optimal $T$-value to the root-zone storage capacity $S_\mathrm{R,max}$. This analysis enables us to identify potential differences in the representation and the dynamics of root-zone storage between models.

### 305  4.3.3  Total storage anomalies

For each model, we calculate time series of total storage (Table 2) and mean monthly total storage anomalies relative to the 2004-2009 time-mean baseline for comparison with GRACE estimates for the Ourthe upstream of Tabreux (ID1) and the Semois upstream of Membre-Pont (ID3). Both catchments coincide with two neighboring GRACE cells, allowing us to test

how well the models reproduce the observed spatial variability. We further relate the modeled range of total storage (maximum
minus minimum total storage over the time series) to Spearman rank correlation coefficients between modeled and GRACE
estimates of total storage anomalies.

### 4.4 Interactions between storage and fluxes during dry periods

The impact of a relatively large (> 200 mm) versus relatively small (< 150 mm) root-zone storage capacity on actual evapo-
ration, streamflow and total storage is assessed during a dry period in September 2016 by selecting two representative models
with high streamflow model performance (GR4H and M5). The plausibility of the hydrological response of these two model
representations is evaluated against remote sensing estimates of root-zone soil moisture and actual evaporation.

### 4.5 Plausibility of process representations

The models are subsequently ranked and evaluated in terms of their consistency with observed streamflow, remote sensing
data and expert knowledge with due consideration of the uncertainty in the evaluation data, as detailed in Section 3.3. We
evaluate the models in terms of their deviations around median annual streamflow, flashiness and baseflow indices, median
annual actual evaporation and interception compared to GLEAM estimates, the number of days with snow over valid MODIS
observations, the number of days per year with empty root-zone storage and the very dry total storage anomalies compared to
GRACE estimates.

## 5 Results

### 5.1 Water balance

#### 5.1.1 Streamflow

All models show high Nash-Sutcliffe Efficiencies of the streamflow and the logarithm of the streamflow ($E_{\mathrm{NS,Q}}$ and $E_{\mathrm{NS,logQ}}$)
with median values of above 0.7 for the post-calibration evaluation period 2008–2017 (Figure 3a and Table 2 for the calculation
of the Euclidean distances). The interannual variability of streamflow agrees strongly with observations for each model in the
period 2008–2017 (Figure 3b). The difference between modeled and observed median streamflow varies between -5.6 % and
5.6 % and the difference in total range varies between -9.6 % and 20 %. This is in line with our results in the previous paper, in
which we also showed that all models perform well in terms of commonly used metrics (de Boer-Euser et al., 2017). However,
there are differences in the partitioning of fast and slow runoff, as shown by the flashiness and baseflow indices ($I_{\mathrm{flashiness}}$ and
$I_{\mathrm{baseflow}}$) in Figure 3c. Largest underestimation of the flashiness index occurs for M2 and dS2 ($\sim$20 %), while FLEX-Topo
shows the highest overestimation (26 %). FLEX-Topo and WALRUS underestimate the baseflow index most (41 % and 70 %
respectively), while GR4H and M5 show the highest overestimation (15 % and 21 % respectively). There is a strong similarity
between modeled and observed hydrographs for one of the best performing models M5, as quantified by its low Euclidean

distance (Figure 3d and Table 2). The GR4H model is the only model which includes deep groundwater losses, but they are very limited and represent only 1.6 % of total modeled streamflow of the Ourthe at Tabreux, or approximately 7 mm yr$^{-1}$.

### 5.1.2 Actual evaporation

Modeled median annual actual evaporation $E_A$ (computed as the sum of soil evaporation, transpiration, (separate) interception evaporation and, if applicable, sublimation, Table 3) for hydrological years between October 2008 and September 2017 varies between 507 and 707 mm yr$^{-1}$ across models, with a median of 522 mm yr$^{-1}$, which is approximately 10 % lower than the GLEAM estimate of 578 mm yr$^{-1}$, as shown in Figure 4a. Annual actual evaporation of the VHM model is very high compared

to the other models, with a median of 707 mm yr$^{-1}$ and approximates potential evaporation (median of 732 mm yr$^{-1}$). Calibration of the VHM model is meant to follow a manual stepwise procedure including the closure of the water balance during the identification of soil moisture processes (Willems, 2014). However, in the automatic calibration prescribed by the current protocol, this step was not performed, which explains the unusual high actual evaporation in spite of relatively similar annual streamflow compared to the other models, as there is no closure of the water balance (Figure 3a).

Interception evaporation is included in four models, with GR4H showing the lowest annual interception evaporation of 100 mm yr$^{-1}$ (19 % of $E_A$ or 10 % of $P$), FLEX-Topo and wflow_hbv have relatively similar amounts of approximatively 250 mm yr$^{-1}$ ($\sim$45 % of $E_A$ or 26 % of $P$) and NAM has the highest annual interception evaporation of 340 mm yr$^{-1}$ (65 % of $E_A$ or 36 % of $P$), as shown in Figure 4a. Differences are related to the presence and maximum size of the interception storage ($I_{max}$), as shown in Table 3. GLEAM interception estimates of 189 mm yr$^{-1}$ are almost twice as high as GR4H

estimates, 25 % lower than FLEX-Topo and wflow_hbv, and 44 % lower than NAM values, suggesting a large uncertainty in the contribution of interception and transpiration to actual evaporation. For comparison, measurements of the fraction of interception evaporation over precipitation in forested areas vary significantly depending on the site location, with estimates of 37 % for a Douglas fir stand in the Netherlands (Cisneros Vaca et al., 2018), 27 %, 32 % and 42 % for three coniferous forests of Great Britain (Gash et al., 1980) and 50 % for a forest in Puerto Rico (Schellekens et al., 1999) and are difficult to extrapolate

to other catchments due to the heterogeneity and complexity of natural systems.

    GLEAM estimates of actual evaporation show relatively high evaporation rates in winter and are never reduced to zero in summer, as opposed to modeled M5 estimates, as shown in Figure 4b. GLEAM actual evaporation minus the separately calculated interception is 94 % of potential evaporation, implying almost no water limited conditions, as opposed to our models in which actual evaporation in summer (Apr–Sep) is, due to water stress, reduced to approximatively 73 % of potential evapora-

365 tion on average for all models except VHM (Figure 4c). Larger differences between models occur in the ratio $E_A/E_P$ during winter (Oct–Mar), when FLEX-Topo, wflow_hbv and VHM show $E_A/E_P$ ratios close to unity, and dS2 the lowest values of $E_A/E_P \sim 0.75$ as shown in Figure 4c. The dS2 model differs from all other models as it relies on a year-round constant water stress coefficient ($C_{cst}$), independent of water supply, while the stress coefficient depends on root-zone soil moisture content in all other models (Table 3).

Most models slightly overestimate summer runoff coefficients with values between 0.22 and 0.26 which are very close to the observed value of 0.22, as shown in Figure 4d. During winter, runoff coefficients vary between 0.55 and 0.71, which is close

to the observed value of 0.66. This implies a relatively high level of agreement between models in reproducing the medium- to long-term partitioning of precipitation into evaporation and drainage and thus in approximating at least long-term conservation of energy (Hrachowitz and Clark, 2017).

## 5.2 Internal model states

### 5.2.1 Snow days

MODIS snow cover is detected over most of the catchment area for some time each year between November 2001 and November 2017, except for the periods of November 2006 to March 2007 and November 2007 to March 2008, when snow is detected in less than half of the catchment cells, as shown in Figure 5a. The number and magnitude of modeled snow storage events varies between models (Figure 5b). The modeled number of snow days per hydrological year varies from ∼28 days for FLEX-Topo, WALRUS and wflow_hbv to ∼62 days for GR4H and ∼90 days for NAM, M4 and M5, as shown in Figure 5c. The variability in median annual maximum snow storage varies from 3 mm for wflow_hbv and ∼5-6 mm for FLEX-Topo and WALRUS to ∼10 mm for GR4H, M4, M5 and 15 mm for NAM. We further evaluate the plausibility of these modeled snow processes by comparing modeled and observed snow cover, for days when a valid MODIS observation is available.

The presence of snow modeled by FLEX-Topo, wflow_hbv and WALRUS coincides for 92 % with the presence of snow observed by MODIS. However, these models fail to model snow for ∼62 % of days when MODIS reports the presence of snow, implying that these models miss many observed snow days, but when they predict snow, it was also observed (Figure 5d).

NAM, M4 and M5, on the other side, predict the presence of snow which coincides with observed snow by MODIS in ∼68 % of the positive predictions, implying a relatively high probability of false alarm snow prediction of ∼32 %. However, they miss only ∼29 % of actual positive snow days observed by MODIS (Figure 5d). This suggests that these models miss fewer observed snow days, but they also overpredict snow days numbers, which could be related to the use of a single temperature threshold to distinguish between snow and rain, as opposed to a temperature interval in the other models (Table 2).

GR4H is in between the two previously mentioned model categories, with a snow prediction which coincides with observed snow by MODIS in 79 % of the positive predictions and therefore only 21 % of false alarms. The model misses 42 % of actual positive snow days observed by MODIS. GR4H therefore shows a more balanced trade-off between the number of false alarms and the amount of observed snow events missed. This is illustrated in Figure 5d.

With an increased threshold to distinguish snow days in MODIS, from 10 % to 15 % of cells in the catchment with a detected snow cover (Figure 5d and Figure 5e respectively), we decrease the number of observed snow days. For all models, this leads to an increase in the ratio of false alarms over predicted snow days but also a decrease of the ratio of missed events over actual observed snow days by MODIS. However, as all models are similarly affected by the change in threshold, our findings on the differences in performance between models show little sensitivity to this threshold.

Despite the large variability in snow response between models, snow processes are represented by a degree-hour method in all models, suggesting a high sensitivity of the snow response to the snow process parametrization (Table 2).

### 5.2.2 Root-zone soil moisture

Vegetation accessible water volumes that can be stored in the root zone largely control the long-term partitioning of precipitation into evaporation and drainage. Most hydrological models include a representation of this root-zone storage capacity $S_{\mathrm{R,max}}$, which is estimated through calibration (Table 2). The calibrated root-zone storage capacities vary between 74 mm and 277 mm across studied models. The root-zone soil moisture content relative to saturation of models with relatively large root-zone storage capacities (here defined as $S_{\mathrm{R,max}} > 200$ mm) tends to never fully dry out (>0.20) and saturate (<0.94) as opposed to models with lower root-zone storage capacities ($S_{\mathrm{R,max}} < 150$ mm), in which the storage tends to either dry out completely and/or fully saturate (Figure 6a). If the vegetation accessible water storage dries out, this will lead to water stress and reduced transpiration. On the other hand, if the root-zone storage is saturated, no more water can be stored, resulting in fast drainage. The size of the root-zone storage capacity is therefore a key control of the hydrological response, allowing us to explain some of the observed variability between models. The range of SCATSAR-SWI1km Soil Water Index (SWI) varies between 0.29 and 0.82 for a value of the characteristic time length ($T$-value) of 15 days and the range reduces as the $T$-value increases (Figure 6b).

We compare the dynamics of modeled and remote sensing estimates of root-zone soil moisture by calculating Spearman rank correlations between modeled root-zone soil moisture and remote sensing estimates of the Soil Water Index for the available $T$-values of 2, 5, 15, 20, 40, 60 and 100 days. As the $T$-value increases, the Soil Water Index is more smoothed and delayed. For each model realization, we identify the $T$-value which yields the highest Spearman rank correlation between Soil Water Index and modeled root-zone soil moisture (Figure 6c). The optimal $T$-value increases with the size of the calibrated root-zone storage capacity and varies between 15 and 60 days. A small root-zone storage capacity is indeed likely to fill through precipitation and empty through evaporation and drainage more rapidly than a large water storage capacity, leading to a higher temporal relative soil moisture variability. The mismatch between the relatively high root-zone storage capacities of VHM ($S_{\mathrm{R,max}} \sim 200$ mm) in relation to the relatively low optimal $T$-values of 20 days is likely related to the unclosed water balance (Section 5.1.2). The similarity between modeled root-zone soil moisture and Soil Water Index with optimal $T$-values is high, as implied by Spearman rank correlations varying between 0.88 and 0.90 across models. However, the disparity in optimal $T$-values between models underlines the different temporal representations of root-zone soil moisture content across models, implying that all these models cannot simultaneously provide a plausible representation of the catchment-scale vegetation accessible water content.

### 5.2.3 Total storage anomalies

Total water storage anomalies obtained from GRACE are compared to the storage as simulated by the models, showing relatively similar seasonal patterns, as illustrated in Figure 7a for model M5. GRACE total storage anomalies of the Semois upstream of Membre-Pont and the Ourthe upstream of Tabreux are mainly represented by two neighboring cells (Figure 7b), allowing us to test how models represent the observed spatial variability. The range of anomalies in the Semois upstream of Membre-Pont is larger than in the Ourthe upstream of Tabreux, implying 18 %, 3 % and 7 % less summer and 19 %, 19 % and

10 % more winter storage in the Semois upstream of Membre-Pont for each of the three GRACE processing centers (Figure 7c). Median precipitation is also 37 % higher in the Semois upstream of Membre-Pont than in the Ourthe upstream of Tabreux during winter months (Oct-Mar), but relatively similar during summer months (Apr-Sep), as shown in Figure 7d. This differ-
440 ence in precipitation potentially leads to a wider range of modeled anomalies in the Semois upstream of Membre-Pont than in the Ourthe upstream of Tabreux for all models, as shown in Figure 7e. This implies that all models reproduce the spatial variability between both catchments observed by GRACE. As the models were calibrated for the Ourthe at Tabreux and parameter sets were transferred to the Semois upstream of Membre-Pont, the forcing data is the main difference to explain the modeled spatial variability.

445 The models are also able to represent the observed temporal dynamics of total storage anomalies, as suggested by Spearman rank correlation coefficients ranging between 0.62 and 0.80 for the Ourthe upstream of Tabreux (Figure 7f). There is, however, no relation between the Spearman rank correlations of the anomalies and the total modeled storage range (difference between maximum and minimum values), as shown in Figure 7f. PRESAGES, WALRUS, VHM and dS2 have the largest ranges of total modeled storage, varying between 260 and 280 mm and are also characterized by a relatively large root-zone storage capacities
450 (PRESAGES and VHM) or no separate root zone (WALRUS and dS2), while the total storage range of all other models is between 200 and 220 mm. The similarity in total storage range between most models is likely related to the identical forcing data and the similarity in the long-term partitioning of precipitation into drainage and evaporation (Section 5.1.2). However, the absolute values of total storage during a specific event or the partitioning in internal storage components may vary between models (Section 5.3).

455 ## 5.3 Interactions between storage and fluxes during dry periods

As previously seen in Figure 6a, the relative root-zone soil moisture content of the GR4H model is always above 0.2 for the three years for which SCATSAR-SWI1km data are available, as opposed to M5 which fully dries out for some time during the summers of 2015–2017. The Normalized Difference Vegetation Index of MODIS (NDVI, Didan, 2015a, b) also does not show a sharp decrease during these periods (Figure 8a,b). Actual evaporation in M5 is strongly reduced during these dry soil moisture
460 periods unlike GR4H, as shown in Figure 8c,d. When zooming into the dry period around September 2016, Figure 8e,f shows median relative root-zone soil moisture in GR4H of ∼0.24 versus ∼0.01 for M5, while SCATSAR-SWI1km has a higher median value of ∼0.55 (for both optimal $T$-values of 20 and 40 days). The dryness of root-zone soil moisture in M5 leads to median daily evaporation of 0.8 mm d$^{-1}$ against 1.3 mm d$^{-1}$ for GR4H and prolonged periods of almost zero evaporation in M5 (e.g. 31/08–03/09, 09/09–15/09 and 22/09–30/09), while this neither occurs in GR4H nor in GLEAM actual evaporation, as
465 shown in Figure 8g,h. Despite the high streamflow performance of model M5 (Figure 3, Table 2), it is unlikely that transpiration is reduced to almost zero for several days in a row each summer in a catchment where approximately half of the area is covered by forests. This is also not supported by the remote-sensing data of soil moisture, NDVI and evaporation. High streamflow performances, therefore, do not warrant the plausibility of internal process representation. Despite the dried-out root-zone storage in M5, there is still water available in the slow storage to sustain a baseflow close to observed values, as shown in
470 Figure 8j,l. The streamflow responses of GR4H and M5 are both close to observations (Figure 8i,j) in spite of differences

in storage and evaporation, suggesting different internal process representations for a similar aggregated streamflow response during a low flow period.

## 5.4 Plausibility of process representations

The models are ranked and evaluated for a selection of criteria using observed streamflow, remote sensing data and expert knowledge (Figure 9). All models deviate less than $\pm 6\%$ from observed median annual streamflow (Figure 9a), which is less than the estimated uncertainty of 11 % (Section 3.3). In contrast, the modeled flashiness and baseflow indices of most models deviate more than the estimated uncertainty (Figure 9b,c). FLEX-Topo is the only model with a clear overestimation of the flashiness index, which relates to the calibration aim of having a flashy model to reproduce small summer peaks (de Boer-Euser et al., 2017).

Modeled median annual total actual evaporation deviates by approximately -10 % from GLEAM estimates, except for the +22 % overestimation of the VHM model due to the issue of the unclosed water balance, as shown in Figure 9d. These results are consistent with the evaluation study of GLEAM compared to other evaporation products (Miralles et al., 2016) which reports higher than average values for GLEAM in Europe ($\sim$+10 % at our latitude).

Four models explicitly account for interception with a separate module. Median annual interception rates deviate substantially from GLEAM estimates (-47 % to +80 %) as shown in Figure 9e. There is a high uncertainty in the partitioning of evaporation into different components in evaporation products and GLEAM likely underestimates interception rates (Miralles et al., 2016; Zhong et al., 2020). Therefore, we consider a large uncertainty of +50 % to evaluate and rank the models. The GR4H interception is lower than GLEAM estimates. However, an interception storage was recently included in an hourly GR model (GR5H), to better represent the interception processes (Ficchì et al., 2019; Thirel et al., 2020).

All models substantially underestimate the number of days when snow is observed by MODIS at the catchment scale for all valid MODIS observations (cloud cover < 40 % and excluding summer months), as shown in Figure 9f. Yet, we estimate a low uncertainty of less than 2 % around this number (Section 3.3). The NAM, M4 and M5 models are closest to MODIS estimates, but they are characterized by high false alarm rates (Figure 5d), which implies a mismatch in the modeled and observed days with snow for valid MODIS observations. Based on expert knowledge (Royal Meteorological Institute Belgium, 2015) and the trade-off between recall and precision (Figure 5d,e), we expect the annual number of days with snow storage to be between 28 and 62 days $yr^{-1}$ as modeled by wflow_hbv, WALRUS, FLEX-Topo and GR4H, whereas the $\sim$90 days $yr^{-1}$ of NAM, M4 and M5 seems too high.

The FLEX-Topo and M2 to M5 models are characterized by an empty root-zone storage for approximately 10 days $yr^{-1}$ ($\overline{S}_R$ < 1 %) as shown in Figure 9g. These models have in common that evaporation from the root-zone occurs at potential rate and is not (or hardly) reduced when soils are becoming dry until the point where the storage is empty. This is the case for models with very low or absence of the evaporation reduction parameter $L_P$. This behavior is not supported by the remote sensing data of evaporation, soil moisture and NDVI (Section 5.3), nor by theory on root water uptake reduction under dry conditions (Feddes et al., 1978). The additional slow groundwater reservoir added in model M5 compared to M2-M4 leads to a smaller root-zone storage capacity as the available storage is partitioned into the root-zone storage and the additional

505 groundwater store. The smaller root-zone storage capacity of model M5 exacerbates the number of annual days with empty storage. This highlights the complex interactions in internal dynamics even in parsimonious lumped models with similar mean annual streamflow performance.

Catchments with relatively large root-zone storage capacities underestimate GRACE estimates of very dry storage anomalies most (Figures 6 and 9h). The uncertainty of GRACE is represented by the estimates of the three processing centers and the

510 ∼18 mm uncertainty estimate mentioned in Section 3.3. FLEX-Topo has a low root-zone storage capacity and is the only model which overestimates the very dry storage anomalies. Models with root-zone storage capacities of around 110 mm to 150 mm show the most consistent behavior with GRACE estimates of very dry storage anomalies.

## 6 Discussion

### 6.1 Implications

While streamflow alone may be used to evaluate hydrological models, we subsequently use these models to understand internal states and fluxes in current and future conditions (Alcamo et al., 2003; Hagemann et al., 2013; Beck et al., 2017) or to make operational streamflow predictions (e.g. HBV and GR types of models are used by the Dutch and French forecasting services). Our findings show that similar streamflow responses obtained by models calibrated according to an identical protocol rely on different internal process representations. In other words, we might get the right answers but for the wrong reasons (Kirchner,

2006), as these models cannot at the same time all be right and different from each other (Beven, 2006).

Almost all models show a similar long-term partitioning of precipitation into drainage and evaporation, as they are forced and constrained by the same data, also leading to relatively similar volumes of total storage. However, the partitioning of total storage in several internal storage components differs between models, resulting in distinct runoff responses as expressed by the baseflow and flashiness indices.

None of the models is systematically consistent with the information available from streamflow observations, remote sensing data and expert knowledge. However, some processes either play a limited role on the overall water balance or can be compensated by other processes. Snow occurs every year but is not a major component of the streamflow regime (de Wit et al., 2007), interception evaporation can be compensated by root-zone evaporation, and very dry periods only occur for several weeks per year when streamflow is already very low. There is also a large uncertainty in each of the data sources, which makes

530 us reluctant to use them to determine hard thresholds to reject models. Instead, we ranked the models for a selection of "soft" criteria and found that NAM, wflow_hbv and PRESAGES are overall most consistent with the evaluation data, with median ranks of 2-3. While an overall ranking may be useful for practitioners, modelers benefit more from the specific ranking for each criteria to detect specific model deficiencies that could be improved in the model structure. An overall ranking is only a mere indication, which should be interpreted carefully due to uncertainty in the evaluation data and the applied calibration strategy.

Higher model performance does not seem to be related to model complexity, but rather to the presence of specific components and to the calibration strategy chosen by each contributing institution.

The presence of interception or a slow storage (absent in M2-M4 but added in M5) affects the representation of other internal processes, including transpiration and/or root-zone soil moisture, implying that individual internal model components are altered by the presence/absence of other potentially compensating processes. Adding an additional internal model component changes the internal representation of water storage and fluxes through the system, which should be kept in mind if model parameters were to be fixed in alternative model structures. Furthermore, model improvements through additional process components and/or adapted parametrization should not only be evaluated in terms of the aggregated response, but also in the partitioning of fluxes and storages through the system (e.g. does the groundwater component improve the baseflow index at the expense of the availability of root-zone soil moisture during dry periods?). Models should be confronted with expert knowledge, e.g. on the occurrence of days with water stress or snow storage, to assess the plausibility of internal states and fluxes (Gharari et al., 2014; Hrachowitz et al., 2014; van Emmerik et al., 2015).

Applying these models to a future, more extreme climate in the same region might lead to contrasting insights regarding impacts of climate change, as also shown by studies of Hagemann et al. (2013), Melsen et al. (2018) and de Niel et al. (2019) in which model structures may lead to different signs of change of mean streamflow. Using one model or the other to assess the effect of rising temperatures on snow could lead to very different time scales of snow storage decline. Vegetation already experiences more intense water stress in some models compared to others and this would be exacerbated in more extreme drought scenarios (Melsen and Guse, 2019). More intense precipitation events could affect interception evaporation and therefore water availability in the root-zone differently from one model to another. Beyond model structure, the experience each modeler has with its model and associated calibration procedure to constrain model parameters may also impact the simulation results (Melsen et al., 2019).

Our findings should, therefore, encourage modelers to use multiple data sources for model calibration and evaluation, as already suggested by many other studies (Samaniego et al., 2010; Rakovec et al., 2016a; Koch et al., 2018; Stisen et al., 2018; Nijzink et al., 2018; Veldkamp et al., 2018; Dembélé et al., 2020). Remote sensing estimates of soil moisture, evaporation and total storage anomalies are available at the global scale and in spite of potential biases with models, the temporal dynamics are useful to constrain our models (McCabe et al., 2017; Sheffield et al., 2018). Additionally, it seems essential to support decision-makers by studies relying on multi-model and multi-parameter systems, as also suggested by Haddeland et al. (2011) and Schewe et al. (2014), to reveal uncertainties inherent to the heterogeneous hydrological world (Beven, 2006; Savenije, 2010; Samaniego et al., 2010; Hrachowitz and Clark, 2017).

This study is the result of a joint research effort of scientists and practitioners gathering each year in Liège at the International Meuse Symposium to exchange interdisciplinary and intersectoral knowledge related to the Meuse basin. Although coordination of large international teams may be challenging, international studies favor a close collaboration between scientists and practitioners that can learn from each other to accelerate modeling advances (Archfield et al., 2015). Another advantage of comparing modeling results of several research groups is to quickly detect small mistakes in the modeling process, including shifts in the time series or using forcing data of one catchment to model another catchment. While hydrograph characteristics were the main focus of the previous study (de Boer-Euser et al., 2017), we gain distinct insights on the plausibility of model behavior by evaluating additional facets of internal process representation using remote sensing data.

## 6.2 Knowledge gaps and limitations

Many aspects of the hydrological response remain unknown and can hardly be evaluated against observations. While in-situ observations of snow, evaporation or soil moisture are rarely available at sufficient spatio-temporal scale, remote sensing estimates have the advantage of high spatial resolution, though they often rely on models themselves and are affected by high and often unknown uncertainty. Comparing models with these independent observations is valuable to evaluate their consistency and detect outliers. However, these observations cannot be considered as representative of the truth as they rely on many assumptions themselves, hindering "real" hypotheses testing. The ratio of actual over potential evaporation as a result of water stress at the catchment-scale, therefore, remains highly uncertain (Coenders-Gerrits et al., 2014; Mianabadi et al., 2019). While areal fractions of snow cover can be estimated by MODIS, the presence of clouds limits the usability of the data and knowledge of catchment-scale snow water equivalent is lacking. If remote sensing estimates of near-surface relative soil moisture are available, root-zone water content remains uncertain and while GRACE provides estimates of total storage anomalies, we lack knowledge on absolute total water storage. The spatial variability and the temporal dynamics of these remote sensing products provide useful, additional, independent information to understand the hydrological puzzle, but certainly not all the answers to evaluate the states typically included in process-based models. Measurements are, therefore, of crucial importance to increase our understanding of hydrological processes at the catchment-scale, which in turn will improve the quality of remote sensing products and model development (Vidon, 2015; Burt, T. P., McDonnell, 2015; van Emmerik et al., 2018).

The evaluation of model behavior is conditional on the calibration procedure, which was freely chosen by the individual contributing institutes. The use of different or more calibration objectives and in-depth uncertainty estimation (Beven and Binley, 1992) may have resulted in different conclusions in terms of the plausibility of the behavior of each model.

We performed a thorough analysis of twelve models, five variables and three catchments. We deliberately chose to limit the number of study catchments to balance depth with breadth, allowing us to dive into process-relevant insights.

## 7 Conclusions

Similar streamflow performance of process-based models, calibrated following an identical protocol, relies on different internal process representations. Most models are relatively similar in terms of the long-term partitioning of precipitation into drainage and evaporation. However, the partitioning between transpiration and interception, snow processes and the representation of root-zone soil moisture varies significantly between models, suggesting variability of water storage and release through the catchment. The comparison of modeled states and fluxes with remote sensing estimates of evaporation, root-zone soil moisture and vegetation indices suggests that models with relatively small root-zone storage capacities and without reduction in root water uptake during dry conditions lead to unrealistic drying-out of the root-zone storage and significant reduction of evaporative fluxes each summer. Expert knowledge in combination with remote sensing data further allows us to "softly" evaluate the plausibility of model behavior by ranking them for a set of criteria. Even if none of the models is systematically consistent with the available data, we did not formally reject specific models due to the uncertainty in the evaluation data

and their changing relevance for the studied catchments. The dissimilarity in internal process representations between models implies that they are not necessarily providing the right answers for the right reasons, as they cannot simultaneously be close to reality and different from each other. While the consequences for streamflow may be limited for the historical data, the differences may exacerbate for more extreme conditions or climate change scenarios. Considering the uncertainty of process representation behind the scenes of streamflow performance and our lack of knowledge and observations on these internal

processes, we invite modelers to evaluate their models using multiple variables, we encourage more experimental research, and highlight the value of multi-model multi-parameter studies to support decision making.

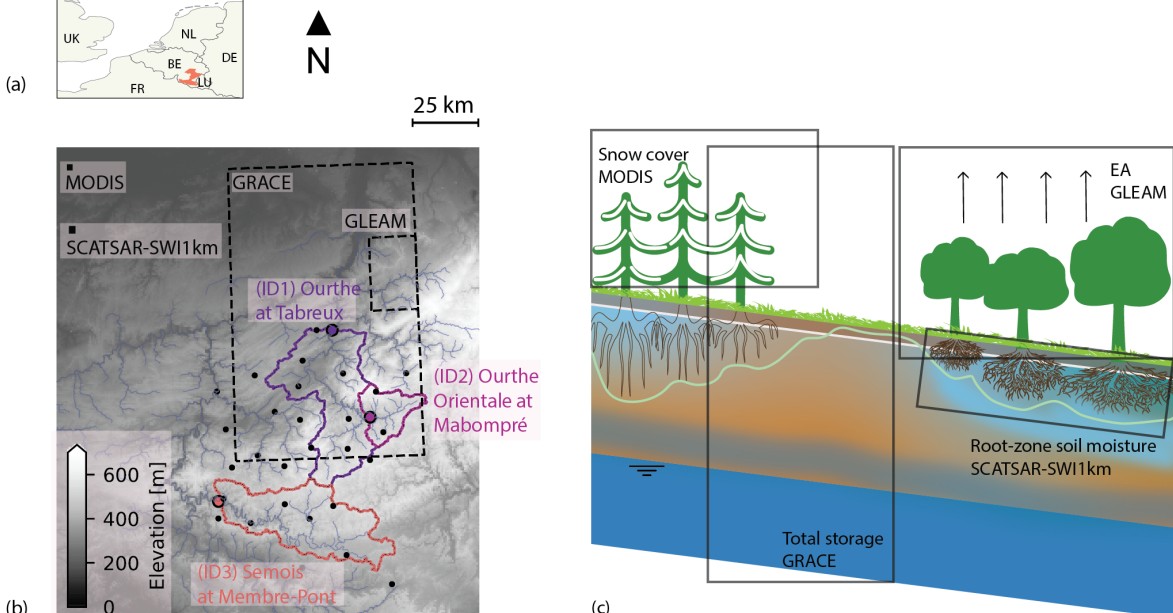

**Figure 1. (a)** Location of the study catchments in Belgium, North-West Europe. **(b)** Digital elevation model and catchments of the Ourthe upstream of Tabreux (ID1), Ourthe Orientale upstream of Mabompré (ID2) and Semois upstream of Membre-Pont (ID3). Pixel size of GRACE, GLEAM, MODIS and SCATSAR-SWI1km. Colored dots are the streamflow gauging locations and black dots are the precipitation stations. **(c)** Perceptual overview of the link between studied fluxes and states and remote sensing products.

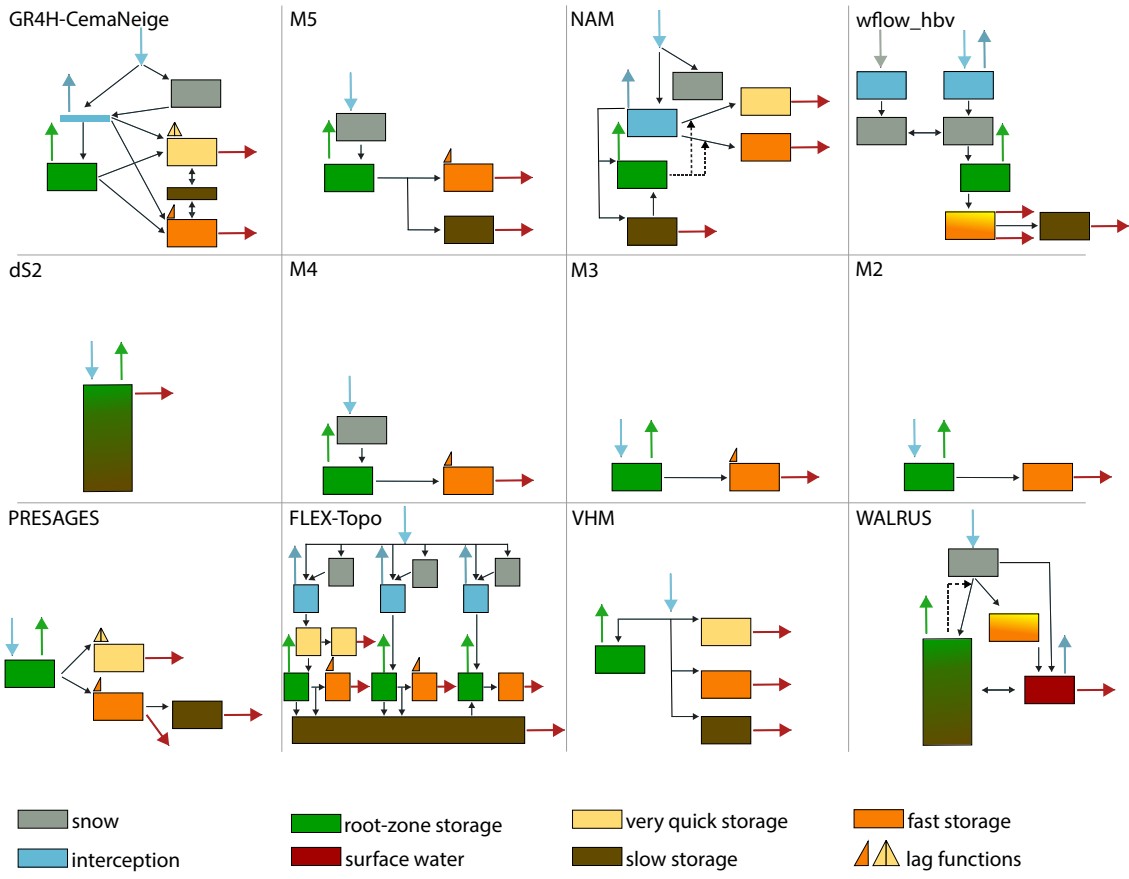

**Figure 2.** Simplified schematic overview of 12 model structures (adapted from de Boer-Euser et al., 2017) with the aim to highlight similarities and differences between the models. Solid arrows indicate fluxes between stores, while dashed arrows indicate the influence of a state to a flux. Colored arrows indicate incoming or outgoing fluxes, whereas black arrows indicate internal fluxes. The narrow blue rectangle in GR4H indicates the presence of an interception module without interception storage capacity (Table 3). Storages with a color gradient indicate the combination of several components in one reservoir.

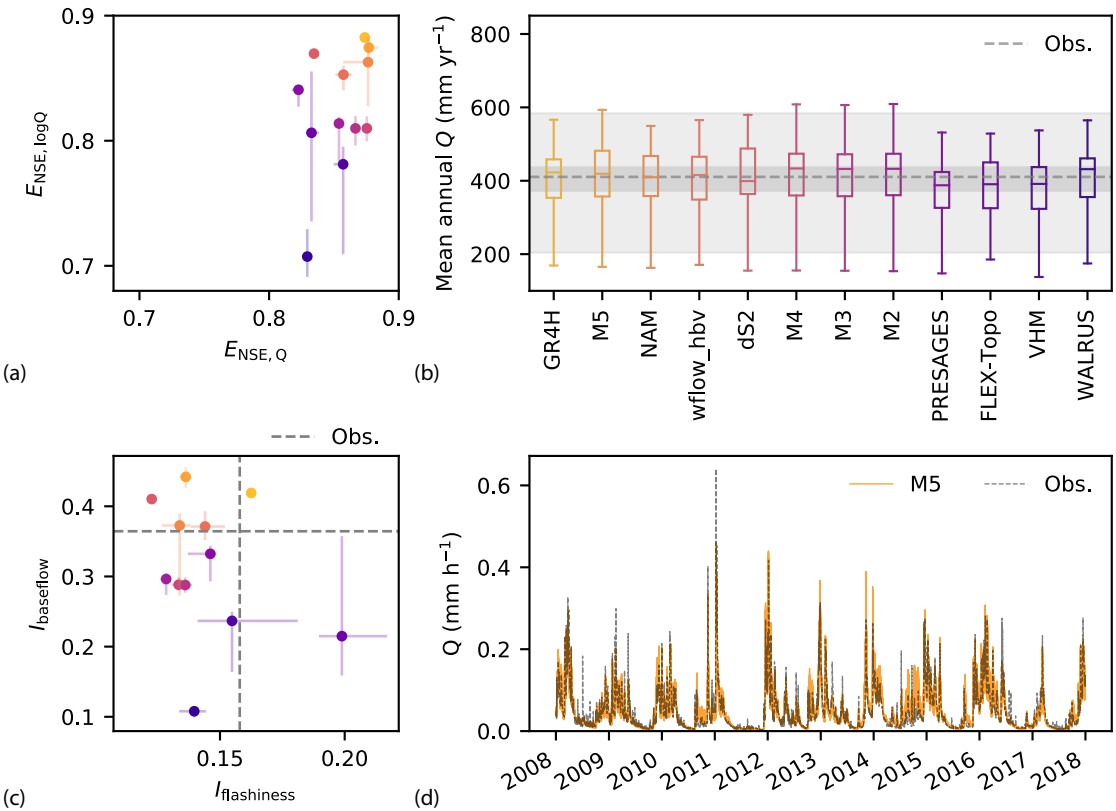

**Figure 3.** Evaluation of modeled streamflow performance for the Ourthe at Tabreux for the period 2008–2017. **(a)** Nash-Sutcliffe Efficiencies of the streamflow $E_{\mathrm{NS,Q}}$ and the logarithm of the streamflow $E_{\mathrm{NS,logQ}}$ (median, 25/75[th] percentiles across parameter sets). **(b)** Modeled mean annual streamflow for hydrological years between 2008–2017 across feasible parameter sets. The models are ranked from the highest to the lowest performance according to the Euclidean distance of streamflow performance (see Table 2). The dashed line and grey shaded areas show median, 25/75[th] and minimum-maximum range of observed mean annual streamflow. **(c)** Baseflow index $I_{\mathrm{baseflow}}$ as a function of the flashiness index $I_{\mathrm{flashiness}}$ (median, 25/75[th] percentiles across parameter sets). Observed values are shown by the grey dashed lines. **(d)** Observed and modeled hydrographs of model M5 with high streamflow model performance (low Euclidean distance).

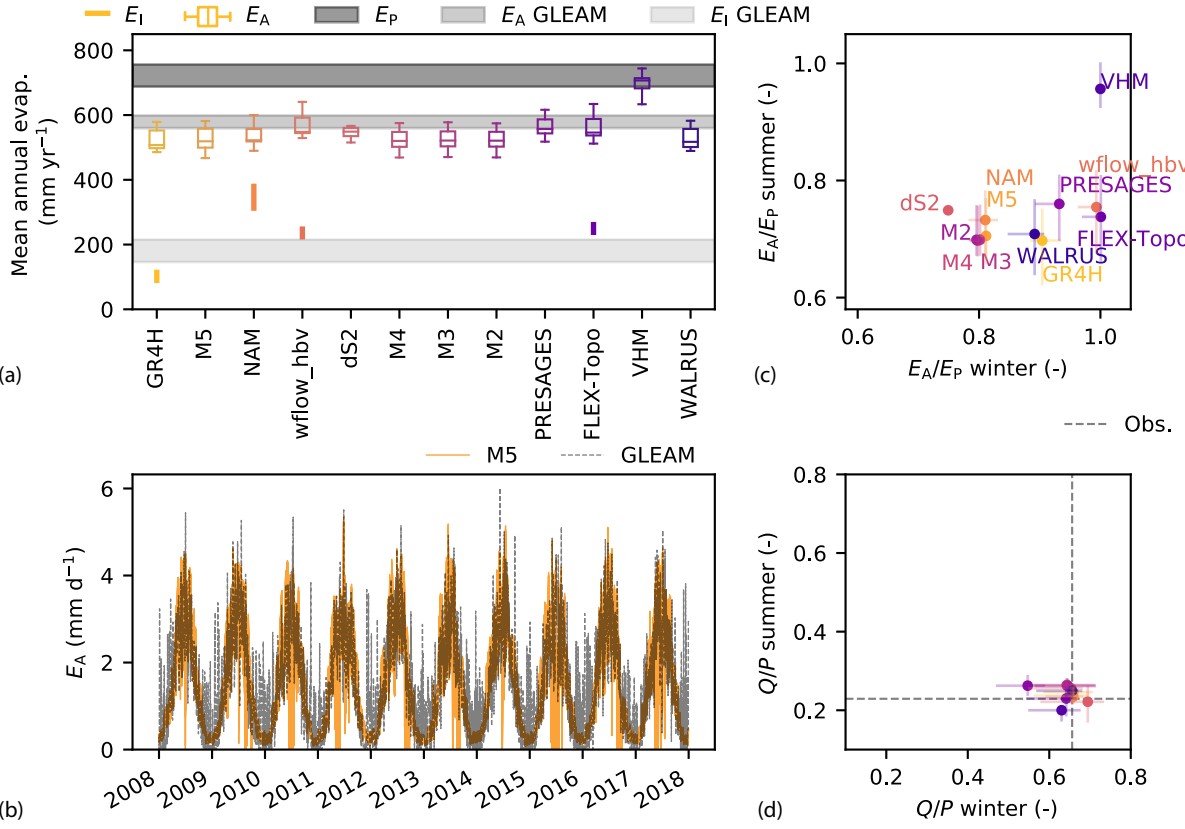

**Figure 4.** Evaluation of modeled evaporation for the Ourthe upstream of Tabreux for the period 2008–2017. **(a)** Modeled mean annual actual evaporation $E_A$ and interception evaporation $E_I$ for hydrological years between 2008–2017 across feasible parameter sets. The dark grey shaded area shows the range of potential evaporation $E_P$ used as input for the models. The light grey shaded area shows GLEAM actual and interception evaporation. **(b)** Daily actual evaporation from GLEAM and modeled by the M5 model. **(c)** Summer against winter $E_A/E_P$ ratios for each model (median and 25/75$^{th}$ percentiles across parameter sets). **(d)** Summer against winter runoff coefficient $Q/P$ for each model (median and 25/75$^{th}$ percentiles across parameter sets), plotted on the same scale. The dashed grey lines indicate the observed median runoff coefficients in summer and winter.

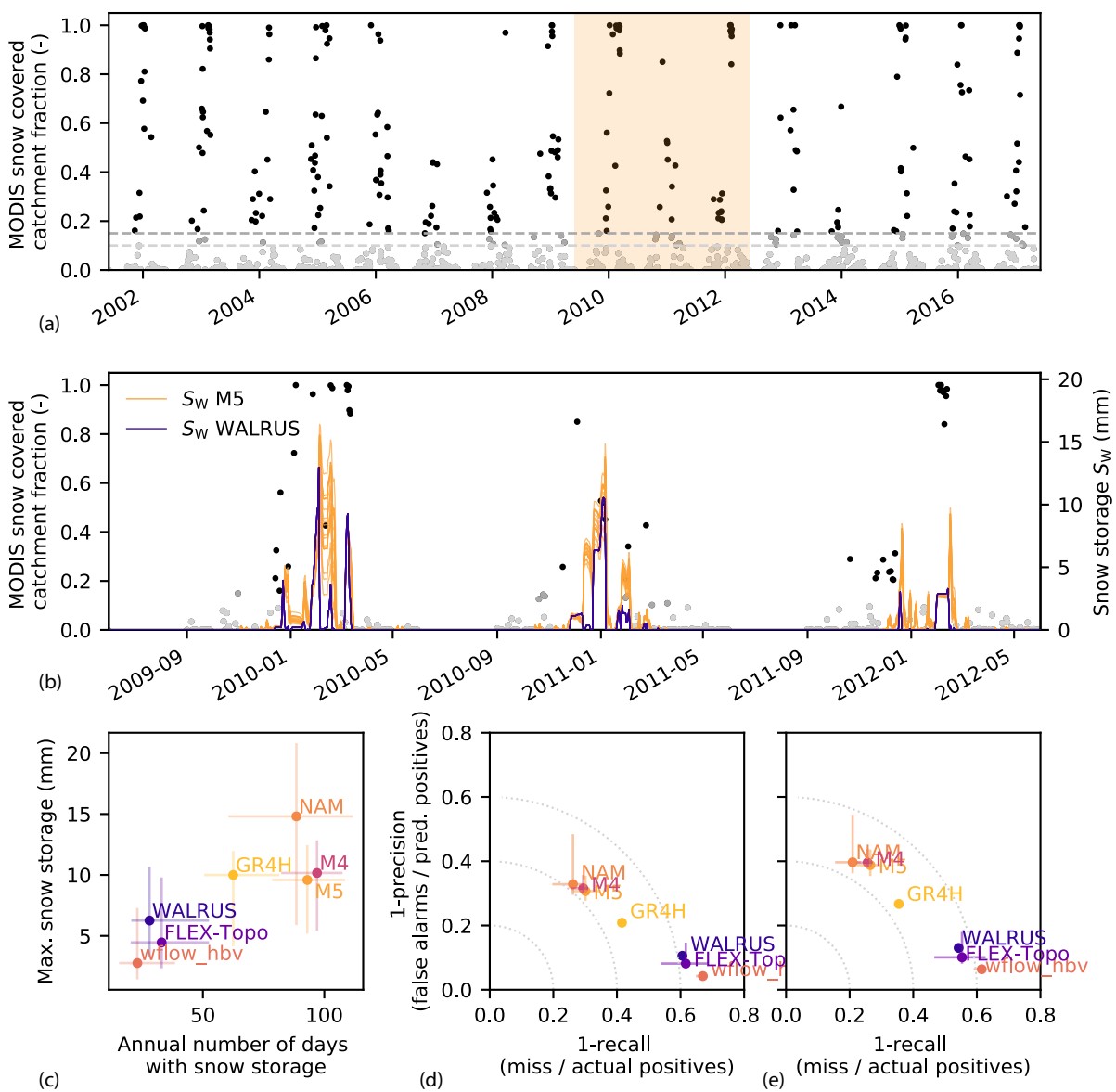

**Figure 5. (a)** Fraction of cells with a MODIS areal fraction snow cover greater than zero in the Ourthe Orientale upstream of Mabompré for the period 2001–2017. MODIS data are available once every three days on average. The dashed lines show the two thresholds of 10 % and 15 % selected to distinguish snow days. **(b)** Modeled snow storage for two contrasting models M5 and WALRUS for the light orange shaded period. **(c)** Median annual maximum snow storage as a function of number of days per year with snow. Light (yellowish) colors indicate models with higher performance (lower Euclidean distances). The vertical and horizontal error bars indicate the 25/75$^{\text{th}}$ percentiles over time and feasible parameter sets **(d,e)** Two-dimensional representation of the false alarm over predicted positives ratio (1-precision) as a function of miss over actual positives ratio (1-recall) when applying a threshold of **(d)** 10 % and **(e)** 15 % of cells within the catchment with snow cover greater than zero. In this representation, the perfect model would be at the origin (100 % hits and 0 % false alarms).The dotted lines show the distance from the origin. The vertical and horizontal error bars indicate the uncertainty within feasible parameter sets.

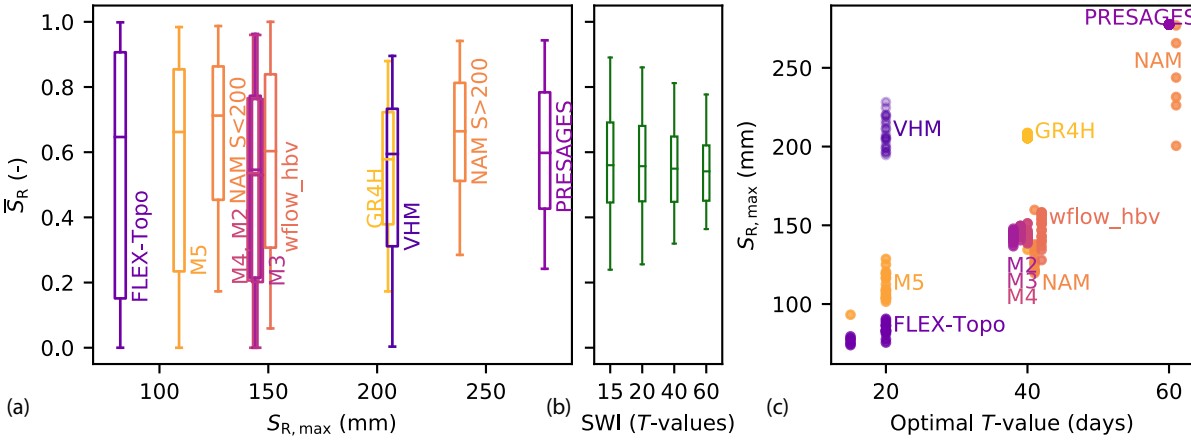

**Figure 6. (a)** Range of relative root-zone soil moisture $\overline{S}_R$ in the Ourthe upstream of Tabreux for the period 2015–2017 as a function of the median root-zone storage capacity ($S_{R,max}$) across parameter sets. The feasible parameters for NAM are split in two groups due to the large variability of $S_{R,max}$ (subsets with $S_{R,max}$ of ~130 mm and ~240 mm). **(b)** Range of the SCATSAR-SWI1km Soil Water Index for several values of the characteristic time length $T$ (days) for the period when SCATSAR-SWI1km is available (2015–2017). **(c)** Root-zone storage capacity $S_{R,max}$ as a function of the optimal $T$-value for each model realization. Optimal $T$-values are derived at the highest Spearman rank correlation between Soil Water Index and modeled root-zone soil moisture.

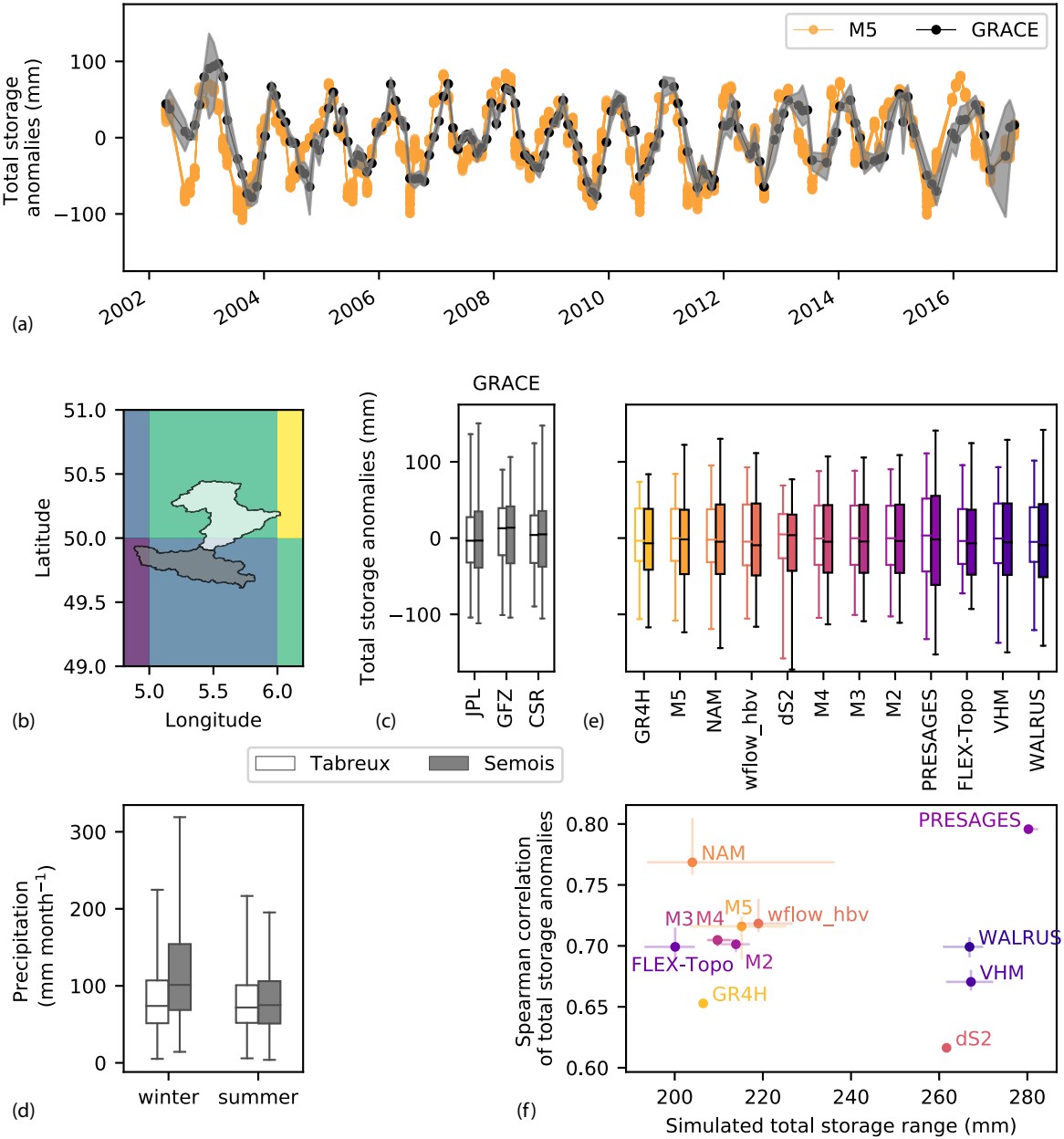

**Figure 7. (a)** Total storage anomalies modeled by M5 and compared to GRACE for the Ourthe upstream of Tabreux. The grey band shows the variability in total storage anomalies of the three processing centers. **(b)** Catchments of the Ourthe upstream of Tabreux (light grey) and Semois upstream of Membre-Pont (dark grey) located in two neighboring GRACE cells. **(c)** Range of GRACE total storage anomalies for the three processing centers for the Semois upstream of Membre-Pont compared to the Ourthe upstream of Tabreux for the period 2001–2017. **(d)** Mean monthly precipitation during winter and summer months in the Semois upstream of Membre-Pont compared to the Ourthe upstream of Tabreux. **(e)** Modeled total storage anomalies for both catchments. **(f)** Spearman rank correlations between GRACE and modeled total storage anomalies as a function of the range of modeled total storage for the Ourthe upstream of Tabreux.

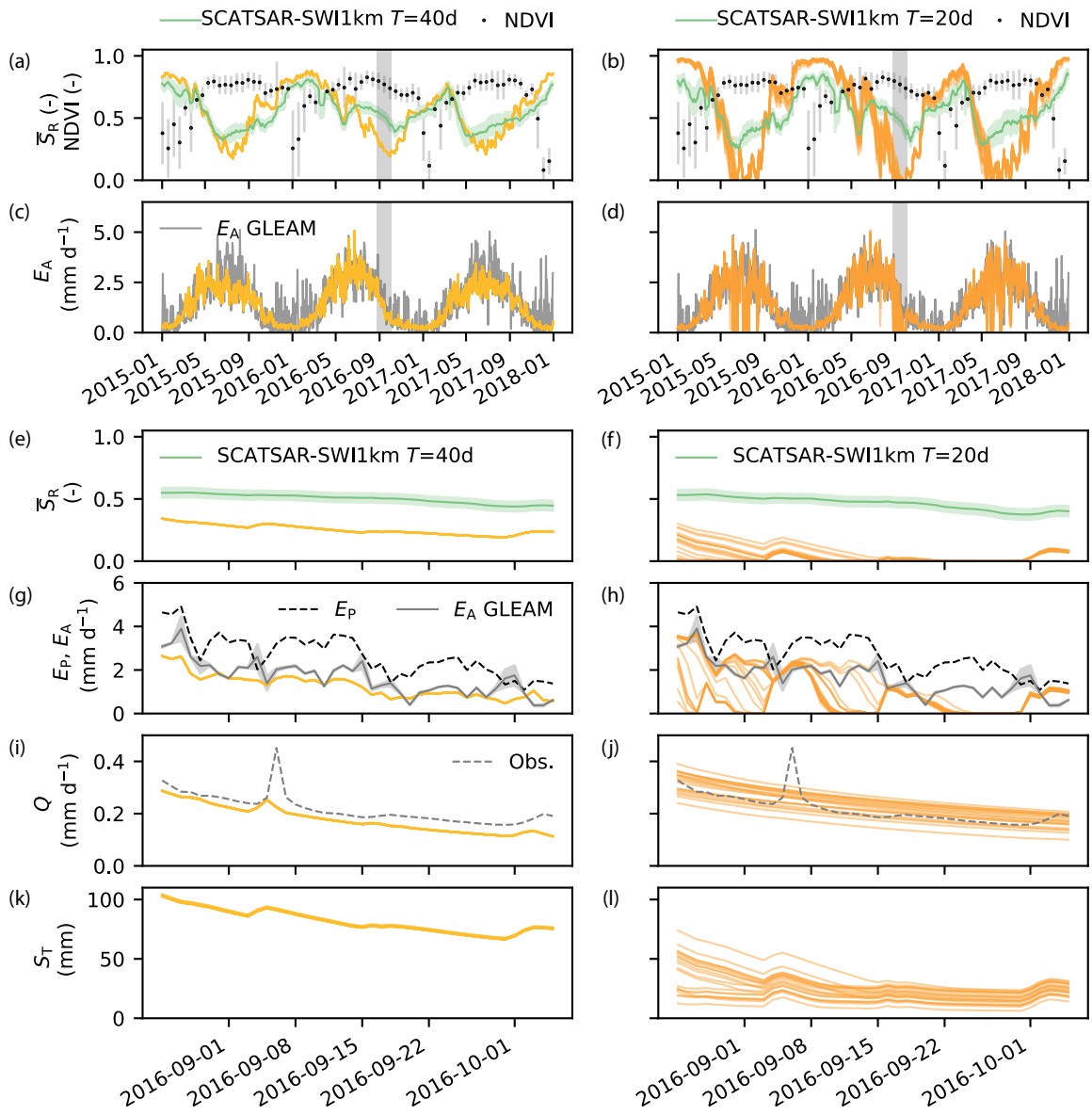

**Figure 8. (a,b)** Modeled relative root-zone soil moisture $\overline{S}_R$, SCATSAR-SWI1km Soil Water Index with optimal $T$-value and NDVI for the period 2015–2017 for GR4H (yellow) and M5 (orange) respectively. The error bars and bands show the standard deviation of the remote sensing data within the catchment area **(c,d)** Actual evaporation $E_A$ by GR4H and M5 for the period 2015–2017, showing a large reduction of evaporation during summer for M5 unlike GR4H and GLEAM actual evaporation **(e,f)** Zoomed-in modeled $\overline{S}_R$ and SCATSAR-SWI1km root-zone soil moisture for the grey shaded period of September 2016 in (a,b,c,d). **(g,h)** Potential, modeled and GLEAM actual evaporation, **(i,j)** Modeled and observed streamflow $Q$, **(k,l)** Total storage $S_T$ for the September 2016 dry period. The narrow uncertainty band of the GR4H model is related to its converging parameter search method.

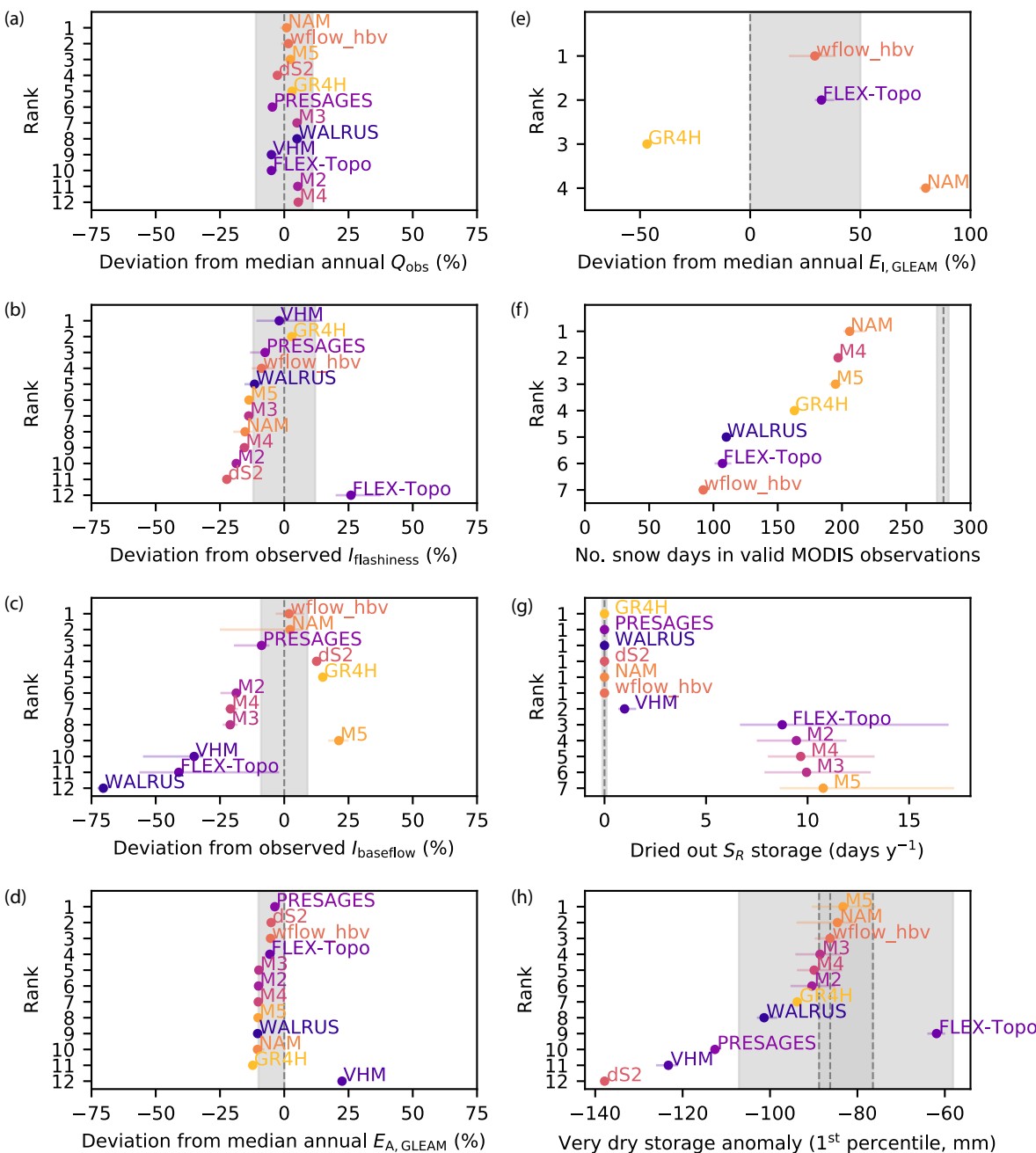

**Figure 9.** Ranking and evaluation of model behavior for a selection of criteria based on observed streamflow, remote sensing data and expert knowledge. The grey shaded areas are soft indications of more plausible behavior based on uncertainty estimates and expert knowledge. Model ranks as a function of the: **(a)** deviation from observed median annual streamflow. **(b)** deviation from the flashiness index. **(c)** deviation from the baseflow index. **(d)** deviation from median annual GLEAM actual evaporation. **(e)** deviation from median annual GLEAM interception for models with explicit separate interception module. **(f)** number of days with snow cover for valid MODIS observations between 2001-2017, for models with a snow module. **(g)** annual number of days when the root-zone storage is dry (filled with less than 1% of its capacity). **(h)** deviation from the $1^{st}$ percentile of GRACE total storage anomalies for the three centers. The error bars show the 25-75$^{th}$ range across the realizations from the ensemble of parameter sets retained as feasible.

**Table 1.** Description of symbols used for fluxes, storages and parameters in Tables 2 and 3

| Symbol | unit | Description |
|---|---|---|
| *Fluxes* | | |
| $E_P$ | mm h$^{-1}$ | Potential evaporation |
| $E_I$ | mm h$^{-1}$ | Interception evaporation |
| $E_R$ | mm h$^{-1}$ | Transpiration and soil evaporation |
| $E_W$ | mm h$^{-1}$ | Sublimation |
| $E_A$ | mm h$^{-1}$ | Total actual evaporation (sum of soil evaporation, transpiration, (separate) interception and, if applicable, sublimation) |
| $P$ | mm h$^{-1}$ | Precipitation |
| $P_R$ | mm h$^{-1}$ | Precipitation entering the root-zone storage (after snow and/or interception if present or fraction/total precipitation) |
| $Q$ | mm h$^{-1}$ | Streamflow |
| $Q_R$ | mm h$^{-1}$ | Flux from root-zone to fast and/or slow runoff storage |
| $Q_P$ | mm h$^{-1}$ | Percolation flux from root-zone storage to slow runoff storage |
| $Q_C$ | mm h$^{-1}$ | Capillary flux from slow runoff storage to root-zone storage |
| $Q_G$ | mm h$^{-1}$ | Seepage (up/down) / extraction |
| *Storages* | | |
| $S_T$ | mm | Total storage |
| $S_W$ | mm | Snow storage |
| $S_I$ | mm | Interception storage |
| $S_R$ | mm | Root-zone storage |
| $\overline{S}_R$ | - | Relative root-zone storage ($S_R/S_{R,max}$) |
| $S_D$ | mm | Storage deficit |
| $S_{VQ}$ | mm | Very quick runoff storage |
| $S_F$ | mm | Fast runoff storage |
| $S_S$ | mm | Slow runoff storage |
| $S_{SW}$ | mm | Surface water storage |
| *Parameters* | | |
| $C_E$ | - | Correction factor for $E_P$ |
| $I_{max}$ | mm | Maximum interception capacity |
| $S_{R,max}$ | mm | Maximum root-zone storage capacity |
| $S_{thresh}$ | mm | Threshold of root-zone storage above which $E_R = E_P$ |
| $L_P$ | - | Threshold of relative root-zone storage above which $E_R = E_P$ |
| $C_{cst}$ | - | Constant water stress coefficient to estimate $E_R$ |
| $a, b, S_0$ | - | Parameters describing the shape of the streamflow sensitivity |
| $a_S$ | - | Fraction of land surface covered by surface water |
| $a_G$ | - | Fraction of land surface not covered by surface water |

**Table 2.** Number of calibrated model parameters, spatial distribution, and model performance calculated for the period 2008–2017 with the Euclidean distance where a value of 0 would indicate a perfect model. Main characteristics describing snow storage, root-zone storage and total storage per model. Notations are defined in Table 1.

| | GR4H | M5 | NAM | wflow_hbv | dS2 | M4 | M3 | M2 | PRESAGES | FLEX-Topo | VHM | WALRUS |
|---|---|---|---|---|---|---|---|---|---|---|---|---|
| Number of calibrated parameters | 4 | 9 | 12 | 9 | 4 | 7 | 6 | 5 | 6 | 20 | 12 | 3 |
| Lumped (L) / Semi-distributed (S) / Distributed (D) | L | L | L | D | L | L | L | L | L | S | L | L |
| Euclidean distance $\sqrt{(1-E_{\mathrm{NS,Q}})^2 + (1-E_{\mathrm{NS,logQ}})^2}$ | 0.17 | 0.18 | 0.18 | 0.20 | 0.21 | 0.23 | 0.23 | 0.24 | 0.24 | 0.26 | 0.26 | 0.34 |
| **Snow storage $S_{\mathrm{W}}$ (compared to MODIS snow cover)** | | | | | | | | | | | | |
| Snow module | ✓ | ✓ | ✓ | ✓ | - | ✓ | - | - | - | ✓ | - | ✓ |
| Degree-hour method | ✓ | ✓ | ✓ | ✓ | - | ✓ | - | - | - | ✓ | - | ✓ |
| Elevation zones | ✓ | - | - | ✓ | - | - | - | - | - | ✓ | - | - |
| Temperature interval for rainfall and snow | ✓ | - | - | ✓ | - | - | - | - | - | ✓ | - | ✓ |
| Melt factor constant in time | - | ✓ | - | ✓ | - | ✓ | - | - | - | ✓ | - | ✓ |
| Melt factor $\sim$ snow storage | ✓ | - | ✓ | - | - | - | - | - | - | - | - | - |
| Refreezing of liquid water | - | - | ✓ | ✓ | - | - | - | - | - | - | - | - |
| Sublimation | - | - | - | - | - | - | - | - | - | ✓ | - | - |
| Calibration snow parameters | - | ✓ | ✓ | - | - | ✓ | - | - | - | ✓ | - | - |
| **Root-zone storage $S_{\mathrm{R}}$ (compared to SCATSAR-SWI1km Soil Water Index)** | | | | | | | | | | | | |
| Separate root-zone module with capacity $S_{\mathrm{R,max}}$ | ✓ | ✓ | ✓ | ✓ | - | ✓ | ✓ | ✓ | ✓ | ✓ | ✓ | - |
| $\frac{\mathrm{d}S_{\mathrm{R}}}{\mathrm{d}t} = P_{\mathrm{R}} - E_{\mathrm{R}}$ | - | - | - | - | - | - | - | - | - | - | ✓ | - |
| $\frac{\mathrm{d}S_{\mathrm{R}}}{\mathrm{d}t} = P_{\mathrm{R}} - E_{\mathrm{R}} + Q_{\mathrm{C}}$ | - | - | ✓ | - | - | - | - | - | - | - | - | - |
| $\frac{\mathrm{d}S_{\mathrm{R}}}{\mathrm{d}t} = P_{\mathrm{R}} - E_{\mathrm{R}} - Q_{\mathrm{R}}$ | ✓ | ✓ | - | ✓ | - | ✓ | ✓ | ✓ | ✓ | - | - | - |
| $\frac{\mathrm{d}S_{\mathrm{R}}}{\mathrm{d}t} = P_{\mathrm{R}} - E_{\mathrm{R}} - Q_{\mathrm{R}} - Q_{\mathrm{P}} + Q_{\mathrm{C}}$ | - | - | - | - | - | - | - | - | - | ✓ | - | - |
| **Total storage $S_{\mathrm{T}}$ (anomalies are compared to GRACE total storage anomalies)** | | | | | | | | | | | | |
| $S_{\mathrm{T}} = -S_{\mathrm{D}} \cdot a_{\mathrm{G}} + S_{\mathrm{F}} \cdot a_{\mathrm{G}} + S_{\mathrm{SW}} \cdot a_{\mathrm{S}}$ | - | - | - | - | - | - | - | - | - | - | - | ✓ |
| $S_{\mathrm{T}}(Q) = \frac{1}{a} \frac{1}{1-b} \cdot Q^{1-b} + S_0$ | - | - | - | - | ✓ | - | - | - | - | - | - | - |
| $S_{\mathrm{T}} = S_{\mathrm{R}} + S_{\mathrm{F}}$ | - | - | - | - | - | - | ✓ | ✓ | - | - | - | - |
| $S_{\mathrm{T}} = S_{\mathrm{W}} + S_{\mathrm{R}} + S_{\mathrm{F}}$ | - | - | - | - | - | ✓ | - | - | - | - | - | - |
| $S_{\mathrm{T}} = S_{\mathrm{W}} + S_{\mathrm{R}} + S_{\mathrm{F}} + S_{\mathrm{S}}$ | - | ✓ | - | - | - | - | - | - | - | - | - | - |
| $S_{\mathrm{T}} = S_{\mathrm{R}} + S_{\mathrm{VQ}} + S_{\mathrm{F}} + S_{\mathrm{S}}$ | - | - | - | - | - | - | - | - | - | ✓ | ✓ | - |
| $S_{\mathrm{T}} = S_{\mathrm{W}} + S_{\mathrm{R}} + S_{\mathrm{VQ}} + S_{\mathrm{F}} + S_{\mathrm{S}}$ | ✓ | - | - | - | - | - | - | - | - | - | - | - |
| $S_{\mathrm{T}} = S_{\mathrm{I}} + S_{\mathrm{W}} + S_{\mathrm{R}} + S_{\mathrm{F}} + S_{\mathrm{S}}$ | - | - | - | ✓ | - | - | - | - | - | - | - | - |
| $S_{\mathrm{T}} = S_{\mathrm{I}} + S_{\mathrm{W}} + S_{\mathrm{R}} + S_{\mathrm{VQ}} + S_{\mathrm{F}} + S_{\mathrm{S}}$ | - | - | ✓ | - | - | - | - | - | - | ✓ | - | - |

**Table 3.** Main characteristics describing evaporation processes per model (with $\checkmark^1$ indicates $L_P = 1$ and $\checkmark^2$ indicates $E_I = 0$). Notations are defined in Table 1.

| | GR4H | M5 | NAM | wflow_hbv | dS2 | M4 | M3 | M2 | PRESAGES | FLEX-Topo | VHM | WALRUS |
|---|---|---|---|---|---|---|---|---|---|---|---|---|
| Correction factor for potential evaporation | - | $\checkmark$ | - | - | - | $\checkmark$ | $\checkmark$ | $\checkmark$ | - | - | - | - |
| Interception evaporation $E_I$ | $\checkmark$ | - | $\checkmark$ | $\checkmark$ | - | - | - | - | - | $\checkmark$ | - | - |
| Maximum interception storage $I_{max}$ | - | - | $\checkmark$ | $\checkmark$ | - | - | - | - | - | $\checkmark$ | - | - |
| $I_{max} \sim 1.1 - 3.4$ mm | - | - | - | - | - | - | - | - | - | $\checkmark$ | - | - |
| $I_{max} \sim 1.4 - 2.9$ mm | - | - | - | $\checkmark$ | - | - | - | - | - | - | - | - |
| $I_{max} \sim 5.3 - 6.9$ mm | - | - | $\checkmark$ | - | - | - | - | - | - | - | - | - |
| $E_I = \begin{cases} E_P, & \text{if } S_I > 0. \\ 0, & \text{otherwise.} \end{cases}$ | - | - | $\checkmark$ | $\checkmark$ | - | - | - | - | - | $\checkmark$ | - | - |
| $E_I = \begin{cases} E_P, & \text{if } P > E_P. \\ P, & \text{otherwise.} \end{cases}$ | $\checkmark$ | - | - | - | - | - | - | - | - | - | - | - |
| Transpiration and soil evaporation $E_R$ | $\checkmark$ | $\checkmark$ | $\checkmark$ | $\checkmark$ | $\checkmark$ | $\checkmark$ | $\checkmark$ | $\checkmark$ | $\checkmark$ | $\checkmark$ | $\checkmark$ | $\checkmark$ |
| $E_R = E_P \cdot C_{cst}$ | - | - | - | - | $\checkmark$ | - | - | - | - | - | - | - |
| $E_R = E_P \cdot \frac{\overline{S}_R \cdot (2 - \overline{S}_R)}{1 + E_P / S_{R,max} \cdot (2 - \overline{S}_R)}$ | $\checkmark$ | - | - | - | - | - | - | - | $\checkmark$ | - | - | - |
| $E_R = E_P \cdot C_E \cdot \frac{\overline{S}_R \cdot (1 + m_1)}{\overline{S}_R + m_1}$, with $m_1 = 10^{-2}$ | - | $\checkmark$ | - | - | - | $\checkmark$ | $\checkmark$ | $\checkmark$ | - | - | - | - |
| $E_R = \begin{cases} (E_P - E_I) \cdot \frac{\overline{S}_R}{L_P}, & \text{if } \overline{S}_R < L_P. \\ E_P - E_I, & \text{otherwise.} \end{cases}$ | - | - | $\checkmark^1$ | $\checkmark$ | - | - | - | - | - | $\checkmark$ | $\checkmark^2$ | - |
| $E_R = E_P \cdot f(S_d)$ | - | - | - | - | - | - | - | - | - | - | - | $\checkmark$ |
| Total actual evaporation $E_A$ | $\checkmark$ | $\checkmark$ | $\checkmark$ | $\checkmark$ | $\checkmark$ | $\checkmark$ | $\checkmark$ | $\checkmark$ | $\checkmark$ | $\checkmark$ | $\checkmark$ | $\checkmark$ |
| $E_A = E_R$ | - | $\checkmark$ | - | - | $\checkmark$ | $\checkmark$ | $\checkmark$ | $\checkmark$ | $\checkmark$ | - | $\checkmark$ | $\checkmark$ |
| $E_A = E_R + E_I$ | $\checkmark$ | - | $\checkmark$ | $\checkmark$ | - | - | - | - | - | - | - | - |
| $E_A = E_R + E_I + E_W$ | - | - | - | - | - | - | - | - | - | $\checkmark$ | - | - |

*Data availability.* Streamflow and precipitation data were provided by the Service Public de Wallonie in Belgium (Direction générale opéra-tionnelle de la Mobilité et des Voies hydrauliques, Département des Etudes et de l'Appui à la Gestion, Direction de la Gestion hydrologique intégrée (Bld du Nord 8-5000 Namur, Belgium)). Hourly radiation data were retrieved from the portal of the Royal Netherlands Meteoro-logical Institute (2018; http://www.knmi.nl/nederland-nu/klimatologie/uurgegevens). Daily temperature data were retrieved from the E-OBS OPeNDAP server http://opendap.knmi.nl/knmi/thredds/dodsC/e-obs_0.25regular/tg_0.25deg_reg_v17.0.nc (Haylock et al., 2008). Actual evaporation estimates from the Global Land Evaporation Amsterdam Model (GLEAM) are available through the SFTP server of GLEAM, (https://www.gleam.eu/, Miralles et al., 2011; Martens et al., 2017). MODIS Snow cover fractions (Hall and Riggs, 2016a, b) are available for download from the Earthdata portal https://earthdata.nasa.gov/. The Soil Water Index SCATSAR-SWI1km are available from the Copernicus Global Land Service at https://land.copernicus.eu/global/products/swi (Bauer-Marschallinger et al., 2018). GRACE land data (Swenson and Wahr, 2006; Swenson, 2012) are available at http://grace.jpl.nasa.gov, supported by the NASA MEaSUREs Program. The modeled states and fluxes of each model will be made available online in the 4TU data repository.

*Author contributions.* LJEB, GT, FF, MH, LAM, AHW designed the study. GT, TdBE, JB, CCB, JdN, SM, BG, FF, JN and LJEB ran their models and provided simluated time series of all variables. LJEB conducted all the analyses and wrote the manuscript. All authors discussed the design, results and contributed to the final manuscript.

*Competing interests.* The authors declare that they have no competing interests.

*Acknowledgements.* We thank Deltares and Rijkswaterstaat for the financial support to conduct this analysis. The authors would like to thank the Service Public de Wallonie, Direction générale opérationnelle de la Mobilité et des Voies hydrauliques, Département des Etudes et de l'Appui à la Gestion, Direction de la Gestion hydrologique intégrée (Bld du Nord 8-5000 Namur, Belgium) for providing the precipitation and streamflow data. We thank Keith Beven, Shervan Gharari and two anonymous referees for their constructive comments which helped us improve the manuscript.

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
