# Peer review of "Behind the scenes of streamflow model performance"

_Hydrology and Earth System Sciences, 2020_

## Referee Comment (RC1) · Keith Beven (Referee) · 30 Apr 2020

This paper takes a diverse collection of hydrological models, previously calibrated to the Oerthe basin, and subjects them to comparison with estimates of evapotranspiration, soil moisture, snow cover and GRACE total estimates. The models all produce "reasonable" streamflow calibrations (I assume, since it seems that none of them have been rejected in the first calibration part of the study). The conclusion is that they do so in different ways, and still none of them are rejected.

Now I understand why it is diplomatic when working within an international project to be kind to all the groups who are participating, but doing so does not produce an outcome that is in any way useful. The models are just shown to be different. Why are these

models not being tested as hypotheses about how the catchment system is working? Indeed, we could rather say on the basis of the evidence presented that none of them are really fit for purpose when the additional variables are taken into account.

Except that it is not quite that simple, because ALL of the additional variables used in this comparison are subject to significant uncertainty and commensurability issues. And without taking some account of those uncertainties no real testing is possible (it is also worth noting that no account is taken of uncertainty in the original calibration exercise either – why not in 2020? It has been recognized as an issue in model calibration for more than 30 years!). The section on knowledge gaps at the end should be moved to before the model comparison is presented, and should explicitly consider the uncertainty and commensurability issues. Nowhere is there any mention of the uncertainties arising in verification studies of these additional variables, but that is surely significant.

To give a particular example: models with and without interception storage. This is an example of why more thought is required about what is actually being compared here. One of the reasons why models choose NOT to have an interception store is to reduce the number of parameters required to be calibrated or estimated a priori. But how this works will also depend on how potential evapotranspiration is estimated. Does it include the effects of a wet canopy – especially over rough canopies. This can be really important (and subject to significant uncertainties in effective roughness and humidity deficits because of sensitivities under such conditions). Here the Hargreaves PE formula does not explicitly consider wet canopy conditions, but GLEAMS, with which model outputs are being compared, does). So in what sense (or degree of uncertainty) are these comparable?

Similarly for the soil moisture comparison. The satellite derived estimates really only deal with near-surface moisture (with a depth that varies with wetness) and that in itself is associated with uncertainty, especially near saturation. There is some discussion here about the issue of comparing relative moisture content in the root zone when the different models parameterize that in different ways and a rather odd correlation

analysis with the T parameter – can you not think about how (and if) that data can be used as a hypothesis test. There are clearly similar issues with GRACE and snow cover data (e.g. is fact that some models do not predict snow storage on a day important if snow covers are small)

So rather than have a "so what?" outcome to this paper, I would suggest instead that it should be reformulated into a hypothesis testing framework (EAWAG might be able to make suggestions about how this should be done). This is a real opportunity to frame the issue in this way. Not that because of the uncertainties and commensurability issues that does not imply that any or all of the models will be rejected. That will partly depend on what assumptions and expert knowledge are made in the analysis (– see L450, except that no expert knowledge has really been used in the study as presented). Effectively what you have here are some indices of dynamic behavior with which to evaluate the models – the expert knowledge needs to come in as to how (or IF) those indices (with all the issues with them) can be used to test the models in any way rigorously. This would require very major revisions to the analysis but would make the whole project so much more worthwhile in advancing the modelling process.

Some particular points

L35 There are other variables that have been used (and much earlier than the studies cited) – eg. saturated contributing areas (Beven and Kirkby, HSB 1979; Güntner et al., HP 1999; Blazkova et al., HP 2002) and patterns of water tables in many piezometers (Seibert et al., HP 1997; Lamb et al. AWR 1998; Blazkova et al. WRR 2002).

L228 drop infinitely – this is misleading. Theoretically yes, but it is directly related to baseflow outflow by water balance and you do not expect baseflow to go to zero in droughts for these catchments. It can also have the advantage of reducing number of parameters required.

L288 How can GLEAM potential ET be less than the annual estimate cited in L271 (and how undertain are those estimates)

L397 ecosystems have adapted – but these ecosystems have not surely. In this area they have been affected by forestry and agricultural practices for thousands of years.

Keith Beven

---

## Short Comment (SC1) · 12 May 2020

**A short comment on Bouaziz et al., 2020 and a question from Prof. Beven:**

I think the work has a lot of potentials and there is room for improvement which motivated me to write this short comment.

1- Following the comments by Prof. Beven, I would like to ask the authors a more direct question: "is there any practical use in exploiting the remote sense data in constraining the hydrological models at the scale of interest?"
   In some applications, bucket-style models are constrained based on evaporation products. I understand that the evaporation products can be used for practical purposes and possibly as a preliminary benchmark, however, my concerns are: (1) the reduced uncertainty coming from confronting the model with another set of products might result in an "illusion of certainty" in simulation and patterns. As an example, refer to Wang et al., 2015a to see the possible uncertainty in the transpiration/evaporation products from a model. (2) the whole modeling purpose is to predict unknown and come up with the temporal and spatial prediction of some states and fluxes. We then set up a model, say it does or doesn't get the spatial pattern, train it with the result of another model, and then it gets the spatial pattern probably right. What is the end goal of this practice? We can probably join efforts with the developers of the already existing products to improve their products rather than just being a user. Or use a machine-learning algorithm to capture what the patterns in the products are. My question in short; do we learn? Or do we produce a similar product (hopefully a better one)?

2- GRACE is rather coarse for the basins of interest. It is suggested that the GRACE data should be used for catchment above 150,000 square kilometers (Rodell et al., 2011). This might be counterintuitive; visualization of GRACE over a large area will show that the data is more diffused that its actual resolution. Also, GRACE is very uncertain in itself, using a mean value of its three or more variations may result in deliberate killing of uncertainty (Scanlon et al., 2018).

3- Checking the consistency of input data is essential before starting the modeling phase. The knowledge gap Section, in my point of view, can be moved earlier in the manuscript and can be populated by quantified evaluation of the available data sets for forcing/calibrating the models. Basically, from the data sets, you have all the components that you need to close the water balance.
   $e = \text{sum}(P-E-Q)\Delta t - \Delta S$
   Can you get $e$ close to zero over a month or a year? (similar experiment to Wang et al., 2015b)
   Do you have a sense of uncertainty or disagreement between the precipitation and rain gauges?
   As low-hanging fruit, it is possible to have an understanding of approximate interception/transpiration for this region. Any flux towers? Study sites from Luxembourg might be helpful? It seems the product you used in this study for evaporation underestimates interception significantly. From their website it seems interception is set to ~10% globally (if I interpreted it correctly). Do you perhaps know this ratio for the region of study from this data (model outcome)? This seems to contradict some earlier paper by co-authors on the global uncertainty of the interception/transpiration. Soil evaporation from the used product may include many assumption or simplification

similar to land models (Bohn and Vivoni, 2016). Another low-hanging fruit! Can we perhaps estimate the recession coefficient from the hydrograph and compare it with calibrated values in the models to see actually which model structure allows for a more accurate estimation of recession coefficient when calibrated and why [as doctor-father always says]. For example, land models are not suited for this recession inference (Gharari et al., 2019).

4- The area is mostly agriculture, is there any regulation on the stream than may affect your inference. Referring to section 5.3, the area is mostly agriculture, to my understanding, the Sumax/root zone storage co-evolution is hypothesized for forests (that has a life of more than a year). Agricultural lands do not follow any of that logic, it does what farmers do (there might be some correlation). Maybe you can argue around the rain-fed nature of agriculture in this region but still, crops have a lifespan of a season. Land models can see the variation of leaf area index (LAI) and with some modification even variation of root zones over period of time. That can be a better testbed for exploring root zoon hypothesis than bucket-style models.

5- "The T -value has previously been positively correlated with root-zone storage capacity, assuming a high temporal variability of root-zone soil moisture and therefore a low T - value for small root-zone storage capacities $S_{R,max}$ (Bouaziz et al., 2020)". Possible that I totally get it wrong, but if I understand correctly, Bouaziz et al., 2020 used a hydrological model in combination with satellite observation. Is this a model result that is used for intercomparing rather than the satellite observation itself? Maybe separate the data (products) into groups of "directly observed" and "inferred based on a model".

6- Upscaling of snow cover to basin level is a tricky business. Snow storage, snowpack extent may not be uniform over an area (Cherkauer et al., 2003). Also, the snowpack can persist with temperature much higher than the phase-change temperature identified in the model. Snowpack may also stays longer under canopy. The phase-change temperature can have a range, for example, for the VIC model this is a transitional span of temperature (for example from -2°C to 2°C) that can be tuned for the same reason (snow precipitation). I would suggest checking the snow extent versus the temperature first. This might give insight into whether or not any model can simulate the observed snow extent given the temperature. Also, snow under canopy may stay longer, does the product you use capture that?

7- As a modeler that might be interested to model the basins of interest, what is the take-home message for me. I assume one of the aims of an inter-comparison project is the knowledge mobilization of already known facts about basin(s) to the wider community. This can be done better in this manuscript I would say. Perhaps, identifying the target audience. Is the manuscript targeted for catchment hydrology? Or Large-scale hydrology? The current manuscript does not server any.
I would say, as coordination of the large team takes a lot of efforts and work, maybe give a new dimension to your paper by elaborating the organizational efforts put into this study (why did you initiate this inter-comparison, why the current list of models and authors, what made you to choose them? what effect it might have on real-world application, etc).

8- I would suggest the authors clarify their research equations in the beginning and come back to the research questions in the conclusions. In the current version, there are no tangible research questions. For example, "Haddeland et al. (2011) and Schewe et al.

(2014) compared global hydrological models and found that differences between models are a major source of uncertainty." I think this is what you can reflect/elaborate on in your conclusions (hopefully quantitively)?

One collusion from this study can be for example, "a two-bucket model with snow component is sufficient enough to get the dynamic of the data we selected". Can this be one of your conclusions?

Some studies from the land modeling community can be helpful in this regard. For example, Bets et al., 2015 provided a structure for the comparison (including evaluation, comparison, benchmarking, fit for purpose, utilizing the available data, etc). Following this structure or similar structures can hopefully clarify the manuscript more. One benchmarking strategy could be ensemble simulation of all models within their prespecified parameter ranges. This can be the basis for comparison when the model is calibrated on the streamflow and subsequently on other data sets such as evaporation in a stepwise fashion. This seems to be not a lot of work as the models are already set up. Moreover, the land model studies can provide more insight into large scale modeling and their related issues. For example, Crow et al. (2003) is a classic example. In my point of view far ahead of its time and not very well received [the same work nowadays would probably have 20 authors with the same citation level in a single year and will be magically highlighted!]. Another great example to show uncertainty in large scale models in reproducing mean and variability (Koster, R.D. and P. Mahanama 2012) with a very simple model. Another example is Hurkmans et al. 2008. These studies and similar works may provide a better understanding of the exploitation of additional data in large scale modeling and associated uncertainties.

9- Concerning FLEX-Topo. It seems to be the only semi-distributed model among all the other models. Have you properly constrained the component of this model (or do you have enough expert knowledge to do so)? It would be good to highlight the advantage of the semi- distributed model here if any. The control over the different components of FLEX-Topo becomes increasingly hard if the code is written separately for each landscape (different structures). I tried to have a similar code for each landscape and recreate the desired structure just by adjusting the parameters. That provides better control over the performance of each landscape. For example, did you check the transpiration of each landscape? Sometimes it is the case that soil moisture from one landscape is empty and the other landscapes are evaporating at the maximum rate.

10- I didn't know that FLEX-Topo got a sublimation component. How that is implemented? Is sublimation a major process in the region of study? I would not say so. Sublimation is also a tricky process; a magical one! it can account for uncertainty in snowpack similar to the transpiration for soil moisture. There is also a refreezing formulation for one of the models. Interested to know how that happens in a model that may not close the energy balance. It would be good to include all the model formulation in the Appendix if not too much work.

11- The figures presenting the results are very hard to follow. I am not sure if I understand most of them. I would suggest simplifying them.

12- A question from Prof. Beven and maybe the authors; is that possible to even reject a model in large scale modeling? From my experience and due to the issue of scale (and observation at that scale), most of the models can be accepted. For example, in a recent modeling effort that we have done (Gharari et al., 2020), the VIC model with the only

micropore and with only macropore water movement yields the same result when calibrated (exploring the inclusion of macropore water movement in land models; aligned with Beven and Germann 1982, to Beven 2018). How should I justify macropore versus micropore at that scale for a colleague whose entire career is focused on how to properly/mathematically represent micropore water movement? What is the path forward? I appreciate your thoughts on that.

I am confident this manuscript will be an interesting one. Hope that my comments are helpful.

Finally, please note that any reference to my work was just for explanation, discussion and clarification. I don't quest for citations.

With kind regards,
Shervan Gharari

**Reference:**

Beven, K. and Germann, P., 1982. Macropores and water flow in soils. *Water resources research*, *18*(5), pp.1311-1325.
.
.
.
Beven, K., 2018. A Century of Denial: Preferential and Nonequilibrium Water Flow in Soils, 1864-1984. *Vadose Zone Journal*, *17*(1).

Bouaziz, L. J., Steele-Dunne, S. C., Schellekens, J., Weerts, A. H., Stam, J., Sprokkereef, E., Winsemius, H. H., Savenije, H. H., and Hrachowitz, M.: Improved understanding of the link between catchment-scale vegetation accessible storage and satellite-derived Soil Water Index, Water Resources Research, https://doi.org/10.1029/2019WR026365, 2020.

Best, M.J., Abramowitz, G., Johnson, H.R., Pitman, A.J., Balsamo, G., Boone, A., Cuntz, M., Decharme, B., Dirmeyer, P.A., Dong, J. and Ek, M., 2015. The plumbing of land surface models: benchmarking model performance. *Journal of Hydrometeorology*, *16*(3), pp.1425-1442.

Bohn, T. J., and E. R. Vivoni, 2016: Process-based characterization of evapotranspiration sources in the North American monsoon region, *Water Resour. Res.*, **52**(1), 358-384, doi:10.1002/2015WR017934.

Wang, S., Huang, J., Yang, D., Pavlic, G., and Li, J., 2015. Long-term water budget imbalances and error sources for cold region drainage basins. *Hydrological processes*, *29*(9), pp.2125-2136.

Wang, S., Pan, M., Mu, Q., Shi, X., Mao, J., Brümmer, C., Jassal, R.S., Krishnan, P., Li, J. and Black, T.A., 2015. Comparing evapotranspiration from eddy covariance measurements, water budgets, remote sensing, and land surface models over Canada. *Journal of Hydrometeorology*, *16*(4), pp.1540-1560.

Crow, W.T., Wood, E.F., and Pan, M., 2003. Multiobjective calibration of land surface model evapotranspiration predictions using streamflow observations and spaceborne surface radiometric temperature retrievals. *Journal of Geophysical Research: Atmospheres*, *108*(D23).

Scanlon, B.R., Zhang, Z., Save, H., Sun, A.Y., Schmied, H.M., Van Beek, L.P., Wiese, D.N., Wada, Y., Long, D., Reedy, R.C. and Longuevergne, L., 2018. Global models underestimate large decadal declining and rising water storage trends relative to GRACE satellite data. *Proceedings of the National Academy of Sciences*, *115*(6), pp.E1080-E1089.

Rodell, M., McWilliams, E.B., Famiglietti, J.S., Beaudoing, H.K., and Nigro, J., 2011. Estimating evapotranspiration using an observation based terrestrial water budget. *Hydrological Processes*, *25*(26), pp.4082-4092.

Haddeland, I., Clark, D. B., Franssen, W., Ludwig, F., Voß, F., Arnell, N. W., Bertrand, N., Best, M., Folwell, S., Gerten, D., Gomes, S., Gosling, S. N., Hagemann, S., Hanasaki, N., Harding, R., Heinke, J., Kabat, P., Koirala, S., Oki, T., Polcher, J., Stacke, T., Viterbo, P., Wee- don, G. P., and Yeh, P.: Multimodel estimate of the global terrestrial water balance: Setup and first results, Journal of Hydrometeorology, 12, 869–884, https://doi.org/10.1175/2011JHM1324.1, 2011.

Schewe, J., Heinke, J., Gerten, D., Haddeland, I., Arnell, N. W., Clark, D. B., Dankers, R., Eisner, S., Fekete, B. M., Colón-González, F. J., Gosling, S. N., Kim, H., Liu, X., Masaki, Y., Portmann, F. T., Satoh, Y., Stacke, T., Tang, Q., Wada, Y., Wisser, D., Albrecht, T., Frieler, K., Piontek, F., Warszawski, L., and Kabat, P.: Multimodel assessment of water scarcity under climate change, Proceedings of the National Academy of Sciences of the United States of America, 111, 3245–3250, https://doi.org/10.1073/pnas.1222460110, 2014.

Koster, R.D., and P. Mahanama, S.P., 2012. Land surface controls on hydroclimatic means and variability. J*ournal of Hydrometeorology* , *13*(5), pp.1604-1620.

Hurkmans, R.T.W.L., De Moel, H., Aerts, J.C.J.H. and Troch, P.A., 2008. Water balance versus land surface model in the simulation of Rhine river discharges. *Water resources research*, *44*(1).

Gharari, S., Clark, M., Mizukami, N., Wong, J.S., Pietroniro, A., and Wheater, H., 2019. Improving the representation of subsurface water movement in land models. *Journal of Hydrometeorology*, (2019).

Gharari, S., Clark, M. P., Mizukami, N., Knoben, W. J. M., Wong, J. S., and Pietroniro, A.: Flexible vector-based spatial configurations in land models, Hydrol. Earth Syst. Sci. Discuss., https://doi.org/10.5194/hess-2020-111, in review, 2020.

Cherkauer, K.A., Bowling, L.C. and Lettenmaier, D.P., 2003. Variable infiltration capacity cold land process model updates. *Global and Planetary Change*, *38*(1-2), pp.151-159.

---

## Referee Comment (RC2) · Anonymous Referee #2 · 1 Jun 2020

Bouaziz et al. (2020) evaluate 12 hydrologic models for three medium-sized Belgian catchments, all established and calibrated by eight research groups. Although the spatially aggregated streamflow performance differences among models are negligible, the internal model states and processes (can) differ significantly. This paper is an interesting diagnostic study, with nice figures. I have some minor comments which the authors should consider to address.

First of all, it is nice to see the huge collaborative efforts across many institutes behind this model inter-comparison study. This study with many details shows large differences among 12 hydrologic models and even larger differences against different remotely sensed products. Something what a reader would expect. I encourage the authors to stress more clearly, what is the main "take-home" message of the main

paper. Because the authors did not use an ensemble of model structures from a modular framework (e.g., FLEX, FUSE), which could properly address those differences or individual model deficiencies step by step (by identifying individual hypothesis), in their study they cannot clearly separate and identify, which hidden hydrological processes can help improve model functioning against those reanalysis products. Could you please comment on this?

Further details in chronologic order:

Line 80,140+: evaporation => "evapotranspiration"? Please don't forget about the plants! Hargreaves-Samani formula is for evapotranspiration, not for evaporation only.

Line 85: streamflow => "runoff", because of the unit

Line 86: You should start this sentence that this is a headwater basin of ID1

Line 101: I guess the authors could have used a bit more advanced method for interpolation rain gauge observation instead of the Thiessen polygons, to better account for input error uncertainty, e.g. kriging or its variants. The uncertainty in the meteorological inputs is not mentioned in the manuscript.

Line 114: PET method is based on Priestley Taylor, which is different from section 3.1. How is it compatible with section 3.1 and overall results?

Line 143: how did you spatially average soil moisture?

Line 153: I guess, your entire study domain is just a single GRACE pixel. I am quite skeptical for using it at all, as it's beyond the limits of its usability. The original raw GRACE signal is based on a much larger region (3degrees). You may better wait for the GRACE-FO, which has much finer native resolution...

Section 4.1 I guess all models were applied in spatially lumped manner, i.e. no spatially distributed mode, isn't it? Please write down explicitly in this section.

Line: 179-181 This analysis was done here, or in previous study? Not clear, please

specify, and provide link to the transferability results. Curious to see them.

How many behavioral parameter sets per model were used? Is the number same per hydrological model? Here referring to error bars in Fig. 3, Fig. 4 and elsewhere.

Figure 9: Is it possible to rank the models according to their performance? Which one seems to be most relevant and how it compares to e.g. an operational model, if that's available? Please think about putting some implications to the paper.
* * *

---

## Author Comment (AC1) · 2 Jun 2020

We thank Prof. Keith Beven for his detailed and constructive review. In the following, we express our view on the raised points, and illustrate our plan on how to improve the paper.

Comment:
*This paper takes a diverse collection of hydrological models, previously calibrated to the Oerthe basin, and subjects them to comparison with estimates of evapotranspiration, soil moisture, snow cover and GRACE total estimates. The models all produce "reasonable" streamflow calibrations (I assume, since it seems that none of them have*

[Figure]

*been rejected in the first calibration part of the study). The conclusion is that they do so in different ways, and still none of them are rejected. Now I understand why it is diplomatic when working within an international project to be kind to all the groups who are participating, but doing so does not produce an outcome that is in any way useful. The models are just shown to be different. Why are these models not being tested as hypotheses about how the catchment system is working? Indeed, we could rather say on the basis of the evidence presented that none of them are really fit for purpose when the additional variables are taken into account.*

Reply:
Indeed, in the current version of the manuscript, our primary objective is to quantify the internal differences between the set of models with similar streamflow output and only a secondary objective is to benchmark the internal state and flux variables against remotely-sensed estimates. However, we agree that deriving qualitative or quantitative measures to evaluate, rank and potentially reject models using the remotely-sensed data in combination with expert knowledge could add value to the study. In response to this criticism, we will introduce quantitative measures to rank and evaluate the models in terms of how plausible it is to consider them behavioral, both in view of the independent remotely-sensed data, and based on expert judgment (e.g. whether model storages are always filling or running empty).

Besides data uncertainty (see comment and reply below), the main reasons we are reluctant to formally and explicitly reject models are the following:

All interpretations and conclusions here (and in any modelling study, really) are also conditional on the individual parameter selection strategies chosen by the individual contributing institutions. The use of different and/or more calibration objectives and/or criteria may have resulted in considerably different model results and associated conclusions. The same is true for the use of different search strategies of the parameter space. In addition, and quite obviously, a rejection of a model (i.e. the combination of model structure, numerical implementation, parameter selection strategy, etc.) would only be valid for the study catchment.

In our opinion, an explicit rejection of one or more models may give the *impression of their general unsuitability*.

Thus, a rejection is therefore not (only) a diplomatic question but, when perceived as *general*, may be unjustified, as these rejected models may be the most suitable models elsewhere.

We admit that this is of course a communication issue. But in the moment a model is formally and explicitly rejected in a paper, it will be perceived as generally useless by many even if it is emphasized that a rejection can essentially always only be conditional on the above points (and many others, such as, needless to say, data uncertainty).

In other words, the label "rejected" will stick, and not the reasons and circumstances. We would really like to avoid that because we think it is neither fair nor justified.

Comment:
*Except that it is not quite that simple, because ALL of the additional variables used in this comparison are subject to significant uncertainty and commensurability issues. And without taking some account of those uncertainties no real testing is possible (it is also worth noting that no account is taken of uncertainty in the original calibration exercise either – why not in 2020? It has been recognized as an issue in model*

*calibration for more than 30 years!).*

Reply:
We agree that assessing the uncertainty of such remote sensing products is important, but not easy. In principle, uncertainty estimates should be provided by the remote sensing models, of which we are final users, but these estimates are usually not available. Even if they were, using them in a meaningful way would uncover many questions, which would go beyond the scope of this work. For this reason, we are highly reluctant to use these data to determine hard rejection thresholds. Rather, we will use them to provide a "soft" assessment of the relative merits of the various models in form of an overall ranking guided by criteria formulated based on "soft", "expert judgement" of trustworthiness of the individual types of remotely-sensed data.

Comment:
*The section on knowledge gaps at the end should be moved to before the model comparison is presented, and should explicitly consider the uncertainty and commensurability issues. Nowhere is there any mention of the uncertainties arising in verification studies of these additional variables, but that is surely significant.*

Reply:
In the next version of the manuscript, we will make the uncertainties and commensurability issues of the evaluation variables clear before using them to evaluate the models.

Comment:
*To give a particular example: models with and without interception storage. This is an example of why more thought is required about what is actually being compared here. One of the reasons why models choose NOT to have an interception store is*
*to reduce the number of parameters required to be calibrated or estimated a priori. But how this works will also depend on how potential evapotranspiration is estimated. Does it include the effects of a wet canopy – especially over rough canopies. This can be really important (and subject to significant uncertainties in effective roughness and humidity deficits because of sensitivities under such conditions). Here the Hargreaves PE formula does not explicitly consider wet canopy conditions, but GLEAMS, with which model outputs are being compared, does). So in what sense (or degree of uncertainty) are these comparable?*

Reply:
This is a very interesting point. We believe that it should not matter too much how potential evaporation is estimated, as this goes as input to hydrological models, and we are comparing the resulting total actual evaporation ($E_A$) between models and testing if $E_A$ is consistent with $E_A$ of GLEAM. In our models, $E_P$ is used as forcing and models will either calculate $E_A$ by explicitly accounting for interception or not. Hence, the $E_A$ from the hydrological models combines, implicitly or explicitly (e.g. when an interception reservoir is included), all forms of evaporation. In this sense, we believe it is comparable with $E_A$ from GLEAM. In the revised version, we will clarify that we do not consider $E_A$ GLEAM to be representative of the truth. However, we believe that the comparison is still valuable to detect outliers, which can be either one/several of the models or the remote-sensing product itself and understand their behavior. GLEAM interception is calculated using precipitation and vegetation characteristics (Miralles et al. 2010). For models with a separate interception module, we also test the consistency of modeled interception $E_I$ with GLEAM $E_I$. In the next version of the manuscript, we will use validation studies of GLEAM (Miralles et al. 2011, Miralles et al. 2016) to describe the uncertainties associated with GLEAM estimates.

Comment:

*Similarly for the soil moisture comparison. The satellite derived estimates really only deal with near-surface moisture (with a depth that varies with wetness) and that in itself is associated with uncertainty, especially near saturation. There is some discussion here about the issue of comparing relative moisture content in the root zone when the different models parameterize that in different ways and a rather odd correlation analysis with the T parameter – can you not think about how (and if) that data can be used as a hypothesis test. There are clearly similar issues with GRACE and snow cover data (e.g. is fact that some models do not predict snow storage on a day important if snow covers are small)*

Reply:
We agree that remotely-sensed soil moisture products provide estimates of near-surface soil moisture, while the represented variable in our models is root-zone soil moisture. The Soil Water Index provides estimates of root-zone soil moisture but requires an estimate of the characteristic time length $T$. In our study, we show that the models have a different representation of root-zone soil moisture dynamics as shown by the variability in optimal $T$-values. As also mentioned in the manuscript, the absolute ranges of remotely-sensed estimates of root-zone soil moisture are hardly comparable with relative soil moisture simulated by our models and many studies apply data matching techniques to rescale the product range towards the model. This implies that only the similarity in dynamic patterns can be used to evaluate the models. One specific aspect of interest for hypothesis testing in this study, is to use the remotely-sensed Soil Water Index to identify periods of drying out, where root-zone soil moisture remains constant at its lowest values. In the next version of the manuscript, we will also define a criterion based on GRACE estimates of total storage anomalies to test the behavior of models in terms of drying out of the catchment.

Even if mean annual snow storage is relatively small, snow can be important for specific events. A highly relevant example in the study region in the 2011 rain- on-snow event that caused widespread flooding in the Ardennes. Using criteria for the recall and precision (Figure 5d,e), we will identify behavioral models to derive plausible snow characteristics of the catchment, which can be confronted with expert knowledge. We will account for the fact that a frequent error of the MODIS product is the snow/cloud discrimination (Hall  Riggs 2007), which could lead to an overestimation of MODIS snow days and therefore a relatively high ratio of miss over actual positives (1-recall).

Comment:

*So rather than have a "so what?" outcome to this paper, I would suggest instead that it should be reformulated into a hypothesis testing framework (EAWAG might be able to make suggestions about how this should be done). This is a real opportunity to frame the issue in this way. Not that because of the uncertainties and commensurability issues that does not imply that any or all of the models will be rejected. That will partly depend on what assumptions and expert knowledge are made in the analysis (– see L450, except that no expert knowledge has really been used in the study as presented). Effectively what you have here are some indices of dynamic behavior with which to evaluate the models – the expert knowledge needs to come in as to how (or IF) those indices (with all the issues with them) can be used to test the models in any way rigorously. This would require very major revisions to the analysis but would make the whole project so much more worthwhile in advancing the modelling process.*

Reply:

Yes, we gladly take up this advice and we agree that the study can benefit from your suggestions to go one step further to answer the "so what?" question. In the revised version, we will define a set of (soft) criteria, in the spirit of behavioral modeling, to evaluate not only how consistent the model-internal dynamics are amongst each other but also if the models provide a consistent behavior with what we expect from expert

knowledge in combination with remotely-sensed data. For each retained parameter set, we will evaluate if models can be considered behavioral. This will provide us with an indication of the plausibility of each model to describe several retained aspects of catchment functioning. These results will be presented in a way to allow us to identify potential trade-offs in model capabilities and understand if certain aspects of the parametrization cause a specific model behavior.

Comment:
*L35 There are other variables that have been used (and much earlier than the studies cited) – eg. saturated contributing areas (Beven and Kirkby, HSB 1979; Güntner et al., HP 1999; Blazkova et al., HP 2002) and patterns of water tables in many piezometers (Seibert et al., HP 1997; Lamb et al. AWR 1998; Blazkova et al. WRR 2002).*

Reply:
Yes agreed, we will add these early studies.

Comment:
*L228 drop infinitely – this is misleading. Theoretically yes, but it is directly related to baseflow outflow by water balance and you do not expect baseflow to go to zero in droughts for these catchments. It can also have the advantage of reducing number of parameters required.*

Reply:
Yes, it was indeed a theoretical 'infinitely'. We will drop it to avoid confusion.

Comment:
*L288 How can GLEAM potential ET be less than the annual estimate cited in L271*

[Figure]

*(and how undertain are those estimates)*

Reply:
The reason why GLEAM potential evaporation ($E_\mathrm{P}$) is less than the annual actual evaporation estimate is because GLEAM total actual evaporation ($E_\mathrm{A,GLEAM}$) is calculated as $E_\mathrm{A,GLEAM} = E_\mathrm{I} + S * E_\mathrm{P}$, with $E_\mathrm{I}$ is interception and $S$ is a stress factor depending on the root-zone available water and dynamic vegetation information (Miralles et al. 2011). GLEAM interception is calculated separately and only depends on precipitation and vegetation characteristics. We will clarify this point in the revised version of the paper.

GLEAM does not provide uncertainty ranges, but there are studies that compare GLEAM evaporation estimates with other evaporation products and FLUXNET stations (Miralles et al. 2011, Miralles et al. 2016). These studies will be used to describe the uncertainty of the remotely-sensed evaporation estimates.

Comment:
*L397 ecosystems have adapted – but these ecosystems have not surely. In this area they have been affected by forestry and agricultural practices for thousands of years.*

Reply:
Yes, the area is affected by commercial forestry and agricultural practices since many years. However, approximately half of the catchment is covered by forests and it seems very unlikely that transpiration is reduced to almost zero for several days in a row each year over the entire catchment. This is also not supported by the satellite-based evaporation and soil moisture products. The MODIS Normalized Difference Vegetation Index (NDVI) data may provide some additional useful information. We will reformulate

this point in the next version.

References
Hall, D. K., Riggs, G. A. (2007). Accuracy assessment of the MODIS snow products. *Hydrological Processes*, 21(12), 1534–1547. https://doi.org/10.1002/hyp.6715

Miralles, D. G., Gash, J. H., Holmes, T. R. H., De Jeu, R. A. M., Dolman, A. J. (2010). Global canopy interception from satellite observations. *Journal of Geophysical Research Atmospheres*, 115(16), 1–8. https://doi.org/10.1029/2009JD013530

Miralles, D. G., Holmes, T. R. H., De Jeu, R. A. M., Gash, J. H., Meesters, A. G. C. A., Dolman, A. J. (2011). Global land-surface evaporation estimated from satellite-based observations. *Hydrology and Earth System Sciences*, 15(2), 453–469. https://doi.org/10.5194/hess-15-453-2011

Miralles, D. G., Jiménez, C., Jung, M., Michel, D., Ershadi, A., Mccabe, M. F., et al. (2016). The WACMOS-ET project - Part 2: Evaluation of global terrestrial evaporation data sets. *Hydrology and Earth System Sciences*, 20(2), 823–842. https://doi.org/10.5194/hess-20-823-2016

---

## Author Comment (AC2) · 16 Jun 2020

We thank the anonymous referee 2 for carefully reading our manuscript and providing interesting suggestions. We provide an answer to each comment below.

Comment:
*Bouaziz et al. (2020) evaluate 12 hydrologic models for three medium-sized Belgian catchments, all established and calibrated by eight research groups. Although the spatially aggregated streamflow performance differences among models are negligible, the internal model states and processes (can) differ significantly. This paper is an interesting diagnostic study, with nice figures. I have some minor comments which the*

[Figure]

*authors should consider to address.*

*First of all, it is nice to see the huge collaborative efforts across many institutes behind this model inter-comparison study. This study with many details shows large differences among 12 hydrologic models and even larger differences against different remotely sensed products. Something what a reader would expect. I encourage the authors to stress more clearly, what is the main "take-home" message of the main paper.*

Reply:
Thank you, this is a good suggestion. In the revised version, we will more clearly stress the main take-home message, which is to underline and demonstrate that models that are calibrated to streamflow can generate similar high-performance levels in reproducing streamflow, but that they use different "pathways" to do so, i.e. all representing the system in a different way. In the next version, we will also emphasize on the use of remotely-sensed products in combination with expert knowledge to evaluate if models can be considered behavioral.

Comment:
*Because the authors did not use an ensemble of model structures from a modular framework (e.g., FLEX, FUSE), which could properly address those differences or individual model deficiencies step by step (by identifying individual hypothesis), in their study they cannot clearly separate and identify, which hidden hydrological processes can help improve model functioning against those reanalysis products. Could you please comment on this?*

Reply:

We agree that the set of models does not easily allow for a step-by-step identification of differences in individual hypothesis, as they are mostly full-grown models. Perhaps only the subset of FLEX models M2 to M5 allows us to identify stepwise differences in internal model representations. However, we grouped models with similar parametrizations in Tables 2 and 3 and focused our analyses on model components that were present in most models (evaporation, snow, root-zone soil moisture, total storage). One important systematic difference that we found amongst the models is the significant drying-out each summer for some models. In the next version of the manuscript, we will hypothesize on the model parametrization that leads to this behavior. We believe that these specific findings can help to identify some model functioning aspects that can be improved by adapting model parametrization. In the revised version, we will also include more detailed findings on the plausibility of model behavior for a selection of criteria related to the remote sensing data and expert knowledge (as suggested by referee Prof. Keith Beven).

Comment:
*Line 80,140+: evaporation => "evapotranspiration"? Please don't forget about the plants! Hargreaves-Samani formula is for evapotranspiration, not for evaporation only.*

Reply:
We will clarify that we have used the term "evaporation" to describe the sum of all evaporation components (including transpiration, soil evaporation, interception, sublimation and open water evaporation when applicable). It is perhaps a matter of taste, but following Savenije (2004) and Miralles et al. (2020), we are using the term evaporation instead of evapotranspiration for all evaporative fluxes. We will make sure to clearly state this in in the text and in Table 1 to avoid confusion.

Comment:

[Figure]

*Line 85: streamflow => "runoff", because of the unit*

Reply:
We strongly prefer using "streamflow" to describe the flow of water in the river and have consistently applied this terminology throughout the manuscript, irrespectively of the unit. In our view, runoff is more generic and can refer to (sub)surface flow, which is not yet in the river.

Comment:
*Line 86: You should start this sentence that this is a headwater basin of ID1*

Reply:
Agree, we will add this in the revised version.

Comment:
*Line 101: I guess the authors could have used a bit more advanced method for interpolation rain gauge observation instead of the Thiessen polygons, to better account for input error uncertainty, e.g. kriging or its variants. The uncertainty in the meteorological inputs is not mentioned in the manuscript.*

Reply:
We agree that there is always uncertainty in meteorological input data and will mention this in the revised version. Another method to interpolate precipitation could also have been feasible. However, the number of available precipitation stations would likely not be enough to perform a meaningful Kriging interpolation. The advantage of Thiessen polygons is that extremes are not averaged out, which would occur in any other type of interpolation. Many threshold processes are controlled by these extremes. Besides,

our primary aim was to make sure that the same forcing data was used by all research groups.

Comment:
*Line 114: PET method is based on Priestley Taylor, which is different from section 3.1. How is it compatible with section 3.1 and overall results?*

Reply:
We believe that the different methods to estimate potential evaporation should not impede us from testing the consistency between the resulting total modeled actual evaporation $E_A$ and estimated $E_A$ from GLEAM. This is also supported by the findings of Oudin et al. (2004), who reported similar model performance irrespective of the method applied to estimate potential evaporation. Additionally, we do not consider $E_A$ from GLEAM to be representative of the truth, but the comparison can enable us to detect outliers (either one/several models or the remote sensing data).

Comment:
*Line 143: how did you spatially average soil moisture?*

Reply:
We calculated the mean soil moisture over all SCATSAR-SWI1km pixels within the Ourthe catchment. We will clarify this in the revised version.

Comment:
*Line 153: I guess, your entire study domain is just a single GRACE pixel. I am quite skeptical for using it at all, as it's beyond the limits of its usability. The original raw GRACE signal is based on a much larger region (3degrees). You may better wait for*

*the GRACE-FO, which has much finer native resolution...*

Reply:
Yes, the catchments indeed fit in single GRACE pixels. At this small scale, we agree that we must be careful with possible 'signal leakage' from surrounding areas, which increase the uncertainty. We believe that the GRACE signal is still informative and the best currently available, despite the larger errors and uncertainties at this small scale compared to large scale spatial averages. It should also be noted that we are not using GRACE for model calibration. Instead we are testing if the modeled regional seasonal water storage anomalies are consistent with GRACE estimates. Additionally, GRACE signals in small scale catchments were shown to provide valuable information for hydrological modeling (Rakovec et al. 2016, Nijzink et al., 2018). In the next version of the manuscript, we will use total storage anomalies provided by the three processing centers instead of taking the mean of the three to better account for the uncertainty. In the future, it would surely be interesting to work with GRACE-FO, but this is unfortunately not available for our study period.

Comment:
*Section 4.1 I guess all models were applied in spatially lumped manner, i.e. no spatially distributed mode, isn't it? Please write down explicitly in this section.*

Reply:
We will clarify if models are lumped or (semi-)distributed in Table 2. The wflow-hbv model is the only fully distributed model, but parameter values are mostly uniform over the catchment area. The FLEX-Topo model is a semi-distributed based on hydrological response unit within each Thiessen polygon. All other models are lumped.

Comment:
*Line: 179-181 This analysis was done here, or in previous study? Not clear, please specify, and provide link to the transferability results. Curious to see them.*

Reply:
We will clarify this part in the next version of the manuscript. The analysis was done in the previous study (de Boer-Euser et al. 2017), in which the models were calibrated for the Ourthe at Tabreux and parameter values were transferred to two neighboring and two nested catchments (including the Semois at Membre-Pont and Ourthe Orientale at Mabompré). The previous study covered the study period 2001-2010. In the current study, we use the previous calibration of the Ourthe at Tabreux for the three catchments and ran the set of models for an additional period from 2011 to 2017.

Comment:
*How many behavioral parameter sets per model were used? Is the number same per hydrological model? Here referring to error bars in Fig. 3, Fig. 4 and elsewhere.*

Reply:
For each model, we retained 20 feasible parameter sets. However, the width of the error bands varies due to the different calibration strategies applied by the modelers.

Comment:
*Figure 9: Is it possible to rank the models according to their performance? Which one seems to be most relevant and how it compares to e.g. an operational model, if that's available? Please think about putting some implications to the paper.*

Reply:
Thank you for this interesting suggestion. In Figure 9, the models are currently ranked from the highest to the lowest performance according to the Euclidean distance of Nash-Sutcliffe efficiencies of the streamflow and the logarithm of the streamflow (see Figure 3 and Table 2). In the next version of the manuscript, we will adapt Figure 9 to rank the models for each criterion.

In the revised version, we will mention that GR types of models (as GR4H) are used for operational purposes in France and that a lumped version of the HBV model is currently used by the Dutch operational system. In fact, each of these models could potentially be used operationally.

In the next version of the manuscript, we will introduce some soft criteria to rank and evaluate models in terms of how plausible it is to consider them behavioral based on the remotely-sensed data and expert knowledge (following the suggestions of referee Prof. Keith Beven).

References

de Boer-Euser, T., Bouaziz, L., de Niel, J., Brauer, C., Dewals, B., Drogue, G., et al. (2017). Looking beyond general metrics for model comparison ndash; Lessons from an international model intercomparison study. *Hydrology and Earth System Sciences*, 21(1), 423–440. https://doi.org/10.5194/hess-21-423-2017

Miralles, D. G., Brutsaert, W., Dolman, A. J., Gash, J. H. (2020). On the use of the term "Evapotranspiration." *Earth and Space Science Open Archive*, 8. https://doi.org/10.1002/essoar.10503229.1
Nijzink, R. C., Almeida, S., Pechlivanidis, I. G., Capell, R., Gustafssons, D., Arheimer, B., et al. (2018). Constraining Conceptual Hydrological Models With Multiple Information Sources. *Water Resources Research*, 54(10), 8332–8362. https://doi.org/10.1029/2017WR021895

Oudin, L., Hervieu, F., Michel, C., Perrin, C., Andréassian, V., Anctil, F., Loumagne, C. (2005). Which potential evapotranspiration input for a lumped rainfall-runoff model? Part 2 - Towards a simple and efficient potential evapotranspiration model for rainfall-runoff modelling. *Journal of Hydrology*, 303(1–4), 290–306. https://doi.org/10.1016/j.jhydrol.2004.08.026

Rakovec, O., Kumar, R., Mai, J., Cuntz, M., Thober, S., Zink, M., et al. (2016). Multiscale and Multivariate Evaluation of Water Fluxes and States over European River Basins. *Journal of Hydrometeorology*, 17(1), 287–307. https://doi.org/10.1175/JHM-D-15-0054.1

Savenije, H. H. G. (2004). The importance of interception and why we should delete the term evapotranspiration from our vocabulary. *Hydrological Processes*, 18(8), 1507–1511. https://doi.org/10.1002/hyp.5563
* * *

---

## Author Comment (AC3) · 30 Jun 2020

We thank Shervan Gharari for the interesting discussion on our manuscript and we reflect on each comment below.

Comment 1:
*Following the comments by Prof. Beven, I would like to ask the authors a more direct question: "is there any practical use in exploiting the remote sense data in constraining the hydrological models at the scale of interest?" In some applications, bucket-style models are constrained based on evaporation products. I understand that the evaporation products can be used for practical purposes and possibly as*

[Figure]

*a preliminary benchmark, however, my concerns are: (1) the reduced uncertainty coming from confronting the model with another set of products might result in an "illusion of certainty" in simulation and patterns. As an example, refer to Wang et al., 2015a to see the possible uncertainty in the transpiration/evaporation products from a model. (2) the whole modeling purpose is to predict unknown and come up with the temporal and spatial prediction of some states and fluxes. We then set up a model, say it does or doesn't get the spatial pattern, train it with the result of another model, and then it gets the spatial pattern probably right. What is the end goal of this practice? We can probably join efforts with the developers of the already existing products to improve their products rather than just being a user. Or use a machine-learning algorithm to capture what the patterns in the products are. My question in short; do we learn? Or do we produce a similar product (hopefully a better one)?*

Reply 1:
Thank you for raising this interesting discussion. However, it should be clear that we are not using the remote sensing data to calibrate the models. The primary aim of our study is to quantify the differences in internal process representation for a set of models with similar streamflow performance. A secondary objective is to benchmark the internal state and flux variables against remote sensing data. We do not consider the remote sensing data to be representative of the truth, but we use the data to detect potential outliers (being the data itself or one/several models).

It should also not be forgotten that streamflow data also relies on a model with associated uncertainty, namely the rating curve. Of course, there is also (and probably more) uncertainty in evaporation data based on remote sensing, but these products are often also validated against in-situ FLUXNET stations. The use of remote sensing data is valuable as additional independent source of information. Therefore, hydrological applications could benefit from the use of remote sensing data for calibration,

depending on the purpose of the application (e.g. flow predictions in a poorly-gauged catchment).

Comment 2:
*GRACE is rather coarse for the basins of interest. It is suggested that the GRACE data should be used for catchment above 150,000 square kilometers (Rodell et al., 2011). This might be counterintuitive; visualization of GRACE over a large area will show that the data is more diffused that its actual resolution. Also, GRACE is very uncertain in itself, using a mean value of its three or more variations may result in deliberate killing of uncertainty (Scanlon et al., 2018).*

Reply 2:
We agree that we should be careful with using the GRACE signal over a single pixel, as possible signal leakage from surrounding areas can increase the uncertainty. Even if the catchment area fits within one GRACE pixel, we hypothesize that the signal is of interest as benchmark against which to evaluate the modeled regional seasonal water storage anomalies. In the next version of the manuscript, we will mention these issues in the Data section instead of the Discussion. We will also show the uncertainty of total storage anomalies provided by the three processing centers.

Comment 3:
*Checking the consistency of input data is essential before starting the modeling phase. The knowledge gap Section, in my point of view, can be moved earlier in the manuscript and can be populated by quantified evaluation of the available data sets for forcing/calibrating the models. Basically, from the data sets, you have all the components that you need to close the water balance.*

[Figure]

$e = sum(P - E - Q) \triangle t - \triangle S$

*Can you get e close to zero over a month or a year? (similar experiment to Wang et al., 2015b) Do you have a sense of uncertainty or disagreement between the precipitation and rain gauges?*

*As low-hanging fruit, it is possible to have an understanding of approximate interception/transpiration for this region. Any flux towers? Study sites from Luxembourg might be helpful? It seems the product you used in this study for evaporation underestimates interception significantly. From their website it seems interception is set to 10% globally (if I interpreted it correctly). Do you perhaps know this ratio for the region of study from this data (model outcome)? This seems to contradict some earlier paper by co-authors on the global uncertainty of the interception/transpiration. Soil evaporation from the used product may include many assumption or simplification similar to land models (Bohn and Vivoni, 2016). Another low-hanging fruit! Can we perhaps estimate the recession coefficient from the hydrograph and compare it with calibrated values in the models to see actually which model structure allows for a more accurate estimation of recession coefficient when calibrated and why [as doctor-father always says]. For example, land models are not suited for this recession inference (Gharari et al., 2019).*

Reply 3:
We agree that checking the consistency of the input data is essential and a standard procedure in any modeling practice. We will mention the uncertainty in the evaluation variables in the Data section rather than in the Discussion. We limited the scope of our study to the evaluation of internal variables against remote sensing data. Comparing catchment scale averages with point measurements of evaporation also faces substantial commensurability issues. However, it is good to know that GLEAM evaporation was evaluated using FLUXNET data (including the station of Lonzee

which is close to the study area). For this station, a correlation coefficient between GLEAM and FLUXNET of 0.91 for the daily time step and similar annual rates are reported in Miralles et al. (2011). In Miralles et al. (2016), the likely underestimation of interception evaporation and the likely overestimation of total actual evaporation compared to other products is discussed. We will make use of these evaluation studies to describe the uncertainty of the evaluation variables and derive soft criteria to evaluate the plausibility of the set of models.

Comment 4:
*The area is mostly agriculture, is there any regulation on the stream than may affect your inference. Referring to section 5.3, the area is mostly agriculture, to my understanding, the Sumax/root zone storage co-evolution is hypothesized for forests (that has a life of more than a year). Agricultural lands do not follow any of that logic, it does what farmers do (there might be some correlation). Maybe you can argue around the rain-fed nature of agriculture in this region but still, crops have a lifespan of a season. Land models can see the variation of leaf area index (LAI) and with some modification even variation of root zones over period of time. That can be a better testbed for exploring root zoon hypothesis than bucket-style models.*

Reply 4:
We agree with your suggestion of looking at vegetation indices for additional information on this matter. Given that almost half of the catchment of the Ourthe at Tabreux is covered by forests (46%), we do not expect transpiration rates to drop to approximately zero for several days in a row each year over the entire catchment.

Comment 5:
*"The $T$-value has previously been positively correlated with root-zone storage capacity, assuming a high temporal variability of root-zone soil moisture and therefore a low*

$T$-value for small root-zone storage capacities $S_{R,max}$ (Bouaziz et al., 2020)". Possible that I totally get it wrong, but if I understand correctly, Bouaziz et al., 2020 used a hydrological model in combination with satellite observation. Is this a model result that is used for intercomparing rather than the satellite observation itself? Maybe separate the data (products) into groups of "directly observed" and "inferred based on a model".

Reply 5:
The modeled root-zone soil moisture of each model is evaluated against satellite observations of the Soil Water Index. The Soil Water Index is provided by the Copernicus Global Land Service for several $T$-values. The Soil Water Index is derived from near-surface soil moisture to represent root-zone soil moisture but requires an estimate of the $T$-value. In Bouaziz et al. (2020), we found a link between the optimal $T$-value and the root-zone storage capacity. So, to answer the question, yes, we are using the by Copernicus provided satellite data for the comparison, and yes the data is inferred based on an algorithm. However, all the evaluation variables we are using are relying on algorithms, even streamflow is not directly measured.

Comment 6:
Upscaling of snow cover to basin level is a tricky business. Snow storage, snowpack extent may not be uniform over an area (Cherkauer et al., 2003). Also, the snowpack can persist with temperature much higher than the phase-change temperature identified in the model. Snowpack may also stays longer under canopy. The phase-change temperature can have a range, for example, for the VIC model this is a transitional span of temperature (for example from -2C to 2C) that can be tuned for the same reason (snow precipitation). I would suggest checking the snow extent versus the temperature first. This might give insight into whether or not any model can simulate the observed snow extent given the temperature. Also, snow under canopy may stay longer, does the product you use capture that?

Reply 6:
Thank you, it is indeed a good suggestion to compare snow extent and temperature. As the snow cover relies on MODIS imagery, it is unlikely that snow under the canopy is captured by the product.

Comment 7:
*As a modeler that might be interested to model the basins of interest, what is the take-home message for me. I assume one of the aims of an inter-comparison project is the knowledge mobilization of already known facts about basin(s) to the wider community. This can be done better in this manuscript I would say. Perhaps, identifying the target audience. Is the manuscript targeted for catchment hydrology? Or Large-scale hydrology? The current manuscript does not server any.*

*I would say, as coordination of the large team takes a lot of efforts and work, maybe give a new dimension to your paper by elaborating the organizational efforts put into this study (why did you initiate this inter-comparison, why the current list of models and authors, what made you to choose them? what effect it might have on real-world application, etc).*

Reply 7:
We will clarify that the take-home message of the manuscript is to underline and demonstrate that models that are calibrated to streamflow can generate similar high-performance levels in reproducing streamflow, but that they use different "pathways" to do so, i.e. all representing the system in a different way. In our opinion, this is relevant for catchment hydrology and large-scale hydrology as both rely on model selection. In this context, the selection of the Ourthe catchment is more a case study

to demonstrate the different internal process representation.

In the next version of the manuscript, we will also use expert knowledge in combination with the remote sensing data to evaluate the plausibility of model behavior for a selection of criteria.

The study is a joint research effort of institutes and universities gathering each year at Liège Université for the symposium on hydrological modelling of the Meuse basin to exchange knowledge and work together on the Meuse basin, as also mentioned in the discussion. In the revised version, we will discuss the challenges inherent to such a comparison study.

Comment 8:
*I would suggest the authors clarify their research equations in the beginning and come back to the research questions in the conclusions. In the current version, there are no tangible research questions. For example, "Haddeland et al. (2011) and Schewe et al. (2014) compared global hydrological models and found that differences between models are a major source of uncertainty." I think this is what you can reflect/elaborate on in your conclusions (hopefully quantitively)?*

*One collusion from this study can be for example, "a two-bucket model with snow component is sufficient enough to get the dynamic of the data we selected". Can this be one of your conclusions?*

*Some studies from the land modeling community can be helpful in this regard. For example, Bets et al., 2015 provided a structure for the comparison (including evaluation, comparison, benchmarking, fit for purpose, utilizing the available data,*

*etc). Following this structure or similar structures can hopefully clarify the manuscript more. One benchmarking strategy could be ensemble simulation of all models within their prespecified parameter ranges. This can be the basis for comparison when the model is calibrated on the streamflow and subsequently on other data sets such as evaporation in a stepwise fashion. This seems to be not a lot of work as the models are already set up.*

*Moreover, the land model studies can provide more insight into large scale modeling and their related issues. For example, Crow et al. (2003) is a classic example. In my point of view far ahead of its time and not very well received [the same work nowadays would probably have 20 authors with the same citation level in a single year and will be magically highlighted!]. Another great example to show uncertainty in large scale models in reproducing mean and variability (Koster, R.D. and P. Mahanama 2012) with a very simple model. Another example is Hurkmans et al. 2008. These studies and similar works may provide a better understanding of the exploitation of additional data in large scale modeling and associated uncertainties.*

Reply 8:
We will sharpen the research question and conclusion. In this study, we test the hypothesis that models with similar streamflow performance have similar internal process representation, as stated in the last paragraph of the introduction. We conclude by saying that models have different internal representations of water storage and release. This suggests that all models can't simultaneously be different and close to reality.

In the next version of the manuscript, we will go one step further by evaluating the plausibility of the models and identify behavioral models in view of the remote sensing data and expert knowledge.

Although an interesting suggestion for follow-up studies to constrain the models using the remote sensing data, this is outside the scope of the current study.

Comment 9:
*Concerning FLEX-Topo. It seems to be the only semi-distributed model among all the other models. Have you properly constrained the component of this model (or do you have enough expert knowledge to do so)? It would be good to highlight the advantage of the semi- distributed model here if any. The control over the different components of FLEX-Topo becomes increasingly hard if the code is written separately for each landscape (different structures). I tried to have a similar code for each landscape and recreate the desired structure just by adjusting the parameters. That provides better control over the performance of each landscape. For example, did you check the transpiration of each landscape? Sometimes it is the case that soil moisture from one landscape is empty and the other landscapes are evaporating at the maximum rate.*

Reply 9:
The calibration of the models was done in the previous study (de Boer-Euser et al., 2017), in which we found that models had similar streamflow performances based on commonly used metrics. Here, we are consistently using the previous calibration to assess internal model representation. Of course, model deficiencies appearing in this study are helpful to improve the parametrization of the model but going into such details for each model is outside the scope of the current study.

Comment 10:
*I didn't know that FLEX-Topo got a sublimation component. How that is implemented? Is sublimation a major process in the region of study? I would not say so. Sublimation*

[Figure]

*is also a tricky process; a magical one! it can account for uncertainty in snowpack similar to the transpiration for soil moisture. There is also a refreezing formulation for one of the models. Interested to know how that happens in a model that may not close the energy balance. It would be good to include all the model formulation in the Appendix if not too much work.*

Reply 10:
The raised discussion is interesting. The sublimation component implemented in the FLEX-Topo model is described in de Boer-Euser (2017). It is a simple representation that allows evaporation from the snowpack at potential rate, provided there is enough snow storage. As precipitation mostly falls as rain and not as snow, we agree that it is not a major process for the study region. Additionally, the magnitude of sublimation is likely to be rather low due to the limited direct radiation in winter in Belgium. We are providing the most relevant equations for our study in Table 2 and 3 and we refer to the references on the model descriptions for more details.

Comment 11:
*The figures presenting the results are very hard to follow. I am not sure if I understand most of them. I would suggest simplifying them.*

Reply 11:
Unfortunately, given the general character of this comment, it is unclear which aspects of the figures are very hard to follow. Following the comments of the referees, we intend to adapt Figure 9 in the next version.

Comment 12:
*A question from Prof. Beven and maybe the authors; is that possible to even reject*

*a model in large scale modeling? From my experience and due to the issue of scale (and observation at that scale), most of the models can be accepted. For example, in a recent modeling effort that we have done (Gharari et al., 2020), the VIC model with the only micropore and with only macropore water movement yields the same result when calibrated (exploring the inclusion of macropore water movement in land models; aligned with Beven and Germann 1982, to Beven 2018). How should I justify macropore versus micropore at that scale for a colleague whose entire career is focused on how to properly/mathematically represent micropore water movement? What is the path forward? I appreciate your thoughts on that.*

Reply 12:

Thank you for raising this interesting comment. In our study, we are reluctant to reject models because any rejection is conditional on many factors, including the individual parameter selection strategies chosen by the individual contributing institutions. And model rejection would only be valid for the study catchment. And these specific circumstances could be forgotten if taken out of context and the label "rejected" would remain, even if this is not justified. Additionally, there is a large uncertainty in the used evaluation variables. The uncertainty estimates of remote sensing data are usually not available and using them in a meaningful way would uncover many questions, which is outside the scope of the current study.

Instead, we propose to evaluate and rank the models and define soft criteria to test the plausibility of model behavior in terms of the remote sensing data and expert knowledge. We will include this in the next version of the manuscript.

Regarding the specific case you are mentioning on macropore versus micropore flow, some answers might be provided in Zehe et al. (2014). We believe this also depends on the study area and on the availability of evaluation variables and their uncertainty

estimates.

References

de Boer-Euser, T., Bouaziz, L., de Niel, J., Brauer, C., Dewals, B., Drogue, G., et al. (2017). Looking beyond general metrics for model comparison ndash; Lessons from an international model intercomparison study. *Hydrology and Earth System Sciences*, 21(1), 423–440. https://doi.org/10.5194/hess-21-423-2017

de Boer-Euser, T. (2017). Added value of distribution in rainfall-runoff models for the Meuse basin. Doctorale thesis TU Delft Repository. https://doi.org/10.4233/uuid:89a78ae9-7ffb-4260-b25d-698854210fa8

Miralles, D. G., Holmes, T. R. H., De Jeu, R. A. M., Gash, J. H., Meesters, A. G. C. A., Dolman, A. J. (2011). Global land-surface evaporation estimated from satellite-based observations. *Hydrology and Earth System Sciences*, 15(2), 453–469. https://doi.org/10.5194/hess-15-453-2011

Miralles, D. G., Jiménez, C., Jung, M., Michel, D., Ershadi, A., Mccabe, M. F., et al. (2016). The WACMOS-ET project - Part 2: Evaluation of global terrestrial evaporation data sets. *Hydrology and Earth System Sciences*, 20(2), 823–842. https://doi.org/10.5194/hess-20-823-2016

Zehe, E., Ehret, U., Pfister, L., Blume, T., Schröder, B., Westhoff, M., et al. (2014). HESS Opinions: From response units to functional units: A thermodynamic reinterpretation of the HRU concept to link spatial organization and functioning of intermediate scale catchments. *Hydrology and Earth System Sciences*, 18(11), 4635–4655. https://doi.org/10.5194/hess-18-4635-2014
Beven, K. and Germann, P., 1982. Macropores and water flow in soils. Water resources research, 18(5), pp.1311-1325.

Beven, K., 2018. A Century of Denial: Preferential and Nonequilibrium Water Flow in Soils, 1864-1984. Vadose Zone Journal, 17(1).

Bouaziz, L. J., Steele-Dunne, S. C., Schellekens, J., Weerts, A. H., Stam, J., Sprokkereef, E., Winsemius, H. H., Savenije, H. H., and Hrachowitz, M.: Improved understanding of the link between catchment-scale vegetation accessible storage and satellite-derived Soil Water Index, Water Resources Research, https://doi.org/10.1029/2019WR026365, 2020.

Best, M.J., Abramowitz, G., Johnson, H.R., Pitman, A.J., Balsamo, G., Boone, A., Cuntz, M., Decharme, B., Dirmeyer, P.A., Dong, J. and Ek, M., 2015. The plumbing of land surface models: benchmarking model performance. Journal of Hydrometeorology, 16(3), pp.1425-1442.

Bohn, T. J., and E. R. Vivoni, 2016: Process-based characterization of evapotranspiration sources in the North American monsoon region, Water Resour. Res., 52(1), 358-384, doi:10.1002/2015WR017934.

Wang, S., Huang, J., Yang, D., Pavlic, G., and Li, J., 2015. Long‐term water budget imbalances and error sources for cold region drainage basins. Hydrological processes, 29(9), pp.2125-2136.

Wang, S., Pan, M., Mu, Q., Shi, X., Mao, J., Brümmer, C., Jassal, R.S., Krishnan, P., Li, J. and Black, T.A., 2015. Comparing evapotranspiration from eddy covariance measurements, water budgets, remote sensing, and land surface models over Canada. Journal of Hydrometeorology, 16(4), pp.1540-1560.

Crow, W.T., Wood, E.F., and Pan, M., 2003. Multiobjective calibration of land surface model evapotranspiration predictions using streamflow observations and spaceborne surface radiometric temperature retrievals. Journal of Geophysical Research: Atmospheres, 108(D23).

Scanlon, B.R., Zhang, Z., Save, H., Sun, A.Y., Schmied, H.M., Van Beek, L.P., Wiese, D.N., Wada, Y., Long, D., Reedy, R.C. and Longuevergne, L., 2018. Global models underestimate large decadal declining and rising water storage trends relative to GRACE satellite data. Proceedings of the National Academy of Sciences, 115(6), pp.E1080-E1089.

Rodell, M., McWilliams, E.B., Famiglietti, J.S., Beaudoing, H.K., and Nigro, J., 2011. Estimating evapotranspiration using an observation based terrestrial water budget. Hydrological Processes, 25(26), pp.4082-4092.

Haddeland, I., Clark, D. B., Franssen, W., Ludwig, F., Voß, F., Arnell, N. W., Bertrand, N., Best, M., Folwell, S., Gerten, D., Gomes, S., Gosling, S. N., Hagemann, S., Hanasaki, N., Harding, R., Heinke, J., Kabat, P., Koirala, S., Oki, T., Polcher, J., Stacke, T., Viterbo, P., Wee- don, G. P., and Yeh, P.: Multimodel estimate of the global terrestrial water balance: Setup and first results, Journal of Hydrometeorology, 12, 869–884, https://doi.org/10.1175/2011JHM1324.1, 2011.

Schewe, J., Heinke, J., Gerten, D., Haddeland, I., Arnell, N. W., Clark, D. B., Dankers, R., Eisner, S., Fekete, B. M., ColoÌAn-GonzaÌAlez, F. J., Gosling, S. N., Kim, H., Liu, X., Masaki, Y., Portmann, F. T., Satoh, Y., Stacke, T., Tang, Q., Wada, Y., Wisser, D., Albrecht, T., Frieler, K., Piontek, F., Warszawski, L., and Kabat, P.: Multimodel assessment of water scarcity under climate change, Proceedings of the National Academy of Sciences of the United States of America, 111, 3245–3250, https://doi.org/10.1073/pnas.1222460110, 2014.

Koster, R.D., and P. Mahanama, S.P., 2012. Land surface controls on hydroclimatic means and variability. Journal of Hydrometeorology , 13(5), pp.1604-1620.

Hurkmans, R.T.W.L., De Moel, H., Aerts, J.C.J.H. and Troch, P.A., 2008. Water balance versus land surface model in the simulation of Rhine river discharges. Water resources research, 44(1).

Gharari, S., Clark, M., Mizukami, N., Wong, J.S., Pietroniro, A., and Wheater, H., 2019. Improving the representation of subsurface water movement in land models. Journal of Hydrometeorology, (2019).

Gharari, S., Clark, M. P., Mizukami, N., Knoben, W. J. M., Wong, J. S., and Pietroniro, A.: Flexible vector-based spatial configurations in land models, Hydrol. Earth Syst. Sci. Discuss., https://doi.org/10.5194/hess-2020-111, in review, 2020.

Cherkauer, K.A., Bowling, L.C. and Lettenmaier, D.P., 2003. Variable infiltration capacity cold land process model updates. Global and Planetary Change, 38(1-2), pp.151-159.

---

## Referee Comment (RC3) · Anonymous Referee #3 · 1 Jul 2020

This manuscript proposes a multi-objective model evaluation to compare a number of different hydrological catchment models. While this is certainly a valuable task, I honestly have very split feelings about this study. The general idea of multi-objective testing is not new, but very important, and the comparison of several models is an interesting novel aspect. However, I have a number of fundamental concerns, which would require new data and computations to be addressed.

1) The study is based on only three catchments. Several studies have shown how variable results between catchments can be, and these days with more and more data sets being available, the use of just three catchments seems a bit surprising for this type of study.

2) The study addresses different storages, including snow storage. However, the im-

portance of snow in the test catchments is minor. I could not find any information on the relative importance of snow (the info of about one month of snow cover is incomplete as this does not say anything about the amount of water stored as snow). Still, my general understanding is that snow does not play any major role in these catchments. This is probably also the reason why the authors can get away with not using any elevation zones for modelling snow processes.

3) Each of the storage estimation used for model testing is associated with significant observation uncertainties. There is also a scale-mismatch which results in additional uncertainties. These issues have to be considered!

4) Another point that seems to be missing is that each of the models of course also is affected by parameter uncertainties (which will influence the simulated storages). Perhaps I am missing something, but as I understand, single parameter sets are used for each model. This is not sufficient; we know that the same model can result in very different internal simulations because of parameter uncertainty. This leaves me wondering how much of the differences presented here are due to parameter uncertainty rather than due to model differences.

---

## Author Comment (AC4) · 17 Jul 2020

We thank the anonymous referee 3 for his/her comments and provide an answer to each point below. However, we are surprised and puzzled by the review as we think that most of the points raised by the referee are covered in the manuscript. The overall assessment and relatively minor comments do not seem to correspond with the associated evaluation report of the referee.

Comment 1:
*This manuscript proposes a multi-objective model evaluation to compare a number of different hydrological catchment models. While this is certainly a valuable task, I*

[Figure]

*honestly have very split feelings about this study. The general idea of multi-objective testing is not new, but very important, and the comparison of several models is an interesting novel aspect. However, I have a number of fundamental concerns, which would require new data and computations to be addressed.*

*The study is based on only three catchments. Several studies have shown how variable results between catchments can be, and these days with more and more data sets being available, the use of just three catchments seems a bit surprising for this type of study.*

Reply 1:

The overall objective and novel aspect of this study is to analyze and quantify the differences in the magnitudes and dynamics of multiple internal state and flux variables of multiple models that provide similar performance characteristics when exclusively evaluating them against observed streamflow. We will further emphasize this in the introduction. More specifically, in addition to streamflow, we quantify the differences of five model internal state and flux variables for twelve models in three catchments. The primary aim of our study is to demonstrate and underline that models that are calibrated to streamflow can generate similar, high performance levels in reproducing streamflow, but that they use different "pathways" to do so, i.e. all representing the system in a different way. A secondary objective is to benchmark the internal state and flux variables against remote sensing data. We will clarify this in the introduction of the revised manuscript.

As in previous comparison studies, there needs to be a trade-off in what can be done in one single experiment. This is not only a question of computational capacity and time restrictions but also a matter of how results can be analyzed and communicated in a feasible and meaningful way. Already now, with five internal variables from twelve

models and three catchments, the sheer amount of results produced makes it difficult to identify and communicate the most relevant points. Extending such a study to say 10 or 50 catchments will add an additional layer of results, which needs to be interpreted and discussed in addition. This will lead to a very unfocussed paper, in which the reader will struggle to find in-depth results. Such trade-offs in the analyzed factors are common in comparison studies and we are in fact not aware of any study that combines an analysis of many models with many variables and many catchments. For example, in their comparison study, Holländer et al. (2009) used ten models in one artificial catchment and assessed evaporation and discharge. The distributed model intercomparison project (DMIP; Smith et al. 2012) worked with 16 models, 17 catchments but mainly assessed streamflow and soil moisture. Noh et al. (2015) and Koch et al. (2016) compared three models with respect to seasonal variability of soil moisture. Orth et al. (2015) used three models to assess streamflow and soil moisture in eight catchments. Le Moine et al. (2007) used two models in 1040 catchments to focus on intercatchment groundwater flows. Rakovec et al. (2016) studied three internal state and flux variables in 400 catchments using a single model. Very recently, Knoben et al. (2020) investigated differences in performance of 36 models in 559 catchments with respect to streamflow as single variable. Their analyses are based on general performance metrics of daily streamflow. The conclusions remain general due to the considerable volume of data produced, allowing for less detail on process-relevant insights. Each of these studies has a specific focus and this is similar for our study. To our knowledge, a study with strong focus on internal model dynamics for multiple models in more than one catchment has not been done in this way before. We deliberately chose to balance depth with breadth and perform a thorough analysis of the set of **twelve models** and **five variables** in the **three catchments** in this study. We will stress this motivation in the revised version of the manuscript.

Comment 2:
*The study addresses different storages, including snow storage. However, the impor-*

*tance of snow in the test catchments is minor. I could not find any information on the relative importance of snow (the info of about one month of snow cover is incomplete as this does not say anything about the amount of water stored as snow). Still, my general understanding is that snow does not play any major role in these catchments. This is probably also the reason why the authors can get away with not using any elevation zones for modelling snow processes.*

Reply 2:
We agree that snow is not a major component of the streamflow regime within these catchments (as briefly mentioned in Section 2). Most of the precipitation falls as rain and the models have high streamflow model performance even without including a snow module. The amount of water stored as snow is shown in Figure 5b and 5c of the manuscript with maximum annual amounts of less than 20 mm. Despite these relatively small amounts, snow can be very important for specific events. In 2011, rain on snow caused widespread flooding in these catchments. The elevation range of the Ourthe Orientale upstream of Mabompré, where we are evaluating snow processes, is approximately 370 m (from 294 m to 662 m). We believe this can be treated as a single elevation zone, in particular as 65% of the catchment falls into a narrow 100 m elevation band (see Fig. 1 of this reply). In any case, given the absence of detailed observed temperature lapse rates, the assumption of a stable environmental lapse rate of e.g. 0.6 degree C/100m, required for an elevation stratification remains very speculative and thus not really warranted by the available data. We will clarify this in the revised version of the manuscript.

Comment 3:
*Each of the storage estimation used for model testing is associated with significant observation uncertainties. There is also a scale-mismatch which results in additional uncertainties. These issues have to be considered!*

Reply 3:
We completely agree that there is considerable and effectively mostly unknown uncertainty in the used remote sensing data. This was the underlying reason why we did not use the remote sensing products for model calibration nor for any type of quantitative model evaluation. We rather only treated these data as additional information against which to indicatively compare the modeled internal state and flux variables. We cannot and do not consider the remote sensing data to be a reliable representation of real-world quantities. However, they are useful to detect potential outliers. The uncertainty associated to remote sensing data should not restrain us from using them at all. However, we will clarify the uncertainties associated to the use of each remote sensing product in the revised version of the manuscript. In the next version of the manuscript, we will define a set of soft criteria to evaluate not only how consistent the model internal dynamics are amongst each other, but also if they provide a consistent behavior with what we expect from expert knowledge and remote sensing data.

Comment 4:
*Another point that seems to be missing is that each of the models of course also is affected by parameter uncertainties (which will influence the simulated storages). Perhaps I am missing something, but as I understand, single parameter sets are used for each model. This is not sufficient; we know that the same model can result in very different internal simulations because of parameter uncertainty. This leaves me wondering how much of the differences presented here are due to parameter uncertainty rather than due to model differences.*

Reply 4:
We absolutely agree that parameter uncertainty can cause differences in model

internal behaviour. Therefore, we of course use an ensemble of feasible parameter sets to account for parameter uncertainty. This is briefly mentioned in Section 4.1 and we will clarify this further in the revised version of the manuscript. The error bars and/or boxplots and/or ensemble of lines in Figure 3, 4, 5, 6, 7, 8 and 9 represent the ensemble of feasible parameter sets, as also mentioned in each caption. The narrower uncertainty ranges of some models are related to the use of different search strategies of the parameter space (see caption of Figure 8).

References
Holländer, H. M., Blume, T., Bormann, H., Buytaert, W., Chirico, G. B., Exbrayat, J. F., et al. (2009). Comparative predictions of discharge from an artificial catchment (Chicken Creek) using sparse data. *Hydrology and Earth System Sciences*, 13(11), 2069–2094. https://doi.org/10.5194/hess-13-2069-2009

Knoben, W. J. M., Freer, J. E., Peel, M. C., Fowler, K. J. A., Woods, R. A. (2020). A brief analysis of conceptual model structure uncertainty using 36 models and 559 catchments. *Water Resources Research*. https://doi.org/10.1029/2019WR025975

Koch, J., Cornelissen, T., Fang, Z., Bogena, H., Diekkrüger, B., Kollet, S., Stisen, S. (2016). Inter-comparison of three distributed hydrological models with respect to seasonal variability of soil moisture patterns at a small forested catchment. *Journal of Hydrology*, 533, 234–249. https://doi.org/10.1016/j.jhydrol.2015.12.002

Le Moine, N., Andréassian, V., Perrin, C., Michel, C. (2007). How can rainfall-runoff models handle intercatchment groundwater flows? Theoretical study based on 1040 French catchments. *Water Resources Research*, 43(6), 1–11. https://doi.org/10.1029/2006WR005608

Orth, R., Staudinger, M., Seneviratne, S. I., Seibert, J., Zappa, M. (2015). Does model performance improve with complexity ? A case study with three hydrological models. *Journal of Hydrology*, 523, 147–159. https://doi.org/10.1016/j.jhydrol.2015.01.044

Noh, S. J., An, H., Kim, S., Kim, H. (2015). Simulation of soil moisture on a hillslope using multiple hydrologic models in comparison to field measurements. *Journal of Hydrology*, 523, 342–355. https://doi.org/10.1016/j.jhydrol.2015.01.047

Rakovec, O., Kumar, R., Mai, J., Cuntz, M., Thober, S., Zink, M., et al. (2016). Multiscale and multivariate evaluation of water fluxes and states over european river Basins. *Journal of Hydrometeorology*, 17(1), 287–307. https://doi.org/10.1175/JHM-D-15-0054.1

Smith, M. B., Koren, V., Zhang, Z., Zhang, Y., Reed, S. M., Cui, Z., et al. (2012). Results of the DMIP 2 Oklahoma experiments. *Journal of Hydrology*, 418–419, 17–48. https://doi.org/10.1016/j.jhydrol.2011.08.056
* * *
[Figure]

[Figure]

**Fig. 1.** Elevation contour lines of the Ourthe Orientale upstream of Mabompré

---

## Referee Report (RR1)

**Review of Manuscript**

**'Behind the scenes of streamflow model performance' (hess-2020-176)**

**by L. J. E. Bouaziz et al.**

Dear Editor, dear Authors,

I have reviewed the aforementioned work. My conclusions and comments are as follows:

**1. Scope**

The article is within the scope of HESS.

**2. Summary**

The authors present an evaluation of a set of twelve conceptual hydrological models - all set up in the same Belgian watersheds. The key aspect of the study is that the models were calibrated against streamflow only, and are now evaluated in terms of several criteria (water balance, streamflow characteristics, runoff coefficient, evapotranspiration, snow storage, root-zone storage, total storage) against remote-sensing data taken as reference 'truth' - except discharge, which is based on local water level observations. The main hypothesis is that the models, showing comparably good performance in terms of streamflow, should do so based on similar internal representations of internal states and fluxes. This hypothesis is tested by determining the (dis-)agreement of model outputs and the corresponding remote sensing and gauge data, and comparing these (dis-)agreements among models. The main findings are that substantial disagreements exist for most models and several criteria, and that these disagreements do not agree among models. Also, no single-best model could be identified, and, taking into account the considerable uncertainties of the ground truthing data, no model could be rejected. With respect to the main hypothesis, the authors therefore conclude that good model performance with respect to discharge at the catchment outlet does not guarantee realistic and unambiguous internal model workings.

For future studies, the authors recommend multi-data calibration and validation (to reduce the equifinality associated with calibration against streamflow only), and that despite their uncertainties, remote-sensing data can play an important role in this, especially the dynamical patterns they provide. They also advocate multi-model, multi-parameter approaches to reveal uncertainties related to model predictions, and taking in-situ measurements to inform process studies.

**3. Evaluation**

Altogether, the study was conducted and presented in a very thorough and accessible manner, and it is a good example of a collaborative effort. I particularly welcome how the authors made uncertainty (of the forcing and ground-truthing data, and of parameter identification during calibration) a central part of their study, and how they do a multi-criteria evaluation – including soft criteria - of their models to gain an as-complete-as-possible picture of model performance.

That said, the substantial weakness of the paper is that there is nothing really new to learn from it. The main hypothesis the authors address is in fact no hypothesis, as the answer to it has already been given in the literature many times. In fact, the authors provide in the introduction a very good literature overview on the inevitable equifinality of model parameter estimation when using discharge as the single, aggregate evaluation criterion (and hence the impossibility to correctly address the 'individual hypotheses', see p. 3 l 57), on approaches to tackle this by multi-criteria calibration, or by using multiple models.

Furthermore, all models used by the authors are conceptual hydrological 'bucket' models (see Fig. 2), there is neither a real bottom-up physics-based model involved, nor a mainly data-based (such as LSTMs or other). For this relatively narrow selection of models, the authors rightly state in the introduction (p 2 l 20-23) that the representations of particular hydrological processes are quite similar. So given this narrow range of models, what we can learn from comparing them also only has a narrow range of applicability.

Furthermore, given the simplicity of the models used, most of the deficits of particular models with respect to particular processes (e.g. FLEX-Topo root-zone storage falling completely dry, see p 16 l 498) can be directly inferred from their structural/functional setup.

Taken together, the conclusion with respect to the main hypothesis is not wrong, but it comes as no surprise, and likewise the recommendation for future studies have also been made elsewhere (in fact, the authors give a good account of the related literature in the discussion section, e.g. p. 18 l 556-557, p. 18 l 560-563, p. 19 l 585-588).

My points of concern relate to the very core of the study, so I do not think they can be solved by a major revision. So it is either 'reject' or 'publish-as-is'. My recommendation - despite the substantial deficits of the study – is that the study deserves publication as a thoroughly done example of state-of-practice hydrology.

One last comment: Given the nice set of data the authors have put together, it would be interesting to reverse the approach: If the models are calibrated on the 'individual hypotheses' only, using the available remote sensing data (and not discharge), how good will they perform with respect to the 'aggregated hypothesis' (discharge)? This could be useful for cases where remote-sensing data are available, but not discharge data (PUB).

Yours sincerely,

Uwe Ehret

---

## Referee Report (RR2)

Model evaluation is an essential and important step in any hydrological modelling study and is typically based on streamflow data. The study of Bouaziz et al. takes a different perspective, in the sense that it assesses internal states and fluxes (streamflow, evapotranspiration, root zone soil water content, and total water storage) of twelve models that were shown to have a comparable (acceptable) streamflow model performance for the Meuse basin. The idea of multi-variable evaluation is not novel, however, it is rarely conducted in such a detailed and extensive way as in this study. I personally like the set-up of the multi variable analysis and I think that the final outcome of this study clearly contributes to improving our knowledge on model evaluation. The current state of the manuscript reflects the vast amount of results related to a multi-model/ multi-variable study and I encourage the authors to invest some more time into analysing and presenting results in a more concise way. I hope that the comments below will be helpful for the authors to improve their study.

**Major comments:**

1. The four variables streamflow, evapotranspiration, root zone soil moisture content, and total water storage are used in very different ways to evaluate model performance. For example, i) the temporal resolution varies from multiple years, to one year, to one season, or to a single month; ii) some variables are evaluated using magnitude related metrics others using dynamic related metrics; and iii) a different number of aspects of a particular variable is analysed. This diversity can be confusing and makes it difficult to understand the big picture as one is confronted with something new in every figure. I highly recommend to think about a way to analyse and present the data/results in a more consistent way.
2. The manuscript describes and presents the methods and results in quite some detail. This makes it difficult to separate the key messages from the more "nice to know" aspects. For example, i) the manuscript contains a total of 46 (sub)figures. Do you really think every single sub(figure) is needed to tell your story?; ii) there are multiple text sections that could easily be shortened by removing details that are not important for understanding the story (see detailed comments). I strongly encourage the authors to better filter information in order to improve the clarity of the manuscript.
3. The results are often only presented for one of the three catchments, which is related to the rather qualitative model evaluation. Showing the results for all three catchments would enhance the robustness and relevance of the results.
4. The study has two main objectives, whereby the second one is to evaluate models using "soft" measures. I didn't have the feeling that the "soft" measures were really an aim of this study – they were rather a tool (that was actually only used in the very last figure). If the use of "soft" measures was one of the objectives, then I would argue that you would have to do some serious testing of this measure that includes a comparison against other typically used evaluation metrics.
5. Model calibration was based on hourly streamflow data, whereas the evaluation was based on data with daily or monthly resolution. Don't you think that the discrepancy in temporal resolutions will lead to an even stronger "overfitting" to streamflow? In other words, would the performance for the internal states and fluxes be better if the model was calibrated against daily streamflow? Furthermore, model evaluation was conducted in a different period than model calibration, which introduces additional uncertainty. Do you think the results would be different if you did the evaluation in the calibration period?

**Minor comments:**

P2 L36-45: This paragraph lists a lot of studies on multi-variable calibration. However, as the list is not complete, I recommend to use "e.g." when listing references.

P3 L66-74: The use of "we" is confusing. Please make sure that it is clear what was part of the previous study and what is part of the current study. Furthermore, this is one of these paragraphs that could be shortened, also because the same information will be repeated in the methods section.

P4 L83-107: The reference for the land cover information is missing.

P5 L117: "…follow the same approach to extend the dataset…". Do your refer to temperature or ET?

P8 L234-252: i) This is one of the sections that could be shortened. E.g., the information about the proxy-basin test or about the evaluation from 2001-2003 and 2008-2010 is not relevant for this study; ii) How many parameter sets did you have per model?; iii) How similar are the calibration and validation period in terms of streamflow, evapotranspiration, root zone soil moisture content, and total water storage?

P9 L275-280: The description of the confusion matrix is rather confusing. I would directly jump to L280, where you actually say what you will evaluate (i.e, "…ratio of days when snow observed by MODIS is correctly identified by the model,…").

P10 L289: Is SR the range of relative root-zone soil moisture or the relative range of root-zone soil moisture? The equation shows a ratio and not a range.

P11 L318: You later mention that your goal was not the reject models or to find the best one. However, ranking models inevitably leads to a comparison. Therefore, wouldn't it be better to make a binary classification, i.e., accept or reject models for a particular variable?

P11 L 324: The results section is often mixed with discussion. Examples are L346-349, L356-359, L39-440, L 465-467, L481-483, L500-503.

P17 L535-536: Where do you show that the calibration strategy influences model performance?

P19 L575: Is it spatial resolution or spatial coverage?

Fig. 1: I think Fig. 1c is not needed as your description in the text is very clear. The variable EA in Fig. 1c is not defined.

Fig. 2: I would suggest to add labels to the individual tanks and fluxes, because colours are not intuitive. Why do wflow_hbv and FLEX-Topo have a different number of elevation bands?

Fig. 3: Why is the range in model performance much larger for NSE,logQ than for NSE,Q? What are the feasible parameter set? How many are they? How representative is Fig. 3d for all the other models (or what do we learn from it)?

Fig. 4: I would chose boxplots to show the interception values to be consistent with evapotranspiration. It is not clear what exactly the bars show (i.e, is it the 25-50% quantile?).

Fig. 9: Could you turn this figure into a heatmap? For which catchment is this figure?

---

## Author Response (AR2)

*Dear Editor, dear Nadav Peleg,*

*We thank you for positively assessing our manuscript. In the revised version, we have reduced the information content of several figures and shortened/clarified several text sections. However, we have kept the diversity of our detailed analyses as this was a strength strongly emphasized by Uwe Ehret (Referee #4). We have emphasized in the Abstract, Introduction and Discussion that the novelty of our study lies in the comprehensive and systematic quantification of differences between multiple state and flux variables of multiple models. Furthermore, we added a supplement to show the heatmap of the ranked models for the selection of criteria, as suggested by the Referee #5.*

*Please find below our replies to the referees' comments and the second revision of our manuscript (with latest track-changes since the first revision).*

*Kind regards,*

*Laurène Bouaziz and co-authors*

*Referee #4 (Uwe Ehret)*

*We thank Dr.-Ing. Uwe Ehret for his positive and critical evaluation of our manuscript. We highly appreciate his recommendation to publish the manuscript and his suggestion for a follow-up study.*

*We agree with the overall evaluation of Dr.-Ing. Uwe Ehret. The main conclusions of the study are indeed not unexpected. It has been known for a long time that different models can exhibit substantial differences in their internal dynamics. This can have a considerable impact on the results and interpretations/conclusions drawn in many hydrological studies, while potentially not being clearly communicated. To our knowledge there are no studies that have simultaneously quantified the differences in several internal model dynamics for multiple models, in a comprehensive and systematic way. With this study, we want to further emphasize and raise awareness that these differences can affect subsequent interpretations in hydrological modeling studies/applications.*

*We have emphasized this in the Introduction, Abstract and Implication sections of the revised manuscript.*

*We thank the anonymous referee #5 for carefully reading our manuscript and his/her valuable suggestions that helped us improve the manuscript.*

**Comment 1**

The four variables streamflow, evapotranspiration, root zone soil moisture content, and total water storage are used in very different ways to evaluate model performance. For example, i) the temporal resolution varies from multiple years, to one year, to one season, or to a single month; ii) some variables are evaluated using magnitude related metrics others using dynamic related metrics; and iii) a different number of aspects of a particular variable is analysed. This diversity can be confusing and makes it difficult to understand the big picture as one is confronted with something new in every figure. I highly recommend to think about a way to analyse and present the data/results in a more consistent way.

**Reply 1**

*We agree with the referee that the data are used in very different ways to evaluate model performance. This enables us to give a diverse and detailed overview of model performance by getting the most out of the available data. The different temporal signatures to evaluate model performance highlight each a different aspect of the hydrological response. Additionally, each variable should be appreciated in a different way. Depending on the uncertainty of the remote sensing product, it is sometimes only possible to evaluate the dynamics and not the magnitudes. As also stressed by Referee #4, the thoroughly conducted analyses are making the strength of the study. This makes us reluctant to reduce the diversity of our analyses. However, we reduced the information content of several figures to make it more concise (see reply to comment 2).*

*In the revised manuscript we have now clarified in Section 4.5 that we are summarizing the main findings ('the big picture') in Figure 9. The detailed analysis for each of the five evaluated variables (streamflow, evaporation, snow, root-zone soil moisture and total water storage) are shown in a separate figure per variable (Figures 3 to 7). The interactions between these variables for a low-flow period (therefore excluding snow) are shown in Figure 8. The main overall findings from these analyses are then synthesized in Figure 9 together with the uncertainty estimates of the remote sensing data.*

**Comment 2**

The manuscript describes and presents the methods and results in quite some detail. This makes it difficult to separate the key messages from the more "nice to know" aspects. For example, i) the manuscript contains a total of 46 (sub)figures. Do you really think every single sub(figure) is needed to tell your story?; ii) there are multiple text sections that could easily be shortened by removing details that are not important for understanding the story (see detailed comments). I strongly encourage the authors to better filter information in order to improve the clarity of the manuscript.

**Reply 2**

*We agree and thank the referee for these suggestions. We have looked in detail at the various subplots and agree that several of them can be removed to make the figures more concise. We also agree with the suggestions of the referee to shorten some sections of the text, as outlined in the detailed comments.*

*In the revised version, we have removed the following subplots:*
- *Figure 1c: we agree with the referee that the description in the text is already very detailed.*
- *Figure 6b: this subplot can be removed as we mention in Section 3.3 that it is difficult to meaningfully compare the range of modeled and remote sensing root-zone soil moisture.*
- *Figure 7b: we now show the two GRACE pixels in Figure 1b.*
- *Figure 7d: the description of the difference in precipitation between both catchments is detailed in the text and does not necessarily need to be shown in the Figure.*

*Additionally, we have shortened some text sections (see replies to the detailed comments).*

*As also mentioned in Reply 1, we have clarified in Section 4.5 that Figure 9 is summarizing the main findings of model performance in relation to the remote sensing products and their uncertainty.*

**Comment 3**

The results are often only presented for one of the three catchments, which is related to the rather qualitative model evaluation. Showing the results for all three catchments would enhance the robustness and relevance of the results.

**Reply 3:**

*As also outlined in the discussion with Referee #3 (https://doi.org/10.5194/hess-2020-176-AC4), we have deliberately chosen to balance depth with breadth and perform a thorough analysis of twelve models and five variables in a limited number of catchments. The catchments are relatively similar, and the results are not very different for the two neighboring/nested catchments. Our study is therefore not a large sample study of many catchments, as there is only so much one can convey in a single study. By showing the results for mainly one catchment, we perform a thorough in-depth analysis. We are reluctant to show the results for the three catchments, as this would not contribute to making the study more concise, as was suggested by the referee in Comment 2 (above).*

**Comment 4**

The study has two main objectives, whereby the second one is to evaluate models using "soft" measures. I didn't have the feeling that the "soft" measures were really an aim of this study – they were rather a tool (that was actually only used in the very last figure). If the use of "soft" measures was one of the objectives, then I would argue that you would have to do some serious testing of this measure that includes a comparison against other typically used evaluation metrics.

**Reply 4**

*We agree that the soft measures are not an objective. The objective is to **evaluate** the models by introducing a set of soft measures, as described in the last paragraph of the introduction.*

*It is not entirely clear to us what the referee exactly means with the other typically used evaluation metrics. The measures shown in Figure 9 are either single values or binary metrics, which would not enable us to calculate 'typical' measures as Nash-Sutcliffe Efficiency. Instead, we are showing the deviation between model behavior and a specific aspect of the hydrological response using streamflow, remote-sensing data or expert knowledge.*

*Furthermore, we think this comment relates to the discussion we had with Prof. Keith Beven (Referee #1, https://doi.org/10.5194/hess-2020-176-AC1). The problem we are facing is that there is a large and often unknown uncertainty in the evaluation data. For this reason, we are highly reluctant to determine hard rejection thresholds from these data. We therefore opt for a soft evaluation of the relative merits of the several models by ranking them for a selection of criteria based on soft, expert judgement of the trustworthiness of the individual types of remote-sensing data. It is a compromise between not using these data at all due to the significant uncertainty and commensurability issues, versus using them in an over-confident way (e.g. for calibration purposes). With the introduced soft measures, we want to make the most of the available data.*

**Comment 5**

Model calibration was based on hourly streamflow data, whereas the evaluation was based on data with daily or monthly resolution. Don't you think that the discrepancy in temporal resolutions will lead to an even stronger "overfitting" to streamflow? In other words, would the performance for the internal states and fluxes be better if the model was calibrated against daily streamflow? Furthermore, model evaluation was conducted in a different period than model calibration, which introduces additional uncertainty. Do you think the results would be different if you did the evaluation in the calibration period?

**Reply 5**

*This is an interesting point of discussion raised by the referee. The fast response time (less than 1 day) in these catchments of the Belgian Ardennes explains our choice to calibrate the models using hourly data in the previous study (de Boer-Euser et al., 2017). We then evaluate model performance in an evaluation period outside the calibration period. We did not test the differences between modeled results from a calibration on daily data and a calibration on hourly data with aggregated results to daily values. However, considering the relatively high model performance in the evaluation period, aggregating the hourly values to daily values will necessarily yield a good representation of the hydrological response at the daily (or higher) time step as aggregating results to a coarser time step will always lead to a better performance by averaging out the most extreme errors.*

*Even if there is a mismatch between the time step used in our calibration and the coarser temporal resolution of the remote sensing data, this is still valuable information that we can use.*

*Furthermore, we argue that evaluating the results in an evaluation period outside the calibration period should provide a more robust overview of model performance. A rigorous model test strategy is the differential split-sample test, described e.g. by Klemeš (1986). While the original differential split-sample test, tests a model during a time period not used for calibration at a location different than the location used for calibration, we here adapt the strategy to test the model during a time period not used for calibration but not (or only partly) at a different location but rather with respect to different variables/signatures.*

**Comment**

P2 L36-45: This paragraph lists a lot of studies on multi-variable calibration. However, as the list is not complete, I recommend to use "e.g." when listing references.

**Reply**

*We agree and have added "e.g." to the listed references.*

**Comment**

P3 L66-74: The use of "we" is confusing. Please make sure that it is clear what was part of the previous study and what is part of the current study. Furthermore, this is one of these paragraphs that could be shortened, also because the same information will be repeated in the methods section.

*Reply:*

*Thank you for pointing out this specific paragraph, we completely agree to shorten this part to avoid repetition. We have adapted this in the revised version and clarified what was part of the previous study as opposed to the current study.*

**Comment**

P4 L83-107: The reference for the land cover information is missing.

**Reply**

*We agree and added the reference in the revised version.*

**Comment**

P5 L117: "…follow the same approach to extend the dataset…". Do your refer to temperature or ET?

**Reply**

*We have clarified in the revised version that we refer to the meteorological forcing dataset (precipitation, temperature and potential evaporation).*

**Comment**

P8 L234-252: i) This is one of the sections that could be shortened. E.g., the information about the proxy-basin test or about the evaluation from 2001-2003 and 2008-2010 is not relevant for this study; ii) How many parameter sets did you have per model?; iii) How similar are the calibration and validation period in terms of streamflow, evapotranspiration, root zone soil moisture content, and total water storage?

**Reply**

*i) We agree with this suggestion and have removed some details on the evaluation procedures of the previous study in the revised version of the manuscript.*

*ii) For each model, the modelers were asked to provide an ensemble of 20 feasible parameter sets, the selection method was not imposed by the protocol as long as it relied on the two objective functions (NSE and NSE_{logQ}).*

*iii) In Figure-A below, we show the observed streamflow time series for the Ourthe at Tabreux. The calibration years can be qualified as relatively normal years in terms of streamflow. Also in terms of the overall water balance, differences in the aridity index and runoff coefficient are minor, as shown in the representation of the Budyko framework in Figure-B. Time series of GLEAM evaporation for the period 2008-2017 (Figure 4b of the manuscript) show a relatively constant behavior, which is also representative of the calibration period. Time series of MODIS snow cover (Figure 5a) show that the calibration period contains both years with relatively much and little snow. GRACE total storage anomalies show a relatively constant behavior for the entire period and therefore also during the calibration period (Figure 7a). The three years where root-zone soil moisture is available are shown in Figure 8a,b, but can, however, not be compared with the calibration years. We also refer to the supplement of the previous study (de Boer-Euser et al. 2017) for additional time series of precipitation, evaporation and streamflow for the calibration and evaluation periods.*

[Figure]

*Figure-A: Observed streamflow between 2000 and 2017 for the Ourthe at Tabreux. The grey part of the hydrograph represents the calibration period.*

[Figure]

*Figure-B: Plotting position of the Ourthe catchment at Tabreux for the calibration and evaluation periods within the Budyko framework.*

**Comment**

P9 L275-280: The description of the confusion matrix is rather confusing. I would directly jump to L280, where you actually say what you will evaluate (i.e, "…ratio of days when snow observed by MODIS is correctly identified by the model,…").

**Reply**

*We fully agree with this suggestion and we shortened and clarified the explanation on the confusion matrix in the revised version of the manuscript. We kept the definitions of hits, miss, false alarm and correct rejections as all readers might not be familiar with these concepts. We then jumped into the two aspects that we show in Figure 5d and 5e. Instead of referring to 1-recall and 1-precision rates, we now refer to these entities as the miss rate and the false discovery rate to increase the clarity throughout the text and in the Figure.*

**Comment**

P10 L289: Is SR the range of relative root-zone soil moisture or the relative range of root-zone soil moisture? The equation shows a ratio and not a range.

**Reply**

*The variable $\overline{S_R}$ refers to the relative root-zone soil moisture, a dimensionless variable calculated as $S_R/S_{R,max}$. The description of this symbol is shown in Table 1. In our analyses we compare the range of the variable $\overline{S_R}$ for the different models. We understand the possible confusion and have tried to clarify this part in the revised version of the manuscript.*

**Comment**

P11 L318: You later mention that your goal was not the reject models or to find the best one. However, ranking models inevitably leads to a comparison. Therefore, wouldn't it be better to make a binary classification, i.e., accept or reject models for a particular variable?

**Reply**

*Here, we refer to the discussion we had with Prof. Keith Beven (Referee #1, https://doi.org/10.5194/hess-2020-176-AC1) and to our answer on Comment 4. Considering the significant uncertainty in the evaluation variables, we are reluctant to use these data to determine hard rejection thresholds. Instead, we deliberately chose for an approach where the models are compared through a ranking to evaluate how plausible it is to consider them behavioral based on the streamflow, remote-sensing data and expert knowledge.*

*Additionally, accepting or rejecting models for a particular variable would not be enough, as we would still not get the right answer for the right reasons if all other variables were rejected. Eventually, we look for models that get the overall response right by having a better performance for all variables, this also supports the chosen ranking approach.*

**Comment**

P11 L 324: The results section is often mixed with discussion. Examples are L346-349, L356-359, L39-440, L 465-467, L481-483, L500-503.

**Reply**

*We agree that some explanations could be perceived as discussion. It is always difficult to clearly separate results from discussion. To better guide the reader through the results, it is sometimes necessary to directly place them in a broader context. Some statements may also be very specific and would not need to be repeated in a separate Discussion section. To clarify this for the reader, we have adapted the 'Results' section to 'Results and Discussion'. The broader Implications and Knowledge gaps and limitations are described in the subsequent sections.*

**Comment**

P17 L535-536: Where do you show that the calibration strategy influences model performance?

**Reply**

*We agree that we do not explicitly show that the calibration strategy influences model performance. This statement reflects some discussions that we had amongst several modelers during our analyses. In order to keep the manuscript more concise, we have removed this statement from the revised version of the manuscript to avoid confusion.*

**Comment**

P19 L575: Is it spatial resolution or spatial coverage?

**Reply**

*This is an interesting point, both coverage and resolution apply here. We have clarified this in the revised version of the manuscript.*

**Comment**

Fig. 1: I think Fig. 1c is not needed as your description in the text is very clear. The variable EA in Fig. 1c is not defined.

**Reply**

*We agree with the suggestion of the referee to remove Figure 1c, as it is perhaps 'nice to have', but not strictly needed. We have removed Figure 1c in the revised version.*

**Comment**

Fig. 2: I would suggest to add labels to the individual tanks and fluxes, because colours are not intuitive. Why do wflow_hbv and FLEX-Topo have a different number of elevation bands?

**Reply**

*While we agree that adding labels to the individual tanks and fluxes would make the schematization more complete, we also think it will make this already busy figure less readable. Therefore, we decided to stick with the color scheme, which is intended to provide a quick overview of the different stores and fluxes of each model. For detailed model descriptions, we refer to the references provided in the text.*

*The parallel stores represented for the FLEX-Topo model represent three parallel hydrological response units (wetland, hillslope and plateau) connected through the groundwater reservoir. The parallel stores for wflow_hbv represent the distributed spatial discretization of the model (as also indicated in Table 2). We have clarified this in the caption of the Figure.*

**Comment**

Fig. 3: Why is the range in model performance much larger for NSE,logQ than for NSE,Q? What are the feasible parameter set? How many are they? How representative is Fig. 3d for all the other models (or what do we learn from it)?

**Reply**

*We agree that some models show a larger range of $NSE_{log,Q}$ than NSE. We have not analyzed this specifically in our manuscript. However, we see that the slope of the recession curve can show some substantial differences among feasible parameter sets. While this difference in slope does not affect the NSE because flows during the recession period are very low, it has a large influence on the $NSE_{logQ}$. This is illustrated in Figure-C, for a model with relatively large difference in the slope of the recession (FLEX-Topo) and a model with relatively similar slopes of the recession among the ensemble of feasible parameter sets (wflow_hbv).*

[Figure]

*Figure-C: Modeled streamflow (log-scale) for the 2010 and 2011 recession of the Ourthe at Tabreux for a model with slight variability in slope of the recession among feasible parameter sets (wflow_hbv) as opposed to a model with a larger spread in slope of the recession among feasible parameter sets (FLEX-Topo).*

*For each model, we retained an ensemble of 20 feasible parameter set based on a free to choose selection method relying on the two objective functions NSE and $NSE_{logQ}$.*

*We included Figure 3d to show the hydrograph of the studied catchment. The use of different signatures to aggregate the hydrological response is very valuable to evaluate model performance. However, we think a visual representation of the hydrograph cannot be omitted from the manuscript, as it can give the readers a quick general visual impression of the type of hydrological response in this catchment.*

**Comment**

Fig. 4: I would chose boxplots to show the interception values to be consistent with evapotranspiration. It is not clear what exactly the bars show (i.e, is it the 25-50% quantile?).

**Reply**

*We agree that showing boxplot would make it more consistent, but we are afraid that it might also make it more confusing as there is already a lot of information in the figure. As the bars are indicated in the legend, we have kept them in the revised version of the manuscript. We have clarified in the caption that the bars represent the minimum and maximum range.*

**Comment**

Fig. 9: Could you turn this figure into a heatmap? For which catchment is this figure?

**Reply**

*We thank you for this very nice suggestion. We have created a heatmap (Figure-D shown below) showing the ranks of the models for each of the criteria shown in Figure 9. We decided to add this figure in the supplement and keep Figure 9 in the main text, as Figure 9 provides not only information on the ranking, but also on the deviations from the remote-sensing data and the associated estimated uncertainty of each criterion.*

*We clarified in the caption of Figure 9 that the results are shown for the Ourthe at Tabreux except for the snow analysis which is shown for the Ourthe Orientale at Mabompré.*

[Figure]

*Figure-D: Model ranks for each of the criteria shown in Figure 9. The results are shown for the Ourthe at Tabreux (ID 1), except for the snow days analysis which is shown for the Ourthe Orientale at Mabompré (ID 2). Blank spots indicate that the models do not include a separate interception module or/and a snow module.*

*References:*

[revised manuscript text omitted]

---

## Author Response (AR3)

*Dear Editor, dear Nadav Peleg,*

*We thank you for the assessment of our manuscript. In the revised version of the manuscript, we have clarified the three minor comments made by Referee #5. Please find below our replies to these three comments and a track change version of the revised manuscript, with main changes in Section 4.1 (Models and protocol). We also include a revised version of the Supplement, where an additional section S1 has been added to describe similarities in the hydro-climatic characteristics of the calibration and evaluation periods.*

*Best wishes for the holiday season,*

*Laurène Bouaziz and co-authors*

*Referee #5*

*We thank the Referee #5 for re-evaluating our manuscript and his/her positive words. We agree with the three suggestions to improve the manuscript and have incorporated them in the revised version.*

**Comment 1**

You mention in your response that 20 acceptable parameter sets were retained after calibration for the multi-variable evaluation of the model. I couldn't find this number anywhere in the manuscript. Please add it somewhere in the calibration section as it is quite an important difference whether one uses e.g. 1, 10, 50, or 100 accepted parameter sets.

**Reply 1**

*Indeed, we have now specified this number in the Methods, Section 4.1 (Models and Protocol).*

**Comment 2**

It was not clear to me from reading the manuscript why results are often presented for only one of the three catchments. I think it is important that you explain the reader your reasons for presenting the results for a particular catchment. One sentence at the end of the models and protocols section should suffice for that.

**Reply 2**

*Thank you for this suggestion, we have now clarified in the revised manuscript (Section 4.1 Models and protocol) that the analyses are mainly presented for the Ourthe at Tabreux (ID1) as this was the catchment used for calibration. However, catchment ID2 is used for the snow analysis due to the narrower elevation range of this catchment. The spatial variability of total storage anomalies is compared using data from ID1 and ID3 catchments.*

**Comment 3**

You provided a quite detailed response to the hydroclimatic similarity/difference of the calibration and validation period. I think it is important to include this response in the manuscript as the similarity of the two periods helps to make an even stronger point on the need for a multi-variable model evaluation.

**Reply 3**

*We agree with this suggestion and included a sentence in Section 4.1 (Models and protocols) on the hydro-climatic similarity between the calibration and post-calibration periods, in which we refer to a new section S1 of the Supplement for a more detailed description. In the Supplement, we added the two figures previously used in our reply, which show the streamflow hydrograph and both calibration and evaluation periods in the Budyko space.*

[revised manuscript text omitted]

---

## Author Response (AR4)

*Dear Editor, dear Nadav Peleg,*

*We thank you for handling and accepting our manuscript for publication in HESS. We have uploaded all files and added the DOI and reference of the dataset with all modeled state and flux variables available in the online 4TU repository (see Data availability section at the end of the manuscript).*

*Best wishes for the new year,*

*Laurène Bouaziz and co-authors*